# Whole-genome doubling drives oncogenic loss of chromatin segregation

Ruxandra A. Lambuta[1,2,5], Luca Nanni[2,3,4,5], Yuanlong Liu[2,3,4], Juan Diaz-Miyar[1,2], Arvind Iyer[2,3,4], Daniele Tavernari[2,3,4], Natalya Katanayeva[1,2], Giovanni Ciriello[2,3,4 ✉] & Elisa Oricchio[1,2 ✉]

Whole-genome doubling (WGD) is a recurrent event in human cancers and it promotes chromosomal instability and acquisition of aneuploidies[1–8]. However, the three-dimensional organization of chromatin in WGD cells and its contribution to oncogenic phenotypes are currently unknown. Here we show that in p53-deficient cells, WGD induces loss of chromatin segregation (LCS). This event is characterized by reduced segregation between short and long chromosomes, A and B subcompartments and adjacent chromatin domains. LCS is driven by the downregulation of CTCF and H3K9me3 in cells that bypassed activation of the tetraploid checkpoint. Longitudinal analyses revealed that LCS primes genomic regions for subcompartment repositioning in WGD cells. This results in chromatin and epigenetic changes associated with oncogene activation in tumours ensuing from WGD cells. Notably, subcompartment repositioning events were largely independent of chromosomal alterations, which indicates that these were complementary mechanisms contributing to tumour development and progression. Overall, LCS initiates chromatin conformation changes that ultimately result in oncogenic epigenetic and transcriptional modifications, which suggests that chromatin evolution is a hallmark of WGD-driven cancer.

WGD is defined by the duplication of the entire set of chromosomes within a cell. It has been observed in early and pre-malignant lesions of various tissues[2,9,10], and it is estimated to occur in approximately 30% of human cancers[3]. WGD favours the acquisition of chromosomal alterations[5–8] in permissive genetic backgrounds, such as in p53- or Rb-deficient cells[3,4], which may promote tumorigenesis[1,5]. However, tetraploidization in single nuclei is equally likely to induce alterations in the three-dimensional (3D) structure and epigenetic features of the chromatin. During interphase, chromatin is organized in a multilayer 3D architecture of compartments, chromatin domains, and loops[11–16], and is closely associated with chromatin activity and cell states[17]. Alterations of the chromatin structure have been reported in many tumour types and are due to altered CTCF or cohesin binding[18,19], chromosome structural variants[20–22] or aberrant histone modifications[22–25]. Here we investigate how chromatin is organized in cells that undergo WGD. Moreover, we study which features of the chromatin structure are affected by WGD and whether changes in chromatin organization emerge and affect cell phenotypes after WGD. Finally, we examine whether these changes correlate with genetic and epigenetic alterations in WGD-driven tumours.

## WGD results in LCS

To understand the impact of WGD on chromatin organization and tumour development, we used three distinct cellular models: (1) the non-transformed diploid cell line hTERT-RPE1 (hereafter referred to as RPE); (2) CP-A cells derived from a patient with Barrett's oesophagus, a pre-cancerous condition that predisposes to oesophageal adenocarcinoma development through WGD[9,26]; and (3) the leukaemic near triploid K562 cell line. To mimic the permissive genetic background observed in human tumours[3,26], we used p53-deficient CP-A (Extended Data Fig. 1a) and RPE cells (previously termed RPE[TP53–/–] cells)[27]. K562 cells already harbour a loss-of-function mutation in the *TP53* gene[28]. WGD cells were obtained through mitotic slippage in two independent CP-A *TP53*[–/–] clones (clone 3 and clone 19) and K562 cells, and through cytokinesis failure using two distinct protocols in RPE *TP53*[–/–] cells (Fig. 1a). To control for chromatin conformation changes associated with chromosomal instability (CIN) but not WGD, we induced CIN in RPE *TP53*[–/–] cells using a MPS1 inhibitor (Fig. 1a). Cell cycle and karyotype analyses confirmed that the number of chromosomes doubled after treatment in most cells (Fig. 1b,c and Extended Data Fig. 1b–g) and that the nuclear size increased (Extended Data Fig. 1h).

Conversely, treatment with the MPS1 inhibitor did not change the ploidy of the RPE *TP53*[–/–] cell population (hereafter, CIN-only RPE *TP53*[–/–]), but the cells exhibited a variable number of chromosomes (Fig. 1c). Hence, we analysed chromatin organization before and after WGD induction through high-throughput chromatin conformation capture (Hi-C) analysis in all models (Supplementary Fig. 1). Despite the doubling of the genome, chromatin organization was highly similar between diploid and tetraploid cells (Extended Data Fig. 2a–e). However, the ratios of the observed number of contacts compared with the expected number of contacts at each locus indicated that the

[1]Swiss Institute for Experimental Cancer Research (ISREC), School of Life Sciences, EPFL, Écublens, Switzerland. [2]Swiss Cancer Center Leman, Lausanne, Switzerland. [3]Department of Computational Biology, University of Lausanne (UNIL), Lausanne, Switzerland. [4]Swiss Institute of Bioinformatics (SIB), Lausanne, Switzerland. [5]These authors contributed equally: Ruxandra A. Lambuta, Luca Nanni. ✉e-mail: giovanni.ciriello@unil.ch; elisa.oricchio@epfl.ch

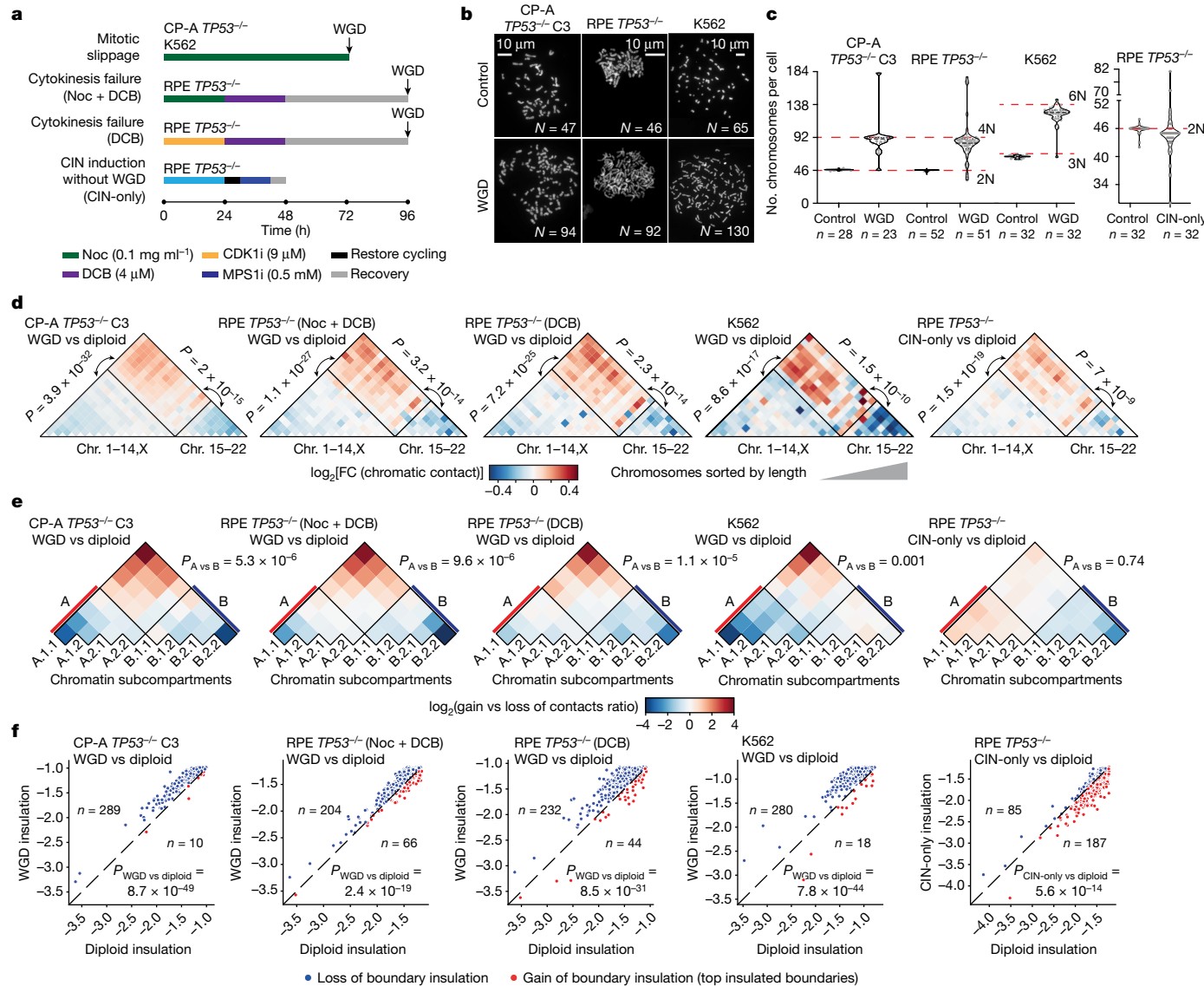

**Fig. 1 | WGD induces LCS. a**, Schematic representation of the WGD and CIN induction experimental approaches in the indicated cell lines. DCB, dihydrocytochalasin B; Noc, nocodazole. **b**, Representative images of metaphase spreads for CP-A *TP53*[−/−] clone 3 (C3), RPE *TP53*[−/−] and K562 cell lines. **c**, Quantification of chromosomes per cell for CP-A C3 *TP53*[−/−], RPE *TP53*[−/−] and K562 cell lines. The number of cells is indicated. For the violin plots, dashed line is the median, dotted lines are quartiles. **d**, Heatmap of the ratios of interchromosomal contact enrichments (observed versus expected) between WGD and control samples for CP-A *TP53*[−/−] cells, RPE *TP53*[−/−] cells and K562 cells and between RPE *TP53*[−/−] CIN-only cells and control cells. Chromosomes were sorted by length. **e**, Heatmap of ratios of genomic bins belonging to the indicated subcompartments that gain or lose contacts in the indicated conditions. **f**, Boundary insulation scores in control cells and WGD or CIN-only cells in the indicated cell lines for the shared top insulating boundaries. For **f**–**h**, *P* values were calculated using two-tailed Wilcoxon test.

enrichment and depletion of contacts were lower in WGD cells than in diploid cells. By contrast, these ratios remained similar in CIN-only and diploid cells (Extended Data Fig. 2f,g). In particular, the number of contacts within a domain or compartment decreased, whereas the number of contacts between different domains and compartments increased. To further investigate the changes in chromatin contact distribution in WGD cells, we assessed the following parameters: (1) changes in contacts between the clusters of long and short chromosomes[11]; (2) contact enrichment within A and B subcompartments, which we inferred using the Calder algorithm[29]; and (3) contact insulation at topologically associating domain (TAD) boundaries[30]. In all WGD-induction models and independent replicates, compared with control cells, WGD cells consistently exhibited the following characteristics: (1) a significantly increased proportion of contacts between long chromosomes (1–14 and X) and short chromosomes (15–22) (Fig. 1d and Extended Data

Fig. 3a,b); (2) a significantly increased proportion of contacts between A and B compartments, especially between the most distant A.1.1 and B.2.2 (Fig. 1e and Extended Data Fig. 3c); and (3) decreased boundary insulation (Fig. 1f and Extended Data Fig. 3d). These effects were only moderately detectable or absent in CIN-only cells (Fig. 1d–f) and did not depend on the resolution of the Hi-C experiment, coverage per haploid copy or ratios of contacts between homologous copies of the same chromosomes (Extended Data Fig. 3e,f).

Gene expression analysis comparing RPE *TP53*[−/−] WGD cells and control cells revealed an overall upregulation of transcription in WGD cells (Extended Data Fig. 4a). Significantly upregulated genes (*n* = 1,268, log$_2$(fold change (FC)) > 1, adjusted *P* < 0.01) were enriched in the interferon signalling pathway (Extended Data Fig. 4b), which is consistent with responses to abnormal mitotic segregation and stress[31]. Conversely, significantly downregulated genes (*n* = 619, log$_2$(FC) < −1,

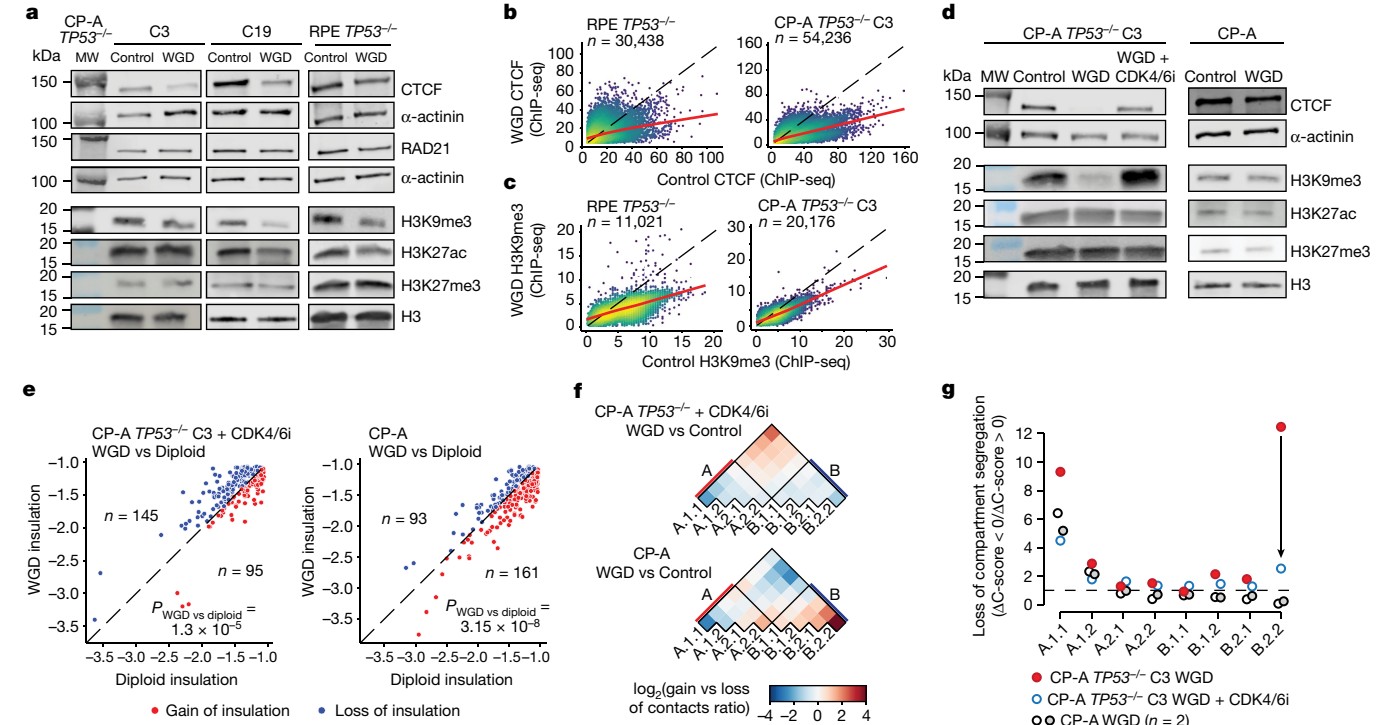

**Fig. 2 | Reduced levels of CTCF and H3K9me3 drive LCS. a**, Representative images of immunoblots of indicated proteins in diploid control and WGD CP-A *TP53*[−/−] (clone 3 and clone 19 (C19)) and RPE *TP53*[−/−] cells. MW, molecular weight **b,c**, ChIP−eq signal (FC over input) of CTCF (**b**) and H3K9me3 (**c**) shared peaks between control and WGD conditions in CP-A *TP53*[−/−] clone 3 and RPE *TP53*[−/−] cells. Dots are coloured by point density in log₁₀ scale, and the regression line is in red. **d**, Representative immunoblots of indicated proteins in diploid control and WGD CP-A cells and in WGD CP-A *TP53*[−/−] cells treated with a CDK4/6i.

**e**, Boundary insulation scores in control and WGD cells for the shared top insulating boundaries between the indicated conditions. *P* values were calculated using two-tailed Wilcoxon test. **f**, Heat map of ratios of genomic bins belonging to the indicated subcompartments that gain versus lose contacts in the indicated conditions. **g**, Loss of compartment segregation score relative to the control condition in each subcompartment domain in the indicated cell lines.

adjusted *P* < 0.01) were enriched in the DNA replication, DNA repair and cell cycle pathways (Extended Data Fig. 4c), which is consistent with downregulation of DNA replication proteins in WGD cells[7]. Changes in expression were not associated with changes in compartment segregation and only moderately with boundary loss of insulation (Extended Data Fig. 4d,e), which indicated that these changes mostly reflected an acute cell response to WGD. In summary, WGD cells, but not CIN-only cells, exhibit LCS manifested in an increased proportion of contacts between long and short chromosomes, distinct chromatin subcompartments, and TADs.

## CTCF and H3K9me3 deficiency determines LCS

Increased contact frequency among long and short chromosomes could be associated with the doubled number of homologous chromosomes. Therefore, we investigated causes of boundary insulation loss and loss of compartment segregation. CTCF and cohesin are crucial proteins for maintaining insulation at TAD boundaries[32,33], whereas enrichment of specific histone marks is associated with chromatin compartmentalization[14,29]. CP-A *TP53*[−/−] cells and RPE *TP53*[−/−] cells that underwent WGD exhibited an approximately 50% reduction in CTCF and H3K9me3 compared with control cells. WGD cells also had a modest decrease in H3K27ac, but no consistent changes in H3K27me3 or the cohesin complex component RAD21 (Fig. 2a and Extended Data Fig. 4f). *CTCF* mRNA abundance was also lower in WGD cells than in diploid control cells (log₂(FC) = −0.6, adjusted *P* = 8.3 × 10[−6]) (Supplementary Table 1). Chromatin immunoprecipitation with high-throughput and sequencing (ChIP−seq) analysis of CTCF showed that WGD cells and diploid cells shared the majority of CTCF peaks (Extended Data Fig. 4g);

however, these peaks typically exhibited lower signal (input-normalized number of reads) in WGD cells than in diploid cells (Fig. 2b and Extended Data Fig. 4h). TAD boundaries that lost insulation showed lower CTCF abundance and fewer numbers of CTCF peaks compared with boundaries that retained or even gain insulation in WGD cells (Extended Data Fig. 4i). This result suggests that reduced CTCF protein levels lead to a stochastic loss of CTCF binding, which in turn results in loss of insulation at boundaries with few CTCF binding sites. In parallel, ChIP−seq analysis of H3K9me3 levels confirmed an overall reduction in WGD cells, particularly at regions that originally exhibited high H3K9me3 levels (Fig. 2c) and in the B.2.2 subcompartment, which is usually enriched for this histone mark (Extended Data Fig. 4j).

In our models, the lack of p53 creates a permissive genetic background that allows WGD cells to bypass the tetraploid checkpoint, tolerate DNA damage and continue to grow[1,4,34]. Thus, we tested whether the lack of checkpoints and uncontrolled proliferation of *TP53*[−/−] cells contribute to the inability of WGD cells to increase CTCF and H3K9me3 levels. First, we induced WGD in *TP53* wild-type cells that activate the tetraploid checkpoint, which stalls cells in the G1 cell cycle phase. Second, we induced WGD in CP-A *TP53*[−/−] cells treated with an inhibitor of CDK4 and CDK6 (CDK4/6i), palbociclib, which leads to a prolonged G1 phase (Extended Data Fig. 4k). We successfully induced WGD in *TP53* wild-type CP-A cells (Extended Data Fig. 4l–n), whereas most *TP53* wild-type RPE cells remained binucleated after treatment with nocodazole and dihydrocytochalasin B and could not be used for further analyses (Extended Data Fig. 4o). In *TP53* wild-type cells, normalized CTCF and H3K9me3 levels were comparable between WGD and diploid cells, and treatment with palbociclib was sufficient to rescue CTCF and H3K9me3 levels in *TP53*[−/−] WGD cells (Fig. 2d and Extended Data

Fig. 4p). After rescuing CTCF and H3K9me3 levels, loss of insulation at TAD boundaries and loss of compartment segregation was strongly reduced or completely absent (Fig. 2e,f) compared with what was observed in CP-A *TP53*[−/−] WGD cells (Fig. 1e,f), in particular within the B.2.2 subcompartment (Fig. 2g). Loss of segregation between long and short chromosomes remained detectable in *TP53* wild-type cells and in WGD cells treated with palbociclib (Extended Data Fig. 4q). This result indicates that this effect is independent of p53, CTCF and H3K9me3 status, and is probably due to the doubled number of chromosomes.

Although *TP53* loss was required to induce LCS after WGD, LCS was not detectable when comparing diploid *TP53*[−/−] cells with diploid *TP53* wild-type cells (Extended Data Fig. 5a–c). CTCF protein expression was also retained (Extended Data Fig. 5d), which is not a direct target of *TP53* (Extended Data Fig. 5e). These data indicate that activation of the p53-dependent tetraploid checkpoint is important to increase protein production and to maintain chromatin conformation and epigenetic status in WGD cells.

## LCS is detectable in WGD single cells

Next we asked whether loss of segregation among chromosomes, compartments, and chromatin domains could also be detected in single cells. We performed single-cell Hi-C (scHi-C) in RPE *TP53*[−/−] diploid cells and WGD cells by isolating individual nuclei from the two cell populations (Supplementary Fig. 2). scHi-C libraries were prepared from 73 individual nuclei, and, after sequencing, we retained 33 control cells and 25 WGD cells (Supplementary Table 2; mean number of contacts per cell = 565,324). Aggregating scHi-C profiles (pseudo-bulk) reproduced the enrichment patterns observed in bulk RPE *TP53*[−/−] Hi-C data (Extended Data Fig. 6a). After comparing the number of contacts between long and short chromosomes and among long chromosomes and short chromosomes, a subpopulation of cells exclusively detectable in the WGD group exhibited an increased proportion of interactions among long and short chromosomes (Fig. 3a). Short–short chromosome contacts were significantly enriched compared with long–short chromosome contacts in cells that did not exhibit LCS, but this difference was no longer detectable in LCS-exhibiting WGD (LCS-WGD) cells (Fig. 3b, Supplementary Fig. 2 and Supplementary Table 2). By ranking chromosome pairs on the basis of their total number of interchromosomal contacts, chromosome 10 and the X chromosome scored at the top in both WGD cells and control cells, consistent with a t(10,X) translocation reported in RPE cells[35] (Fig. 3c). Notably, long–short chromosome pairs obtained lower ranks than short–short chromosome pairs in control cells, but not in LCS-WGD cells. For these cells, the top scoring chromosome pairs included pairs such as chromosome 1–chromosome 16 and chromosome 5–chromosome 15 (Fig. 3c).

Next we inferred A and B compartments in single cells to assess compartment segregation. LCS-WGD cells also exhibited significantly reduced compartment segregation (Fig. 3d), which indicated that the LCS features observed at the population level are intrinsically present within this group of single cells. The sparsity of the scHi-C data did not enable the assessment of boundary insulation. Last, copy number variants (CNVs) inferred from scHi-C coverage showed that LCS in WGD cells did not associate with the number of CNVs or the fraction of the genome altered (FGA) (Extended Data Fig. 6b,c). Hence, LCS could be detected in single cells and did not depend on CNV acquisition.

## Genomic evolution of WGD cells

Following WGD, both RPE *TP53*[−/−] cells and CP-A *TP53*[−/−] cells showed a transition to a heterogenous and aneuploid karyotype within 48 h (Extended Data Fig. 7a–c), which is consistent with WGD and loss of p53 favouring aneuploidy and CIN[1,9,36]. As early as 24 h after WGD (post-WGD), we detected CIN characteristics, such as chromosome

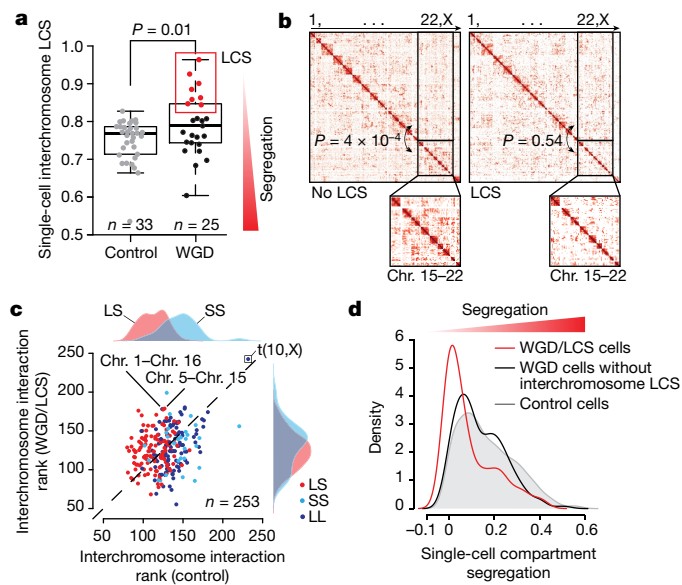

**Fig. 3 | LCS in WGD single cells. a**, Interchromosome LCS score calculated for each cell. Black dots indicate WGD cells not exhibiting LCS, red dots indicate WGD cells exhibiting LCS. For the boxplots, the central line is the median, the bounding box corresponds to the 25–75th percentiles and the whiskers extend up to 1.5-times the interquartile range. *P* value was calculated using two-tailed Wilcoxon test. **b**, Representative Hi-C maps at 10 Mb resolution of WGD RPE *TP53*[−/−] cells with (right) and without LCS (left). Short–short (SS) and long–short (LS) chromosome contacts were compared using two-tailed Wilcoxon test. **c**, Average interchromosomal interaction rank across single cells for each pair of chromosomes in control and WGD-LCS cells. The chromosome pair (10,X) is highlighted. **d**, Single-cell compartment segregation score distribution.

breakages and telomere fusions in CP-A *TP53*[−/−] cells (Extended Data Fig. 7d), and multipolar spindles and bipolar division with clustered centrosomes in RPE *TP53*[−/−] cells (Extended Data Fig. 7e). To elucidate the evolution of CIN and chromatin 3D structures in post-WGD cells, we analysed genomic and chromatin conformation changes in RPE *TP53*[−/−] cell populations at different time points in vitro (up to 20 weeks) and in vivo (Fig. 4a). At 6 weeks post-WGD, RPE *TP53*[−/−] cells in vitro exhibited heterogeneous ploidy, whereas the population became nearly diploid at 20 weeks post-WGD (Fig. 4b). At these two time points, cells were subcutaneously injected into immunocompromised mice. All animals engrafted with RPE *TP53*[−/−] cells at 6 weeks post-WGD (*n* = 12) or 20 weeks post-WGD (*n* = 6) developed tumours within 2.5 and 1.5 months, respectively (Fig. 4c and Extended Data Fig. 8a). By contrast, RPE *TP53*[−/−] diploid cells did not induce tumorigenesis (Fig. 4c), which indicated that the oncogenic capacity of these cells was acquired after WGD.

We next performed whole-genome sequencing (WGS) analyses of in vitro and in vivo post-WGD samples. The data showed that the number of acquired mutations in 6-weeks post-WGD RPE *TP53*[−/−] cells was about 1.8-times higher than in control cells kept in culture for the same amount of time, and post-WGD mutations had lower variant allele frequencies (Extended Data Fig. 8b). Across all samples, we detected a heterozygous clonal *NRAS* Q61R mutation (variant allele frequency > 0.4), which is a known oncogenic variant[37]. Nevertheless, this mutation was already present in RPE *TP53*[−/−] cells before WGD, and it was not sufficient to induce tumorigenesis in mice (Fig. 4c). Conversely, mutations that were acquired post-WGD did not include known oncogenic variants (Supplementary Table 3).

RPE *TP53*[−/−] diploid cells and WGD cells exhibited a nearly unaltered genome (FGA < 1%), except for a shallow loss on chromosome 13p (Fig. 4d and Supplementary Table 3). In vitro samples at 6-weeks and

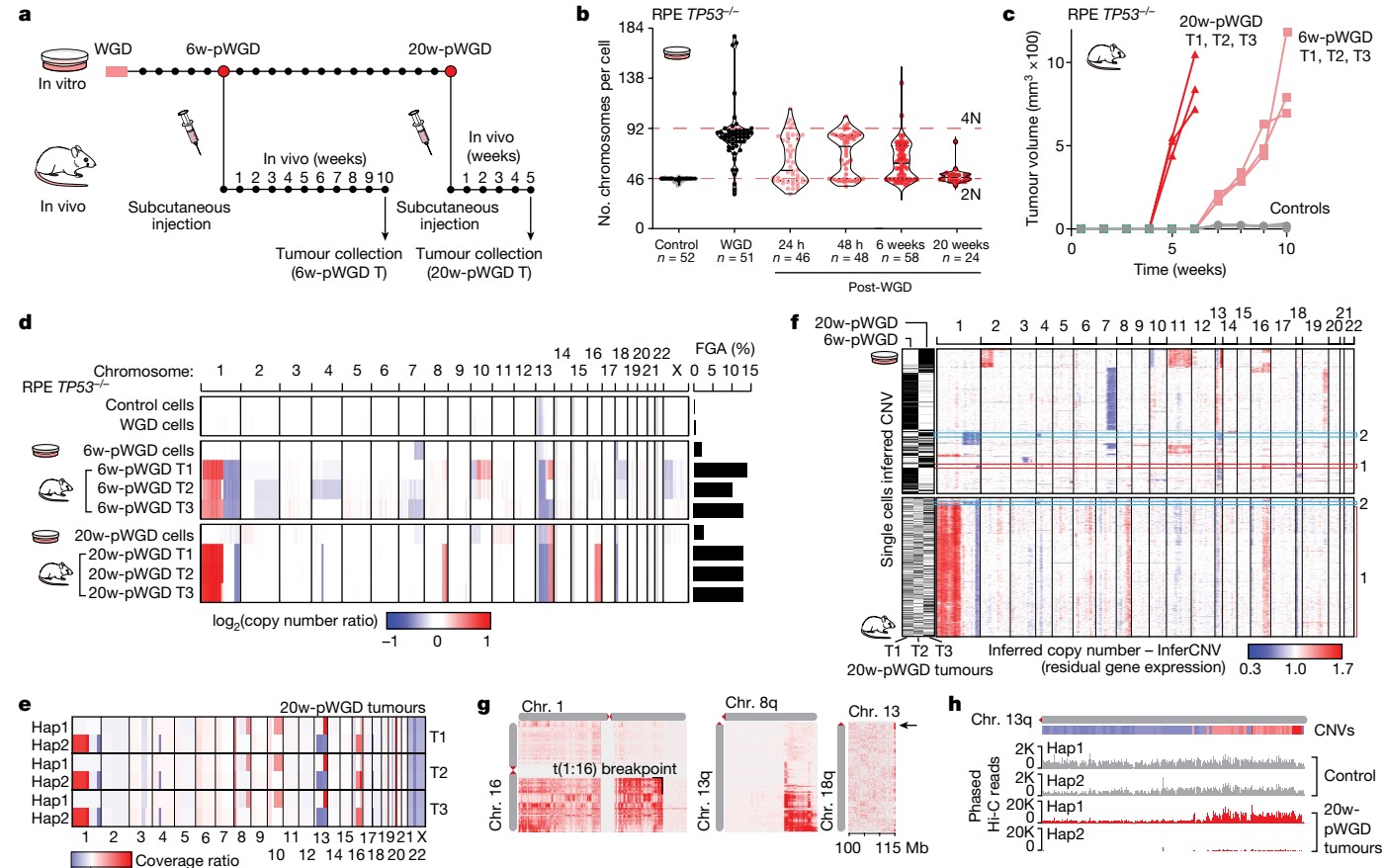

**Fig. 4 | Chromosomal alterations acquired in tumours after WGD.**
**a**, Timeline of in vitro and in vivo experiments for long-term post-WGD cells.
**b**, Chromosome per cell counts in RPE *TP53*−/− cells of control, WGD, 6-weeks post-WGD (6w-pWGD) and 20-weeks post-WGD (20w-pWGD) populations. The number of cells considered for each condition is indicated. **c**, Tumour volumes (mm³ × 100) from the time of subcutaneous injection in NOD SCID gamma (NSG) mice of RPE *TP53*−/− control (*n* = 3), 6-weeks post-WGD (*n* = 3) and 20-weeks post-WGD (*n* = 3) cells. **d**, Copy number alterations determined by WGS data in RPE *TP53*−/− control, WGD and post-WGD samples. Bar plots show the FGA for each sample **e**, Haplotype resolved copy number profile of each of the three 20-weeks post-WGD tumours. **f**, Copy number profile for each single cell

inferred from scRNA-seq data in RPE *TP53*−/− 6-weeks post-WGD, 20-weeks post-WGD and 20-weeks post-WGD tumours. The sample of origin of each cell is indicated on the left. Clonal populations detected in tumours and in vitro samples are highlighted (clones 1 and 2). **g**, Interchromosomal Hi-C contact maps exhibiting contact patterns consistent with chromosomal translocations in the RPE *TP53*−/− 20-weeks post-WGD derived tumour 2 (T2). Three chromosomal translocations are highlighted: t(1:16), t(8:13) and t(13:18). **h**, Distribution of phased Hi-C reads between haplotype 1 (Hap1) and haplotype 2 (Hap2) in RPE *TP53*−/− control and 20-weeks post-WGD derived tumour 2 samples for chromosome 13q. The corresponding copy number status for tumour 2 is shown on the top (red, copy number gains; blue, copy number losses).

20-weeks post-WGD exhibited evidence of acquired CNVs (FGA = 2% and 2.5%, respectively), although a higher number of CNVs became evident only in the in vivo tumour samples generated from either 6-week or 20-week post-WGD cells (Fig. 4d; mean FGA = 12% and 13%, respectively) (Supplementary Table 4). Shared CNV breakpoints and altered haplotypes indicated that tumours derived from 20-week post-WGD RPE *TP53*−/− cells originated from the selection and expansion of the same clone in vivo (Fig. 4d,e and Extended Data Fig. 8c). CNV acquisition was observed in nine additional tumours originated from three independent WGD experiments (Extended Data Fig. 8d). Notably, tumours derived from independent experiments sometimes acquired similar CNVs, which indicated the occurrence of convergent evolution, as recently observed in animal models after a transient induction of CIN[38].

The relatively low number of CNVs detected in the in vitro samples could be explained by subclonal heterogeneity. To test this hypothesis, we analysed all samples by single-cell RNA-sequencing (scRNA-seq) and inferred the copy number status from the read sequencing depth using the algorithm InferCNV[39]. InferCNV analysis revealed that 6-week and 20-week post-WGD in vitro samples exhibited highly

heterogenous copy number changes and clustered in distinct sub-clones, which were present in different proportions in the two samples (Fig. 4f and Extended Data Fig. 8e). By contrast, tumour samples derived from 20-week post-WGD cells were largely composed of a single clone (Fig. 4f), which exhibited CNVs consistent with those detected by WGS analyses and could already be detected in vitro, along with a less prevalent one (Fig. 4f, clones 1 and 2 on the right). Beyond CNVs, analysis of WGS and Hi-C data from these tumour samples revealed a new chromosomal translocation between chromosomes 1 and 16 (Fig. 4g), which were among the chromosome pairs that had the most increased contact frequency in LCS-WGD cells (Fig. 3c). Moreover, two translocations involving chromosome 13, one with chromosome 8 and one with chromosome 18, the latter involving the telomeric region of chromosome 13, were also observed (Fig. 4g). Notably, we could not find evidence of these translocations in control cells or WGD cells, which indicated that these events were acquired after WGD. Loss of the telomeric end in chromosome 13 was accompanied by complex chromosomal rearrangements on the second part of the q arm, and involved alternating high copy number gains (up to five copy gains) and copy number losses (Extended Data Fig. 8f). This rearrangement

pattern is characteristic of multiple breakage–fusion–bridge cycles[40,41]. Moreover, all these chromosomal rearrangements occurred in only one of the two haplotypes (Hap1), whereas the other was lost (Hap2) (Fig. 4h).

Copy number losses or gains determined by WGS analyses were associated with reduced and increased gene expression, respectively, as estimated by scRNA-seq (Extended Data Fig. 8g and Supplementary Table 5). These losses and gains accounted for around 20% of differentially expressed genes (adjusted $P < 0.001$, absolute $\log_2(FC) > 0.3$). These changes comprised upregulation of inducers of cell proliferation and migration such as *CDC42*, *NRAS* and *JUN* (chromosome 1p)[42,43], and downregulation of *CENPF* (chromosome 1q), which is associated with mitotic errors[44]. *NRAS* copy number gain was accompanied by an increase in Q61R variant allele frequency (VAF$_{tumour1}$ = 0.9, VAF$_{tumour2}$ = 0.75, VAF$_{tumour3}$ = 0.74), which suggested that the mutated allele was in the amplified haplotype. *JUN* overexpression was concomitant with an upregulation of components of the AP-1 transcription factor complex (*JUND*, *JUNB*, *FOS* and *FOSB*) and its downstream targets (Extended Data Fig. 8h). In summary, tumours originated from RPE *TP53*[−/−] cells that underwent WGD exhibited hallmarks of WGD-driven human tumours, such as increased CIN and complex rearrangements potentially associated with oncogene activation.

## Chromatin evolution of WGD cells

Next we investigated the long-term effects of WGD on chromatin 3D organization and its functional consequences. Hi-C analyses showed that tumours generated from 20-week post-WGD cells partially retained LCS features (Extended Data Fig. 9 and Supplementary Fig. 3), although these could be confounded by the high number of aneuploidies and changes in chromatin organization. Indeed, compared with RPE *TP53*[−/−] control cells, tumour samples exhibited greater differences than WGD cells in both compartment domain ranks and subcompartment assignments inferred using Calder[29] (Extended Data Fig. 10a,b). By developing a new algorithmic approach, we searched for regions that significantly changed subcompartment (Extended Data Fig. 10c), termed compartment repositioning events (CoREs). In total, we found 487 (tumour 1), 481 (tumour 2) and 478 (tumour 3) significant CoREs, which indicated changes towards either a more active (activating CoRE) or a more inactive (inactivating CoRE) subcompartment (Fig. 5a, Extended Data Fig. 10d,e and Supplementary Table 6). Genome-wide subcompartment changes and CoREs correlated with changes in histone mark intensities (Fig. 5b and Extended Data Fig. 10f,g), particularly H3K9me3 and H3K27ac, which suggested that they could underlie changes in regulatory interactions[22]. CoREs covered 17–18% of the genome and were found in similar proportions in chromosomes affected or unaffected by CNVs (Fig. 5c and Extended Data Fig. 10h). CoREs detected using our algorithm were largely recapitulated using an independent strategy (Extended Data Fig. 10i–l). Differentially expressed genes between tumours and RPE *TP53*[−/−] control cells were observed in similar numbers within a CNV or within a CoRE (Fig. 5d). CoREs were more likely to include or be near (<1 Mb) a differentially expressed gene than randomly selected genomic regions of the same size (Fig. 5e). Moreover, upregulated and downregulated genes were enriched in CoREs that changed towards a more active or inactive compartment, respectively (Fig. 5f). For example, we found activating CoREs in correspondence with upregulated oncogenes such as *JUN*, which was also amplified, and β-catenin (encoded by *CTNNB1*)[45], which was among the most significant CoREs in all three tumour samples (Fig. 5a,g). By contrast, inactivating CoREs comprised downregulated tumour suppressors and DNA repair genes such as *BRCA1* and *XRCC5* (refs. [46,47]), and the kinesin family member *KIF11*, the loss of which is associated with CIN[48] (Fig. 5a,g). The CoRE associated with *CTNNB1* was upstream of the gene and corresponded to a change from the most inactive subcompartment (B.2.2) in RPE *TP53*[−/−] diploid cells to the most active subcompartment (A.1.1)

in all three tumour samples (Fig. 5h, top, and Extended Data Fig. 11a). Within this CoRE in the tumour samples, we detected the formation of multiple H3K27ac peaks and a reduction in H3K9me3, but minor or no changes in CTCF and other histone marks (Fig. 5h and Extended Data Fig. 11a). Accumulation of H3K27ac indicated the formation of a new large enhancer, and it was associated with increased contact frequencies and significant interactions between the *CTNNB1* promoter and the enhancer region (Fig. 5i). A similar formation of H3K27ac peaks and enhancer–promoter interactions were found in a CoRE downstream of *JUN* (Extended Data Fig. 11b,c), which indicated a synergistic activation of the oncogene mediated by whole-arm chromosome gain (chromosome 1p; Fig. 4d), subcompartment repositioning, and histone acetylation changes.

Next we examined subcompartment repositioning involving the tumour suppressors *XRCC5* (Fig. 5j and Extended Data Fig. 11d) and *KIF11* (Extended Data Fig. 11e). As in the previous cases, the CoREs did not include the gene sequence but were either upstream or downstream of it. In both cases, CoREs changed from A to B subcompartments in tumours, and this repositioning was concomitant with increased H3K27me3 levels (Fig. 5j and Extended Data Fig. 11e) and loss of chromatin interactions with *XRCC5* and *KIF11* promoters (Fig. 5k and Extended Data Fig. 11f).

We noted that subcompartment repositioning events involving *CTNNB1* and *XRCC5* could be traced back to more moderate but concordant subcompartment changes already occurring in WGD cells (Fig. 5h,j). Notably, subcompartment changes detectable in WGD cells were concordant for 78–82% of the CoREs (termed consistent CoREs), frequently following a monotonic trajectory towards a more active or inactive compartment (Fig. 5l). These results were confirmed using an independent approach to select CoREs (Extended Data Fig. 11g,h). Overall, LCS initiates subcompartment changes that can result in CoREs, which leads to the deregulation of oncogenes and tumour suppressors independently of genetic alterations.

## Tracing subcompartment changes in CP-A *TP53*[−/−] cells

To confirm our results in an independent model and experiments, we followed chromatin evolution in a subset of CP-A *TP53*[−/−] cells that spontaneously acquired high ploidy (Extended Data Fig. 12a), which suggested that they underwent WGD, and in CP-A *TP53*[−/−] clones in which WGD was induced (Fig. 6a). In the extremely small high ploidy cell population, we detected new translocations and compartment repositioning events (Extended Data Fig. 12b–e). However, in this model, cells probably underwent WGD at different time points and it was not possible to determine the timing of these events. Conversely, CP-A *TP53*[−/−] cells in which WGD was synchronously induced exhibited only minor compartment changes (Extended Data Fig. 12d,f) and gradual aneuploidization at 6-weeks and 20-weeks post-WGD (Supplementary Fig. 4). As these cells did not engraft in immunocompromised animals, we used the soft-agar assay to determine malignant transformation by assessing colony formation. Both clones 3 and 19 were able to form colonies post-WGD, and the colony size increased over time (Fig. 6b). By contrast, no colonies (clone 19) or only a limited number of small colonies (clone 3) were detectable in cells that did not undergo WGD (Fig. 6b).

Next we performed Hi-C on 4 large colonies (2 from clone 3, 6 weeks post-WGD, and 2 from clone 19, 20 weeks post-WGD) and inferred CNVs and chromatin conformation changes. All colonies exhibited CNVs (Extended Data Fig. 12g) and CoREs (Fig. 6c), which were more similar among colonies derived from the same clone. Notably, 70–90% of the CoREs could be traced back to moderate but consistent compartment changes occurring in CP-A *TP53*[−/−] WGD cells (consistent CoREs) (Fig. 6d and Extended Data Fig. 12h). The overlap between CoREs found in two colonies derived from the same clone was higher when only consistent CoREs were considered (Fig. 6e). This result indicates that these

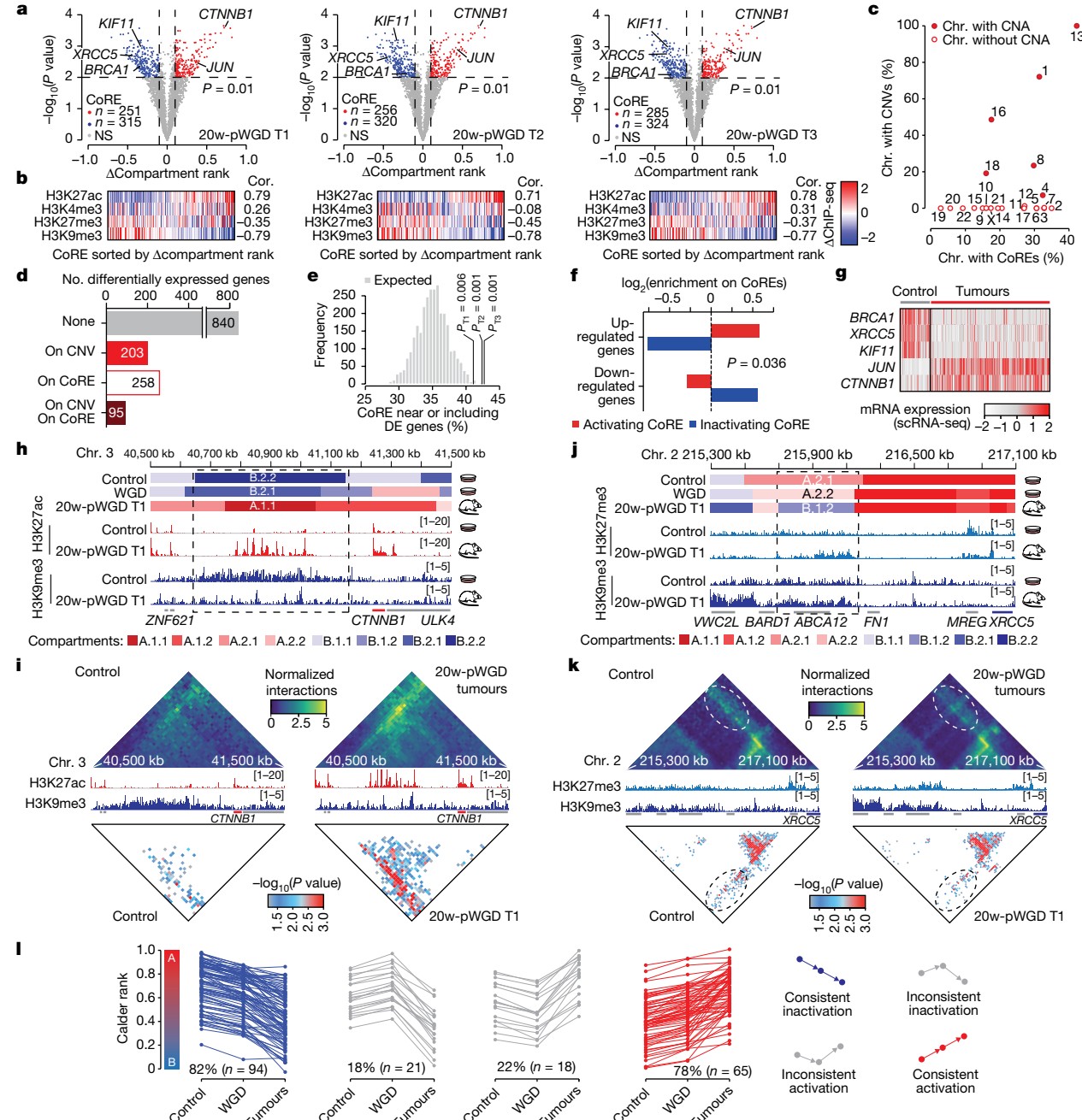

**Fig. 5 | Comparison of subcompartment repositioning in tumours after WGD. a**, Volcano plots of segmented genomic regions between each tumour and control cells. Selected CoREs are labelled on the basis of genes overlapping or in the proximity (±1 Mb) of the region. *P* values calculated using DiffComp. NS, not significant. **b**, Differential ChIP–seq signal in CoRE regions between control samples and each 20-weeks post-WGD tumour. **c**, Correlation between percentage of chromosomes affected by CoREs and CNVs for each chromosome in RPE *TP53*⁻/⁻ 20-weeks post-WGD tumours. **d**, The number of differentially expressed genes in regions unaffected (None) or affected by CNVs, CoREs or both. **e**, Expected and observed percentage of CoREs near to (±1 Mb) or overlapping with differentially expressed genes in the RPE *TP53*⁻/⁻ 20-weeks post-WGD tumours and control samples. **f**, Enrichment of differentially expressed genes in 20-weeks post-WGD tumours versus control in activating or inactivating CoREs. **g**, Normalized expression levels in single cells of selected

differentially expressed genes between RPE *TP53*⁻/⁻ control samples and 20-weeks post-WGD tumours. **h,j**, Detailed characterization of the compartment and histone modification changes in the regions of chromosome 3 (**h**) and chromosome 2 (**j**) in RPE *TP53*⁻/⁻ control and tumour 1. Top, subcompartment assignments inferred by Calder. Bottom, histone mark intensities. **i,k**, Distance-normalized interaction maps at 25 kb resolution in the regions of chromosome 3 (**i**) and chromosome 21 (**j**) in RPE *TP53*⁻/⁻ control and tumour samples (top). Histone mark intensities for the corresponding sample (middle), significant interactions RPE *TP53*⁻/⁻ control and tumour 1 samples (bottom). *P* values calculated using HiC-DC **l**, Compartment rank in control, WGD and tumours for each activating and inactivating CoRE region. Lines connect compartment ranks belonging to the same CoRE region. For **e** and **f**, *P* values were derived by data permutation (*n* = 1,000).

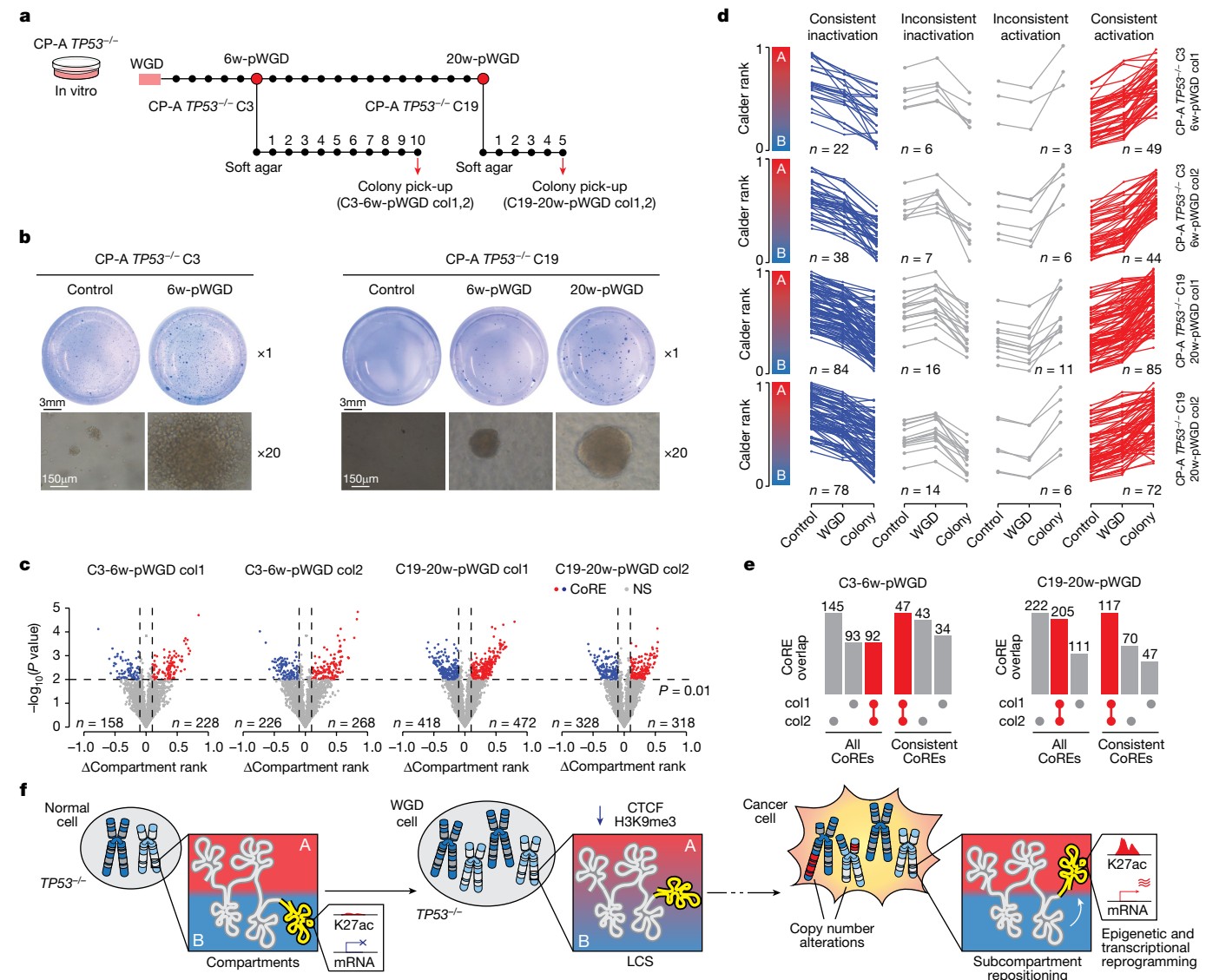

**Fig. 6 | Copy number alterations and subcompartment repositioning in CP-A *TP53*⁻/⁻ soft-agar colonies. a**, Timeline of in vitro and soft-agar colony-formation assay for CP-A *TP53*⁻/⁻ post-WGD cells. **b**, Images of crystal violet colony staining and representative images of individual colonies for CP-A *TP53*⁻/⁻ clone 3 and clone 19 (*n* = 3 independent wells). **c**, Volcano plots of segmented genomic regions, between each colony (col1 and col2) and control cells. *P* values calculated using DiffComp. **d**, Compartment rank in control (left), WGD (centre), and colonies (right) for each activating and inactivating CoRE. Lines connect compartment ranks belonging to the same CoRE. **e**, Overlap between CoREs identified in colonies derived from the same CP-A *TP53*⁻/⁻ clone when considering all or only consistent CoREs. Grey bars represent CoREs specific to one of the two colonies, whereas red bars denote common CoREs. **f**, Schematic representation of loss of chromatin segregation and subcompartment repositioning induced by WGD.

were early events that emerged and were shared by most of the cells before they were transferred in soft agar. In summary, our results show that WGD induces both CIN and LCS that lead to the emergence of chromosomal alterations and subcompartment repositioning, which ultimately favour the selection of oncogenic epigenetic and transcriptional changes (Fig. 6f).

## Discussion

Here we showed that WGD predisposes to the acquisition of a malignant phenotype, not only because of the emergence of CIN but also because of the reduced segregation of chromatin structural elements such as TADs and compartments. Increased contacts between usually well-segregated subcompartments culminate in subcompartment repositioning and epigenetic changes that support the activation of oncogenic transcriptional programmes.

However, to fully characterize the dynamic acquisition and selection of tumorigenic alterations, high-throughput and longitudinal single-cell molecular profiles are required. For example, it is tempting to speculate that increased contact frequency between chromosomes 1 and 16 in WGD cells (Fig. 3c) favoured the emergence of the translocation later observed in tumours (Fig. 4g). More generally, it will be interesting to explore whether, similar to chromosomal alterations, heterogeneous chromatin 3D organizations exist at early time points after WGD and lead to the selection of tumour-promoting chromosome interactions and compartment changes. To test these hypotheses, highly multiplexed scHi-C experiments are required, possibly paired with barcoding technologies[49,50] and computational approaches to infer and trace chromatin structural elements across multiple time points. Expanding the scope of scHi-C data and analyses will be important to understand the contribution of chromatin 3D heterogeneity in malignant transformation. Notably, evidence from previous studies[7] and our

work indicates that the oncogenic transformation of tetraploid cells is linked to a protein shortage , and activation of the tetraploid checkpoint is essential in non-cancerous tetraploid cells to restore protein levels. However, these findings were the results of targeted experiments focused on specific proteins. Future studies should investigate protein changes after WGD induction in an unbiased manner to determine whether additional phenotypes in WGD cells can be explained by insufficient protein synthesis.

Overall, our study demonstrated that in parallel to CIN, WGD induces LCS, which primes genomic regions for compartment changes that are selected and/or stabilized in tumour cells and are accompanied by epigenetic and transcriptional changes. These results provide a new lens to investigate the role of WGD and chromatin evolution in oncogenesis and tumour progression.

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

## Methods

### Cell culture

hTERT-RPE-1 WT and hTERT RPE-1 *TP53*[-/-] (46, XX)[27] cells were a gift from J. Korbel. The cells were grown in DMEM/F-12, GlutaMAX (10565018) supplemented with 10% FBS (Thermo Fisher Scientific, 10270106) and 1% antibiotic–antimycotic (Thermo Fisher Scientific, 15240062). CP-A (KR-42421) (47, XY) cells were purchased from the American Type Culture Collection (CRL-4027). CP-A *TP53*[-/-] cells were generated in this study using a CRISPR–Cas9 approach. The cells were grown in MCDB-153 medium (Sigma-Aldrich, M7403) supplemented with 20 mg l[-1] adenine (Sigma-Aldrich, A2786), 400 µg l[-1] hydrocortisone (Sigma-Aldrich, H0135), 50 mg l[-1] bovine pituitary extract (Thermo Fisher Scientific, 13028014), 1× insulin-transferrin-sodium selenite media supplement (Sigma-Aldrich, I1884), 8.4 µg l[-1] cholera toxin (Sigma-Aldrich, C8052), 4 mM glutamine (Sigma-Aldrich, G7513), 5% FBS and 1% antibiotic–antimycotic. K562 (67, XX) cells were purchased from DSMZ (ACC 10) and cultured in RPMI medium (Thermo Fisher Scientific, 11875093) supplemented with 10% FBS and 1% penicillin–streptomycin (Thermo Fisher Scientific, 15140122). All cell lines were grown in a sterile, humidified incubator at 37 °C with 5% $CO_2$ and passaged every 3–5 days, depending on the cell line, to maintain appropriate cell densities.

### Mice

All animals used in the study were NOD SCID gamma (NSG) female mice maintained at the EPFL animal facilities. Mice were kept in a 12 h-light 12 h-dark cycle, at 18–23 °C with 40–60% humidity, as recommended and in accordance with the regulations of the Animal Welfare Act (SR 455) and Animal Welfare Ordinance (SR 455.1). Mice were subcutaneously injected with 5 million cells in a 2:1 ratio of cell mixture to Matrigel basement membrane matrix (Corning, 354234), and tumour growth was monitored. Animal experiments were performed in accordance with the Swiss Federal Veterinary Office guidelines and as authorized by the Cantonal Veterinary Office (animal licence VD2932.1). Animals were sacrificed if the tumour volume was ≥1 cm³.

### Tissue dissociation

Subcutaneous tumours from mice were dissociated using a human tumour dissociation kit (Miltenyi Biotec, 130-095-929) with an enzyme cocktail and a gentleMACS dissociator with heaters (Miltenyi Biotec). The cell suspension was then strained through a 40 µm cell strainer (Corning, 352340). Samples were treated with 1× Red Blood Cell Lysis solution (Miltenyi Biotec, 130-094-183) for 10 min at 4 °C, and then spun down at 300*g* for 5 min and resuspended in 0.5% BSA in PBS. Last, mouse cells were removed from the sample using a Mouse Cell Depletion kit (Miltenyi Biotec, 130-104-694) following the manufacturer's protocol.

### CRISPR cloning

The sgRNA sequences targeting *TP53* (ref. [27]) were cloned into the pSpCas9(BB)-2A-GFP vector (PX458), which was a gift from F. Zhang (Addgene plasmid 48138; http://n2t.net/addgene:48138; RRID:Addgene_48138). In brief, 10 µM final concentration of each forward and reverse oligonucleotide were annealed in 1× T4 ligation buffer (New England Biolabs, B0202S) at 37 °C for 30 min, heated up at 95 °C for 5 min and ramped down by 0.1 °C s[-1] to room temperature. In parallel, 10 µg of PX458 vector was digested with 10 U of BsmBI (New England Biolabs, R0580L) in 1× NEBuffer 3.1 at 55 °C for 1 h. Digested plasmid was run on a 1% agarose gel, extracted and purified using NucleoSpin Gel and PCR Clean-up (Macherey-Nagel, 740609) following the manufacturer's instructions. Annealed CRISPRs and digested plasmid were ligated using 5 U T4 DNA ligase (Thermo Fisher Scientific, EL0011) in 1× T4 DNA ligase buffer for 10 min at room temperature. The plasmid was then added to DH5α chemically competent bacteria and kept for 30 min on ice, followed by heat shock at 42 °C for 45 s. The bacteria were cooled down on ice and recovered in SOC medium for 1 h at 37 °C. Transformed bacteria were grown on ampicillin-containing growth medium at 37 °C overnight. Bacterial colonies were picked and expanded in LB broth supplemented with ampicillin for 12 h at 37 °C. Plasmid DNA was extracted using a Plasmid Plus Midi kit (Qiagen, 12945) according to the manufacturer's protocol. The plasmids were verified by Sanger sequencing (Microsynth) with hU6 primers.

### CP-A *TP53*[-/-] cell line generation

CP-A WT cells were grown to 60–70% confluency in 10 cm plates. Next, 5 µg of PX458 plasmid containing *TP53*-targeting sgRNAs were diluted into 200 µl Opti-MEM I reduced serum medium (Thermo Fisher Scientific, 31985062). Then 15 µl FuGENE HD transfection reagent (for a 3:1 transfection reagent:DNA ratio) (Promega, E2312) was added to the DNA and incubated at room temperature for 15 min. The mixture was then added to a plate drop-by-drop and mixed by shaking. Cells were incubated for 48 h at 37 °C in a humidified incubator. Transfected cells, expressing GFP, were single-cell sorted on a BD FACSAria Fusion instrument (BD Biosciences). Clones were allowed to expand, then individually tested by immunoblotting for TP53 protein levels following 24 h of treatment with 3 µM doxorubicin (Cayman Chemical, 15007).

### WGD induction

Cells were seeded to 60–70% density. For mitotic slippage induction, 0.1 µg ml[-1] nocodazole (Sigma-Aldrich, M1404) was added to the growth medium and CP-A and K562 cells were incubated for 72 and 48 h, respectively. For cells with an elongated G1 phase after tetraploidization, WGD in CP-A *TP53*[-/-] cells was induced with 0.1 µg ml[-1] nocodazole for 72 h and treated with 0.5 µM of the CDK4/6i palbociclib (Sigma-Aldrich, PZ0383) for the last 16 h of the WGD induction protocol. For cytokinesis failure inductions, RPE cells were incubated for 24 h with 0.1 µg ml[-1] nocodazole-containing medium. Following nocodazole treatment, the cells were exposed for an additional 24 h to 4 µM dihydrocytochalasin B (Cayman Chemical, 20845). The treatment was removed and cells were allowed to recover for 48 h to allow transition from a binucleated to a mononucleated state. Alternatively, WGD was induced through cytokinesis failure in RPE *TP53*[-/-] cells by incubation for 24 h with 9 µM of the CDK1 inhibitor RO-3306 (Sigma-Aldrich, SML0569) for G2 synchronization. The compound was washed off and cells were then treated with 4 µM dihydrocytochalasin B for 24 h. Cells were allowed to recover for 48 h, and tetraploid cells were sorted on the basis of cell cycle staining with 1 µg ml[-1] Hoechst 33342 (Thermo Fisher Scientific, H1399).

### Isolation of spontaneous high-ploidy cells

CP-A *TP53*[-/-] cells were stained with 1 µg ml[-1] Hoechst 33342 for cell cycle profiling. Dividing cells with high ploidy (high Hoechst 33342 signal, >4N peak) were bulk sorted. Cells were allowed to recover overnight and then fixed for downstream analyses.

### CIN induction

CIN was induced in RPE *TP53*[-/-] cells using a modified protocol described previously[51]. In brief, the cells were synchronized at the G1/S border with 5 mM thymidine (Sigma-Aldrich, T9250) for 24 h. Six hours after thymidine block release, the cells were treated with 500 nM of the MPS1 inhibitor reversine[52] (Sigma-Aldrich, R3904) for 12 h. Before processing for downstream analyses, cells were allowed to recover for 6 h.

### Cell cycle staining

Cells were collected and washed with PBS (Thermo Fisher Scientific, 10010023). Permeabilization was performed in 0.01% Triton X-100 (AppliChem, A1388) in PBS for 30–60 min at 4 °C. Following PBS washes, the cells were fixed and stained with FxCycle PI/RNase staining solution (Thermo Fisher Scientific, F10797) overnight at 4 °C in the absence of light. Propidium iodide intensity for cell cycle detection was measured using Guava easyCyte (Luminex) and Galios (Beckman Coulter) cytometers and analysed using FlowJo (v.10.8) (BD).

## Karyotyping

Cells were treated with 20 ng ml$^{-1}$ KaryoMAX colcemid solution (Thermo Fisher Scientific, 15212012) for 2 h at 37 °C in a humidified incubator. Cells were collected in 0.8% sodium citrate solution (Sigma-Aldrich, S4641) and maintained at 37 °C for 30 min. The cell suspension was fixed with 3:1 methanol:acetic acid (Chemie Brunschwig, M/4000/17; FSHA/0406/PB08) added drop-by-drop, washed twice in the fixative solution and incubated overnight at −20 °C. Cells were dropped onto a glass slide (Thermo Fisher Scientific, J1800AMNZ). Slides were incubated for 2 min in a humidified chamber at 65 °C and air-dried at room temperature for 30 min. Slides were mounted and DAPI-stained concomitantly with ProLong Diamond antifade mountant with DAPI (Thermo Fisher Scientific,. P36962) according to the manufacturer's instructions. Metaphases were imaged at ×100 resolution on a Zeiss Axioplan upright microscope. Images were analysed using Fiji (v.2.9.0)[53].

## Immunoblotting

For non-histone proteins, cells were incubated in RIPA buffer consisting of 50 mM Tris-HCl pH 8.0, 1 mM EDTA, 1% Triton X-100, 0.5% sodium deoxycholate (Sigma-Aldrich, D6750), 0.1% SDS and 150 mM NaCl, for 30 min on ice for protein extraction. For histone extraction, cells were initially incubated with PBS lysis buffer consisting of 1% Triton X-100, 1 mM DTT (AppliChem, A2948), 1× protease inhibitor cocktail for 15 min at 4 °C and spun down at 12,000$g$. The resulting pellet was incubated overnight with 0.2 N hydrochloric acid (AppliChem, A5634). Lysates were then centrifuged at 12,000$g$ for 10 min at 4 °C. Supernatant containing the protein fraction was isolated and mixed with 6× Laemmli sample buffer (12% SDS w/v, 60 mM Tris-HCl, pH 6.8, 50% glycerol (Fisher Scientific, G/0650), 600 mM DTT and 0.06% bromophenol blue (Sigma-Aldrich, B5525)) at 96 °C for 5 min. Mid-molecular weight proteins and histones were then separated on 12% or 15% SDS–PAGE gels, respectively, whereas high-molecular weight proteins were separated on a 7.5% Mini-PROTEAN TGX precast protein gel (Bio-Rad, 4561023). All gels were transferred onto 0.2 μm nitrocellulose membranes (Bio-Rad, 1704270) using a Trans-Blot Turbo transfer system (Bio-Rad, 1704150) according to the manufacturer's specifications. The membranes were blocked in a solution containing 5% milk (AppliChem, A0830) and 0.1% Tween-20 (Fisher Bioreagents, 10113103) in PBS for 30 min at room temperature. Blots were incubated in the same milk solution at either 4 °C overnight with primary antibodies against TP53 (Santa Cruz Biotechnology, sc-126; 1:500), β-actin (Cell Signaling Technology, 4967; 1:5,000), CTCF (Active Motif, 61311; 1:1,000), RAD21 (Abcam, ab992; 1:5,000) and α-actinin (Cell Signaling Technology, 6487; 1:1,000), or for 1 h at room temperature with primary antibodies against trimethyl-histone H3 (Lys9) (Cell Signaling Technology, 13969; 1:1,000), acetyl-histone H3 (Lys27) (Cell Signaling Technology, 8173; 1:1,000), trimethyl-histone H3 (Lys27) (Cell Signaling Technology, 9733; 1:1,000), and histone H3 (Cell Signaling Technology, 4499; 1:5,000). The membranes were incubated with fluorescent labelled goat anti-mouse (LI-COR Biosciences, 926-68070; 1:10,000) or goat anti-rabbit (LI-COR Biosciences, 926-32211; 1:10,000) for 2 h at room temperature and imaged using an Odyssey CLx imaging system (LI-COR Biosciences). Alternatively, the membranes were incubated with HRP-conjugated goat anti-mouse antibody (Merck, AP308P; 1:5,000) or goat anti-rabbit antibody (Merck, AP307P) for 1 h at room temperature. Blots were incubated with Amersham ECL western blotting detection reagent (GE Healthcare, RPN2232) according to the manufacturer's instructions, and captured using a Fusion FX6 Edge imaging system (Witec). Images were analysed using Fiji (v.2.9.0)[53].

## Immunofluorescence

Cells were cultured on coverslips coated with poly-D-lysine (Sigma-Aldrich, P7280) and incubated in standard conditions. Cells on coverslips were fixed with ice-cold methanol at 4 °C for 30 min. Cells were washed with PBS multiple times and incubated with 5% BSA (Sigma-Aldrich, A7906) at room temperature for 30 min. Next coverslips were incubated with primary antibodies at the indicated concentrations against pericentrin (0.1 μg ml$^{-1}$; Abcam, ab4448) and α-tubulin (0.5 μg ml$^{-1}$; Sigma-Aldrich, T6074) diluted in 1% BSA for 1 h at room temperature in a humidified chamber. Coverslips were washed with PBS and incubated with the fluorescent secondary antibodies anti-mouse IgG-Alexa Fluor 594 (2 μg ml$^{-1}$; Thermo Fisher Scientific, A-11005) and anti-rabbit IgG-Alexa Fluor 488 (2 μg ml$^{-1}$; Thermo Fisher Scientific, A-11034) diluted in 1% BSA for 1 h at room temperature in the dark. Coverslips were washed in PBS followed by mounting and counterstaining with DAPI with ProLong Diamond antifade mountant. Cell images were captured at ×63 resolution on a Zeiss Axioplan upright microscope. Images were analysed using Fiji (v.2.9.0)[53].

## Soft-agar assay

For each condition, 100,000 cells resuspended in complete MDCB-153 medium were mixed in a 1:1 ratio with 0.7% sterile noble agar (Thermo Fisher Scientific, J10907). Cells were plated on Costar ultralow attachment plates (Corning, 3473) on top of a mixture of 1:1 MCDB-153 medium and 1.4% sterile noble agar. The mixture was allowed to solidify in a humidified atmosphere at 37 °C overnight, then fresh complete MCDB-153 medium was added on top of the layers of agar. Samples were incubated for up to 10 weeks in normal conditions, and the medium was replaced twice a week. Individual colonies were picked from the agar layer and cultured for downstream analysis. Last, the plates were stained with a solution of 0.5% crystal violet (Thermo Fisher Scientific, 405830250) in 20% ethanol for 30 min, washed with PBS and imaged.

## Hi-C library preparation and analysis

**Hi-C library preparation.** Bulk Hi-C library preparation was performed as previously described[11,22], with minor modifications. Around 1–2 million cells were collected and fixed with 2% formaldehyde (Thermo Fisher Scientific, 11483217). The reaction was quenched with 200 mM glycine (VWR, 101194M) final concentration, cells were washed with PBS (Thermo Fisher Scientific, 10010023) and lysed in a solution containing 10 mM Tris-HCl pH 8.0 (Thermo Fisher Scientific, 15568025), 10 mM NaCl (Sigma-Aldrich, S6546), 0.2% IGEPAL CA-630 (Sigma-Aldrich, I8896) and 1× proteinase inhibitor cocktail (Roche, 11697498001) at 4 °C for 30 min. Resulting nuclei were resuspended in 1× NEB3.1 buffer (New England Biolabs, B7203S). The suspension of nuclei was incubated with 0.11% SDS (Carl Roth, CN30) final concentration for 10 min at 65 °C. The reaction was quenched with 1% Triton X-100 (AppliChem, A1388), and nuclei were digested with 100 U MboI restriction enzyme (New England Biolabs, R0147) at 37 °C overnight. The restriction enzyme was inactivated according to the manufacturer's specifications, and digested nuclei were washed and resuspended in 1× NEB3.1. Digested ends were then marked with biotin through incubation in 0.03 mM biotin-14-dATP (Thermo Fisher Scientific, 19524016), 0.03 mM dCTP, 0.03 mM dGTP, 0.03 mM dTTP (Promega, U1420) and 50 U Klenow DNA polymerase I (New England Biolabs, M0210M) for 4 h at room temperature. Resulting blunt-ends were proximally ligated with 50 U T4 DNA ligase (Thermo Fisher Scientific, EL0011), 1× T4 DNA ligase buffer (Thermo Fisher Scientific, B69), 5% PEG, 1% Triton X-100 and 0.1 mg ml$^{-1}$ BSA (New England Biolabs, B9000S) for 4 h at room temperature. Crosslink reversal was performed on the proximity-ligated chromatin through incubation with 300 mM NaCl and 1% SDS overnight at 68 °C. The sample was then treated with 50 μg ml$^{-1}$ RNase A (Thermo Fisher Scientific, EN0531) for 30 min at 37 °C, followed by 400 μg ml$^{-1}$ proteinase K (Promega, V3021) at 65 °C for 1 h. DNA was purified by precipitation with 1.6 volumes of pure ethanol and 0.1 volumes of sodium acetate, pH 5.2 (Thermo Fisher Scientific, R1181) at −80 °C. DNA was eluted and then fragmented by sonication at 80 V peak incidence power, 10% duty factor, 200 cycles per burst for 60–80 s with an E220 focused-ultrasonicator (Covaris). Sheared DNA was size-selected for library preparation using AMPure

XP beads (Beckman Coulter, A63881). Next, biotin-marked fragments were isolated using Dynabeads MyOne Streptavidin C1 (Thermo Fisher Scientific, 65001), and all subsequent steps were performed on the bead-bound DNA fraction. Hi-C library preparation continued with an end polishing reaction, which involved incubation of DNA with 1× T4 ligase buffer (New England Biolabs, B0202S), 2.5 mM each dNTP, 50 U T4 polynucleotide kinase (New England Biolabs, M0201), 12 U T4 DNA polymerase (New England Biolabs, M0203) and 5 U Klenow DNA polymerase I, at room temperature for 30 min. PolyA tail was added by incubating the DNA sample in 1× NEBuffer 2 (New England Biolabs, B7002S) with 0.5 mM dATP and 25 U Klenow fragment (3′→5′ exonuclease) (New England Biolabs, M0212) at 37 °C for 30 min. DNA fragment ends were then ligated to Illumina TruSeq unique dual indexes (Integrated DNA Technologies) in 1× T4 ligation buffer with 5% PEG and 15 U T4 DNA ligase for 2 h at room temperature or overnight at 16 °C. Last, libraries were PCR amplified using Illumina forward (AATGATACGGC GACCACCGAGATCTACAC) and reverse (CAAGCAGAAGACGGCATAC GAGAT) primers and KAPA HiFi HotStart ReadyMix (Roche, KK2602) for 6–10 cycles. Resulting fragments were size-selected using AMPure XP beads. Libraries were sequenced in a PE150 configuration on HiSeq X, NovaSeq 6000 or HiSeq 2500 systems (Illumina).

**Generation of Hi-C contact maps.** For each library replicate, reads were mapped to the human hg19 reference genome using bwa mem (v.0.7.17)[54] and processed using the Juicer pipeline (v.1.6)[55]. For each sample, Hi-C maps were generated at the following resolutions: 10 kb, 20 kb, 25 kb, 50 kb, 100 kb, 250 kb, 500 kb, 1 Mb and 10 Mb. Once the concordance between replicates of the same biological condition was assessed, Hi-C maps of the same condition were merged using the mega.sh script provided in the Juicer pipeline. All Hi-C maps were normalized using the Knight–Ruiz method (KR)[56] implemented in the Juicer pipeline.

**Definition of Hi-C compartments.** Compartments were called using the Calder pipeline[29] on KR-normalized Hi-C maps at 50 kb resolution. Calder returns a segmentation of the genome in compartments where each segment is assigned both a compartment rank (a real number between 0 and 1) and a compartment label (B.2.2, B.2.1, …, A.1.2, A.1.1), which is a discretization of the rank in eight different categories. Compartment ranks and labels correlate with the chromatin state of the DNA region, with values close to 0 being more B-like compartments and values close to 1 being more A-like compartments.

**Assessment of similarity between Hi-C contact maps.** Pairwise comparisons between intrachromosomal contact maps were based on the following metrics: a correlation measure between the contacts, stratified by the distance between the interacting loci; the conservation of compartment domains and their boundaries; the correlation at the level of boundary insulation; and the correlation at the level of Calder compartment rank.

Replicates of the same biological condition (control versus control, WGD versus WGD) and samples of different conditions (control versus WGD, control vs 20-weeks post-WGD tumours) were compared, as well as samples of a different cell line (control versus GM12878 from ref. [12]). Inter-replicate comparisons and intercell line comparisons gave a reference baseline of random fluctuations and extensive chromatin changes, respectively, for each score.

**Correlation of contacts (stratum-adjusted correlation coefficient).** The stratum-adjusted correlation coefficient[57] was used as implemented in the HiCRep.py package[58]. The maximum genomic distance to test was set to 10 Mb, and Hi-C maps were binned at 100 kb and smoothed with a window of $H = 3$ bins. For each comparison, a stratum-adjusted correlation coefficient value was computed for each chromosome.

**Conservation of compartment domains.** Compartment domains were called at 50 kb resolution using the Calder pipeline on the KR-normalized Hi-C matrices.

Given two compartment domain sets identified on the same chromosome in two samples, the measure of concordance[59] was calculated, which was previously defined to compare two clustering assignments. The measure of concordance is a real number bounded between 0 and 1, with 1 representing identical chromosome segmentation and 0 maximum discordance.

**Conservation of insulating boundaries.** Hi-C insulation was computed as previously described[30]. Insulation scores for each chromosome were calculated using the FANC library[60], specifically, the InsulationScores. from_hic function on the KR-normalized intrachromosomal Hi-C matrices at 50 kb resolution using a sliding window of 1 Mb. Sliding windows with more than 20% of missing values were not considered. Scores were normalized by the geometric mean chromosome-wise and finally $log_2$-scaled. The final score is therefore centred at 0, with local minima representing putatively TAD boundaries.

Comparisons between samples were performed by computing the Spearman correlation coefficient of the insulation scores for each chromosome.

**Hi-C compartment similarity.** The compartment segmentation given by Calder was split in bins of 50 kb, assigning to each bin the compartment rank of the segment it belongs to. The similarity between two samples was then computed separately for each chromosome as the Spearman correlation of the two binned rank vectors.

**Hi-C interchromosomal similarity.** The interchromosomal interactions for each pair of Hi-C maps were compared by considering separately the interactions between each pair of different chromosomes in the two samples. The Spearman correlation coefficient of the raw interaction counts was computed between the two samples for each chromosome pair. For each Hi-C comparison, therefore, a correlation value for each pair of chromosomes was obtained.

**Analysis of Hi-C interchromosomal interactions.** To determine interaction biases between pairs of chromosomes, Hi-C interactions were aggregated between each pair of chromosomes, obtaining a 23 × 23 interaction matrix *I*. The matrix was balanced using iterative correction[61] to remove interaction biases due to the length of the chromosomes (such that the marginal sum of each chromosome is 1). This resulted in a normalized matrix $I_{ICE}$. This normalization is similar to the one presented in ref. [11], with the advantage of ensuring constant marginals. When compared, both normalizations produced comparable results.

Chromosomes were then divided into two clusters on the basis of their interaction profile in $I_{ICE}$: chromosomes from 1 to 14 and X were categorized as long, whereas chromosomes from 15 to 22 were categorized as short.

To compare control and WGD interchromosomal interaction matrices, their ratio $R = \log_2[I_{ICE}(WGD)/I_{ICE}(Control)]$ was computed. Chromosome interactions were then split into three categories on the basis of the chromosome cluster of their ends: long–long, long–short and short–short. Chromosome interaction categories were compared by computing a Mann–Whitney test $P$ value between $R$ values of each pair of categories.

**Hi-C interchromosomal map balancing at 10 Mb resolution.** To visualize interchromosomal Hi-C maps at 10 Mb resolution, Iterative Correction using the Cooler package[62] was performed. Counts were normalized such that each bin had total number of interchromosomal interactions equal to 1.

## Analysis of Hi-C intercompartmental interactions

The genomic segmentation in eight classes given by Calder (B.2.2 to A.1.1) was considered. Each 50 kb genomic bin was then associated to the compartment level it belongs. For each chromosome, its intrachromosomal contacts were extracted at 1 Mb resolution and then normalized by genomic distance[12] using the FANC package[60]. These interactions were then upscaled to 50 kb resolution by assigning to each $50 \times 50$ kb pixel the value of the $1 \times 1$-Mb superpixel it belonged to. This procedure was performed to smooth the normalized interaction values and to ensure enough coverage for each genomic distance. For each 50 kb genomic bin $b$, the sum of the normalized interactions between that bin and the bins belonging to the eight compartment level classes was computed separately, thus obtaining one value $s_b(\text{comp})$ for each compartment level comp. These values were then divided by the total sum of interactions of that bin $T_b$. To consider the bias induced by the amount of chromosome covered by each compartment level, these values were further divided by the percentage of bins belonging to each compartment level $B_{\text{comp}}$, thus obtaining $z_b(\text{comp}) = s_b(\text{comp})/T_b B_{\text{comp}}$. The obtained value was finally divided by their sum to obtain for each bin $f_b(\text{comp}) = z_b(\text{comp})/\Sigma_c z_b(c)$, which is a number between 0 and 1 for each compartment level representing the level of segregation of each compartment level for that bin. For each bin $b$, it was defined as $C\text{Score}_b$ the segregation level of the compartment level to which the bin belongs to. This definition is an adaptation of the compartment score computed in ref. [63], but applied at the bin level.

Given two conditions, for example, WGD and control, the difference for each pair of subcompartments $\text{comp}_1$, $\text{comp}_2$ was computed as follows:

$$\sigma_{\text{comp}_1}(\text{comp}_2) = \frac{\text{No. times } f_b^{\text{WGD}}(\text{comp}_2) < f_b^{\text{Control}}(\text{comp}_2)}{\text{No. times } f_b^{\text{WGD}}(\text{comp}_2) \geq f_b^{\text{Control}}(\text{comp}_2)},$$
$$\text{for } b \in \text{comp}_1$$

and

$$\sigma(\text{comp}_1, \text{comp}_2) = -\log_2\left(\frac{1}{2}(\sigma_{\text{comp}_1}(\text{comp}_2) + \sigma_{\text{comp}_2}(\text{comp}_1))\right).$$

$\sigma_{\text{comp}_1}(\text{comp}_2)$ represents the ratio between the number of bins in $\text{comp}_1$, which lose compartment segregation with $\text{comp}_2$, and the number of bins in $\text{comp}_1$, which gain compartment segregation with $\text{comp}_2$.
$\sigma(\text{comp}_1, \text{comp}_2)$ is simply the average of $\sigma_{\text{comp}_1}(\text{comp}_2)$ and $\sigma_{\text{comp}_2}(\text{comp}_1)$, which makes it a symmetric measurement of average segregation changes between compartment levels $\text{comp}_1$ and $\text{comp}_2$. The $-\log_2$ of this number was computed for representation purposes, with positive and negative $\log_2$ ratios indicating gain and loss of contacts, respectively, between the two compartment levels.

To specifically assess the extent of loss of segregation for a specific compartment level comp, a similar strategy was adopted:

$$\sigma_{\text{comp}} = \frac{\text{No. times } C\text{Score}_b^{\text{WGD}} < C\text{Score}_b^{\text{Control}}}{\text{No. times } C\text{Score}_b^{\text{WGD}} \geq C\text{Score}_b^{\text{Control}}}, \text{ for } b \in \text{comp}$$

was computed, which represents the ratio between the number of bins in comp losing segregation and the number of bins in comp gaining segregation. Values higher than 1 indicate loss of segregation, whereas values below 1 indicate gain of segregation.

**Boundary insulation analysis.** TAD boundaries in control and WGD samples were determined from insulation scores using the fanc.Boundaries.from_insulation_score function from the FANC package[60], looking at local minima of the score in the 400 kb region around the bin. Each boundary was assigned the insulation score corresponding to

its position. Lower values of the score signify higher insulation capability of the boundary. The boundaries shared between the two samples (±50 kb) were then extracted. The top 300 insulating boundaries were selected as follows: control and WGD boundaries were separately ranked on the basis of their insulation scores. For each condition, the top 300 ranked boundaries were selected and their maximum insulation score (corresponding to the weaker boundary in the set) was determined, which was called $I_{\text{top300}}^{\text{Control}}$ and $I_{\text{top300}}^{\text{WGD}}$, respectively. An insulation threshold $I_{\text{top300}} = \max(I_{\text{top300}}^{\text{Control}}, I_{\text{top300}}^{\text{WGD}})$ was defined. Finally, shared boundaries between control and WGD having insulation scores smaller than $I_{\text{top300}}$ were selected. It should be noted that this approach does not ensure that the final number of selected boundaries is exactly 300.

## Independence of LCS measurement from Hi-C resolution and coverage per haploid copy

The aggregated map of RPE $TP53^{-/-}$ WGD cells (218 million reads) was compared with one of the control replicates maps (108 million reads). Conversely, one replicate of the control (108 million reads) was compared with the aggregated map of the same control (221 million reads).

**Detection of regions of significant CoREs.** To determine significant CoREs, we developed an algorithm to identify contiguous genomic regions with consistently different compartment ranks computed using Calder. We refer to this method as DiffComp. A segmentation algorithm was designed as follows. Given two Hi-C experiments $X$ and $Y$, the genomic segmentations of both in compartment domains was determined using Calder on the 50 kb resolution KR-normalized Hi-C matrices. Both segmentations were then binned in 50 kb bins, assigning to each bin its relative compartment rank. Thus, for each chromosome, compartmentalization in the two samples is represented as two numerical vectors $\mathbf{C}_X$, $\mathbf{C}_Y$.

The pairwise rank difference for each genomic bin were computed as $\Delta\mathbf{R}_{XY} = \mathbf{C}_X - \mathbf{C}_Y$. This vector represents the differential rank between the two experiments, with positive values indicating a shift towards active compartments and negative values indicating a shift towards inactive compartments.

The genome was segmented based on $\Delta\mathbf{R}_{XY}$ using a recursive strategy. Given $\sigma^*$, which represents the maximum allowed standard deviation in the signal that a segment can have before being split into subsegments, the procedure involves the following process.

Each chromosome is initially considered a single whole segment and then

(1) The standard deviation of the segment $\sigma(s)$ and its average value mean$(s)$ were calculated.
(2) If $\sigma(s) < \sigma^*$, then the procedure stops and the segment is assigned mean$(s)$ as value, which represents its subcompartment repositioning score.
(3) Otherwise, the segment is split into subsegments depending on whether they are above or below the mean$(s)$ value.
(4) For each of the subsegments, the procedure is repeated from point (1).

The expected distribution of compartment changes can be computed using technical or biological replicates of the same experiment. An expected differential vector $\Delta\mathbf{R}_E = \mathbf{C}_{R1} - \mathbf{C}_{R2}$ was computed using two replicates of RPE $TP53^{-/-}$ control.

In this analysis, $\sigma^* = 0.1$ was fixed, which is 1.3-times the standard deviation of $\Delta\mathbf{R}_E$. For each detected compartment repositioning segment $s$, an empirical $P$ value as $P\text{value}(s) = P(\max\{|\Delta\mathbf{R}_{XY}(s)|\} < |E|)$ was computed, where $E \in \Delta\mathbf{R}_E$. This $P$ value depends both on the average value of the segment and on its length, for which longer segments have higher statistical power.

The output of this method is a list of CoRE regions together with their average compartment repositioning score (which can vary from −1 and 1) and their computed empirical $P$ value.

For each comparison studied, CoRE regions having an absolute average value above 0.1, an empirical $P$ value below 0.01 and a segment length above 300 kb were considered.

## CoRE overlap in CP-A *TP53*[-/-] colonies

To assess the amount of overlap between two sets of CoREs $C_1$, $C_2$ belonging to different sample comparisons, the two sets were divided in activations and inactivations on the basis of the sign of the compartment repositioning score ($C_1^A$, $C_1^I$, $C_2^A$, $C_2^I$). The CoREs of the same type coming from both sample comparisons ($C_{12}^A$, $C_{12}^I$) were merged together by stacking overlapping regions together (using the bedtools merge command from bedtools (v.2.30.0)[64,65], finally creating a consensus set of CoREs concatenating the two sets ($C_{12} = [C_{12}^A, C_{12}^I]$). Each consensus CoRE was checked for overlapping with a CoRE of the same type in $C_1$ and $C_2$. Consistent CoREs were considered overlapping when two CoREs of the same type were overlapping and at least one of the two was a consistent CoRE.

## Tracing compartment repositioning from WGD to tumour time points.

For each of the tumours, the CoRE regions that passed the previously defined statistical filters were considered. For each of these regions $s$, the corresponding Calder segmentation in the WGD and control time points were extracted. The average compartment rank of the CoRE region in the two previous time points were then computed, which were defined as $r_{WGD}(s)$ and $r_{Control}(s)$, respectively. The compartment rank of the CoRE region in the tumour were defined as $r_{Tumour}(s) = r_{Control}(s) + mean(s)$, where $mean(s)$ is given by the CoRE detection algorithm. A parameter $\varepsilon$ was then defined, which is the minimum absolute rank difference between WGD and control, namely $|r_{WGD}(s) - r_{Control}(s)|$, to classify the CoRE region as activating or inactivating in WGD with respect to control.

Given $\varepsilon$, CoRE regions in tumours can be discriminated on the basis of the type of change in tumours (activating or inactivating) as well as the type of change at WGD (unchanged, activating or inactivating). The number of CoRE regions belonging to each of the six combinations was counted.

A CoRE region was defined as 'consistent' when it belongs to the activation–activation or the inactivation–inactivation class. The percentage of consistent CoRE regions were counted with different choices of parameter $\varepsilon$. Observing the steepness of the curves in the three tumour samples, a shared parameter to $\varepsilon = 0.05$ was fixed.

## Comparing different segmentation algorithms for CoRE detection.

The segmentation strategy in DiffComp was compared to the circular binary segmentation (CBS) algorithm, which was previously developed for the segmentation of copy number changes[66]. CBS was applied to the Calder differential rank vector $\Delta\mathbf{R}_{XY} = \mathbf{C}_X - \mathbf{C}_Y$, which was computed as explained above. Segments detected using CBS were annotated with their compartment repositioning score and $P$ values as described above. CoREs were then filtered on the basis of the repositioning score, $P$ value and size as defined above. The CoREs detected using DiffComp and the ones detected with CBS were compared using as the benchmark the RPE *TP53*[-/-] 20 week post-WGD tumour 1 versus RPE *TP53*[-/-] control comparison. We then compared the breakpoint positions between the two segmentation strategies, the corresponding sets of significant CoREs and the traceability of these events to subcompartment changes occurring in WGD cells.

## Hi-C phasing in RPE *TP53*[-/-] 20-week post-WGD tumours.

Integrated phasing was performed using Hi-C reads from the pooled RPE Hi-C replicates (control). Single-nucleotide variants (SNVs) were first identified from the Hi-C reads using Freebayes[67] (version v1.3.2-46-g2c1e395-dirty). SNVs were phased into two haplotypes, namely Hap1 and Hap2, using a previously described integrated phasing strategy[68]. In brief, population-based phasing was first conducted using SHAPEIT2 (ref. [69]; version v2.904.3.10.0-693.11.6.el7.x86_64) with hg19 1000 Genomes project phase 3 as a reference panel. Pseudo-reads generated from the population haplotype likelihood were then combined with the Hi-C reads as input to HapCUT2 (ref. [70]) for the second round of phasing. This approach returned several phasing blocks for each chromosome. Phasing information was retained only from the most variants phased block, which harbours the majority of input SNVs (>90%). Only Hi-C interactions for which anchors mapped strictly to one of the two haplotypes were retained for analysis, thus obtaining three sets of interaction types: Hap1–Hap1, Hap1–Hap2 and Hap2–Hap2.

## Analysis of contacts between homologous chromosomes after WGD

After WGD, the rates both *in cis* and *in trans* contacts are expected to increase. In detail, putative *in cis* contacts should increase by a factor of 3 following WGD, and *in trans* contacts should increase by a factor of 4. Hence, it is expected that the ratio of *in trans* (T) versus *in cis* (C) contacts increases after WGD as described below:

$$r_{Control} = \frac{T}{2C}; \quad r_{WGD} = \frac{4T}{6C}; \quad \frac{r_{WGD}}{r_{Control}} = \frac{4}{3} = 1.33$$

To verify this prediction, *in cis* interactions were defined as all the Hi-C-phased interactions of the type Hap1–Hap1 and Hap2–Hap2, and *in trans* interactions all the Hi-C-phased interactions of type Hap1–Hap2. The following was computed:

$$r_{Control} = \frac{No. (Hap_1 - Hap_2)^{Control}}{No. (Hap_1 - Hap_1)^{Control} + No. (Hap_2 - Hap_2)^{Control}}$$

$$r_{WGD} = \frac{No. (Hap_1 - Hap_2)^{WGD}}{No. (Hap_1 - Hap_1)^{WGD} + No. (Hap_2 - Hap_2)^{WGD}}$$

and $\frac{r_{WGD}}{r_{Control}}$ separately for each chromosome, and for the genome-wide average ratio. Finally, $\frac{r_{WGD}}{r_{Control}} = 1.25$ was obtained, close to the predicted value.

## Calling copy number alterations from bulk and phased Hi-C reads

A strategy to impute broad CNVs from Hi-C data was designed as follows:

(1) For each bin $b$, its coverage $n_b$ was computed.
(2) Bins overlapping genomic gaps and bins having $n_b < \bar{R} - \gamma M$ were excluded by the analysis, with $\bar{R}$ being the genome-wide coverage median, $M$ being the median genome-wide absolute deviation of the coverage and $\gamma \in \mathbb{N}$ begin a defined parameter.
(3) $n_b$ was normalized by the median chromosome coverage ($\bar{R}_C$), obtaining $\widetilde{n}_b = n_b / \bar{R}_C$. This step enables to identify copy number changes at the subchromosomal level.
(4) The CBS algorithm[66] was run on $\widetilde{n}_b$. If a chromosome has no breakpoints, the entire chromosome is defined as a segment.
(5) For each segment $s$, the median value of its genome-wide normalized coverage, $w_s = median(n_b / \bar{R})_{b \in s}$ was computed.
(6) CNVs were defined as follows: all segments or chromosomes having $|w_s - 1| \geq t$, with $t$ being a defined threshold representing the minimum absolute difference from the genome-wide median coverage a segment has to have to be defined as a CNV.

For bulk Hi-C data of the CP-A *TP53*[-/-] colonies, a bin size of 2 Mb was used, with $t = 0.4$ and $\gamma = 7$. For phased Hi-C data, $\gamma = 4$ was used.

## Detecting significant interactions in RPE *TP53*[-/-] control cells and post-WGD tumours

HiC-DC[71] was used to compute the statistical significance of chromatin interactions at the bin level (20 kb resolution). The degree of freedom

in the hurdle negative binomial regression model was set as 6. The sample size parameter was determined by trying 20 values in the (0.5,1) range with equal distance, then choosing the maximum value that did not result in optimization failure in R. Other parameters of HiC-DC were set as default.

### RNA-seq protocol and analysis

**RNA-seq library preparation.** RNA was extracted from RPE *TP53*[-/-] control cells and WGD cells using a RNeasy Mini kit (Qiagen, 74104) following the manufacturer's protocol. Resulting RNA was processed for sequencing using a TruSeq Stranded mRNA kit (Illumina, 20020594) according to the supplier's recommendations. Libraries were then sequenced on an Illumina NovaSeq 6000 platform in a PE150 configuration.

**RNA-seq data processing and analysis.** RNA-seq fastq files were analysed using the nfcore/rnaseq pipeline (v.3.8; https://nf-co.re/rnaseq) using as the aligner star_rsem (ref. [72]), mapping the reads to the hg19 genome. Differentially expressed genes between WGD and control were determined using DESeq2 (ref. [73]). Genes having an absolute $\log_2(FC)$ above 0.1 and a *P* value of <0.01 were considered significantly differentially expressed. Gene set enrichment analysis was performed using Enrichr[74].

**Relationship between gene expression changes and LCS at WGD.** Each gene was associated to the 50 kb bin containing its transcription start site. Each gene bin was then associated to the compartment rank computed by Calder in RPE *TP53*[-/-] control and WGD and computed the difference (Δcompartment). To check the association with boundary insulation changes after WGD, the genes for which the transcription start site was in proximity of an insulation boundary in RPE *TP53*[-/-] control (±50 kb) having an insulation score below −0.1 were considered. The percentage of upregulated and downregulated genes in proximity of boundaries gaining and losing insulation and the fold changes against the percentages in the total set of genes were computed.

### scHi-C protocol and analysis

**scHi-C library preparation.** Single-cell Hi-C was performed using a modified protocol described previously[75]. Fixation of the nuclei with formaldehyde, MboI digestion and biotin fill-in was performed in a pool of 1 million cells following the same procedure as described for bulk processing. Next in-nucleus proximity ligation was done with 50 U T4 DNA ligase, 1× T4 DNA ligase buffer, 5% PEG, 1% Triton X-100 and 0.1 mg ml⁻¹ BSA at 16 °C overnight, with light mixing. Only pools of nuclei with at least 75% of the population showing an integral nuclear membrane were considered for further processing. Nuclei were strained multiple times through a 10 μm nylon net filter (Merck-Millipore, NY1009000). Sample preparation was done using DispenKit (SEED Biosciences), and single nuclei were dispensed in skirted Eppendorf twin.tec PCR plate 96-wells (Eppendorf, 0030128648) containing 50 μl of NEBuffer 3.1 (New England Biolabs, B7203S) using the single cell isolator DispenCell (SEED Biosciences) following the manufacturer's instructions. A nucleus passing through the tip of the DispenCell, which acts as a Coulter counter[76], leaves an electrical impedance change mark, which is proportional to the volume of the nucleus. For RPE *TP53*[-/-] cells, a minimum impedance change of 75 Ω was detected for the diploid nucleus and 200 Ω for the tetraploid nucleus. A lower impedance change was associated with debris or unsuccessful induction of genome doubling in the case of the WGD condition; thus, such nuclei were not considered for further processing. To avoid processing of nuclei aggregates, dispensed single nuclei associated with a threshold higher than 400 Ω and 800 Ω for diploid and tetraploid nuclei, respectively, were also discarded. These ranges were set from quality metrics of a test scHi-C batch. Specifically, an unreasonable number of unique interactions of each fragment end (for example, >2 for diploid loci) would indicate a nuclei aggregate rather than a single nucleus. Following dispensing, the single nuclei were

de-crosslinked by incubation at 65 °C overnight. Next each selected nucleus was mixed with 25 μl Dynabeads MyOne Streptavidin C1 and transferred to a 1.5 ml tube and incubated at room temperature for 1 h on a rotating wheel. The bead-bound fragments were digested with 10 U of AluI restriction enzyme (New England Biolabs, R0137) in 1× rCutSmart buffer (New England Biolabs, B6004) at 37 °C for 2 h. A-tailing reaction and adapter ligation were performed for each single cell as for the bulk Hi-C processing. Similarly, PCR amplification of the single cell libraries was performed in the same master mix as described above for bulk Hi-C, for 27–30 cycles. Libraries were cleaned using AMPure XP beads and then ran on a 2% agarose gel at 100 V for 50–60 min. Successful libraries presented a 300–700 bp smear, which was cut from the gel. DNA purification from the agarose was performed using NucleoSpin Gel and PCR clean-up (Macherey-Nagel, 740609) following the manufacturer's instructions. An additional AMPure XP bead size selection was generally necessary to remove any primer dimer contamination. Libraries were sequenced on a NextSeq550 platform (Illumina) in a PE75 configuration.

**scHi-C contact map generation and quality filtering.** For each cell, paired-end R1 and R2 fastq files were separately aligned to the hg19 reference genome using bwa mem (v.0.7.17). The scHiCExplorer pipeline[77] was used to generate Hi-C contact maps in Cooler format[62]. Quality control of scHi-C interactions was performed as previously reported[75,78]. Specifically, as it was previously reported that end-pairs covered by only one read are probably results of alignment or pairing errors of the sequencing machine[78], they were removed from the analysis. Cells for which the percentage of singleton interactions was above 75% were also removed from the analysis. Additionally, cells with fewer than 100,000 unfiltered interactions were removed. Finally, for the remaining end-pairs, which were supported by at least two duplicated reads, duplicates were removed. One cell was removed from the analysis because of the absence of interchromosomal interactions involving chromosome 1, which indicated the occurrence of technical issues during the library preparation.

**Analysis of single-cell Hi-C inter-chromosomal interactions.** Similar to bulk Hi-C, for each cell *x* passing the previously defined filters, inter-chromosomal interactions between each pair of chromosomes were aggregated and the ICE-balanced interaction matrix $I_{ICE}(x)$ was obtained. A loss of chromosomal segregation score was then defined for each cell $LCS(x) = LS(x)/(LL(x) + SS(x))$, where: $LS(x) =$ the number of balanced interactions between long and short chromosomes; $LL(x) =$ the number of balanced interactions between long and long chromosomes; and $SS(x) =$ the number of balanced interactions between short and short chromosomes.

The higher the LCS score, the higher the loss of chromosomal segregation in the analysed cell. The scores were compared between the control and WGD RPE *TP53*[-/-] populations and a Wilcoxon two-tailed *P* value was computed.

Chromosome pairs having the highest interaction enrichment or depletion in the WGD population were identified by sorting chromosome pairs for each cell by their number of balanced interactions and then computing the average rank of each pair in the control and WGD populations.

**Definition of scHi-C compartment segregation.** A simplified strategy for compartment imputation was adopted. For each cell, the intrachromosomal contact matrices of each chromosome at 1 Mb resolution were extracted. Bins having zero marginal counts were removed from the analysis. The observed over expected matrix was then calculated as previously described[11], whereby the contact decay profile was computed in logarithmically increasing bins using the cooltools package[62]. The normalized matrix was centred around 0 by subtracting 1. Next, the Pearson correlation of each pair of bins was computed and, finally,

the first two principal component analysis (PCA) components for each bin of the matrix were extracted. A and B compartments were assigned to each scHi-C bin associating the relative compartment in the bulk Hi-C by aggregating Calder segmentations of RPE *TP53*[−/−] control samples into A and B regions. Segregation scores for single cells were then computed as the silhouette score between A and B clusters[79] for each chromosome, using the two previously determined PCA components as point coordinates in the two-dimensional cartesian plane.

**Definition of single-cell compartment consistency across conditions.** scHi-C bins were clustered on the basis of the two PCA components using the KMeans algorithm[80], with the number of clusters fixed to 2. Next the adjusted Rand index[81] between the *K*-means clusters and the bulk A and B clusters was computed. Only intrachromosomal maps with a score above 0 were considered for analysis. The two *K*-means clusters were renamed into A and B, such that the correlation with the bulk compartments was maximized. The consistency of compartment calls across cells of the same biological condition (control or WGD) was calculated for each 1 Mb bin as follows: if A and B are respectively the number of cells in which the bin was called as the A or B compartment, the consistency of the bin is max(A, B)/(A + B).

**Pseudo-bulk scHi-C analysis.** Interchromosomal pseudo-bulk Hi-C values were derived from individual cell interchromosomal counts by summing all interactions for each chromosome pair, thus obtaining a 23 × 23 chromosome interaction matrix *I*. Observed and expected (O/E) chromosome level interactions were derived as described in ref. [11]. In brief, the number of observed interactions between a pair of chromosomes $I_{ij}$ was divided by the number of possible interactions between the two chromosomes $E_{ij}$. $E_{ij}$ was empirically estimated as the product between the total number of interchromosomal interactions of the first chromosome ($C_i$) and the second chromosome ($C_j$), divided by double the total number of interchromosomal interactions ($N$):

$$E_{ij} = \frac{C_i \times C_j}{2N}$$

A chromosome-level interaction enrichment matrix OE = *I/E* was obtained.

To remove noise, a correlation was computed for each chromosome pair (*i, j*) as the Spearman correlation of the vectors corresponding to the rows of the two chromosomes in the OE matrix ($OE_i$, $OE_j$). A 23 × 23 correlation matrix $\rho$ was obtained such that $\rho_{ij}$ = Spearman ($OE_i$, $OE_j$).

Differences in correlation between control and WGD in pseudo-bulk Hi-C were calculated as the $\log_2$ ratio of the correlations shifted by 1:

$$\sigma_{i,j} = \log_2\left(\frac{\rho_{ij}^{WGD} + 1}{\rho_{ij}^{Control} + 1}\right)$$

## Calling copy number alterations from scHi-C
The same procedure as defined for bulk and phased Hi-C data was performed, using as bin size of 5 Mb and a maximum absolute deviation threshold $t = 0.4$.

## scRNA-seq and analysis
**Sequencing.** scRNA-seq was performed using a Chromium Next GEM Single Cell 3′ kit v.3.1 (10x Genomics) following the manufacturer's protocol. The number of cells targeted for each condition was 3,000. Resulting libraries were sequenced on a NovaSeq 6000 or HiSeq 4000 system (Illumina).

**scRNA-seq read alignment and data processing.** The sequencing reads for all the samples (RPE *TP53*[−/−] control, RPE *TP53*[−/−] 6-weeks post-WGD, RPE *TP53*[−/−] 20-weeks post-WGD, RPE *TP53*[−/−] 20-weeks post-WGD tumours T1–T3) were aligned using the human reference transcriptome (hg19, Ensembl-87 build) with the 10x Genomics Cell Ranger pipeline (v.3.1.1)[82] with default parameters. The Seurat R package (v.3.1.5)[83] was utilized for data processing. Raw unique molecular identifier (UMI) read count data for each sample were read as Seurat data objects by keeping genes expressed in at least one cell. The following number of cells were acquired and retained after filtering for each sample: RPE *TP53*[−/−] control, 3,996 cells acquired and 2,180 cells retained; RPE *TP53*[−/−] 6-weeks post-WGD, 3,716 acquired and 3,475 cells retained; RPE *TP53*[−/−] 20-weeks post-WGD, 2,727 cells acquired and 1,976 cells retained; RPE *TP53*[−/−] 20-weeks post-WGD T1, 3,359 cells acquired and 2,851 cells retained; RPE *TP53*[−/−] 20-weeks post-WGD T2, 3,467 cells acquired and 2,721 cells retained; and RPE *TP53*[−/−] 20-weeks post-WGD, 3,790 cells acquired and 3,078 cells retained. On the intersection of genes across the dataset, 17,187 genes were found. The 17,187 genes were kept in all datasets and the merge function from the library was applied to merge the data. The dataset consisted of 17,187 genes and 21,055 cells in total. Cells having fewer than 200 and more than 10,000 genes expressed were removed from the analysis. Cells that had more than 8% of mitochondrial UMI genes expressed were also removed. The final dataset consisted of 17,187 genes and 16,281 cells. At this stage, the library depths were standardized using the NormalizeData function from Seurat, in which UMI counts for each cell were divided by the total UMI counts for that cell with a scaling factor set at 10,000. The expression matrix thus obtained was natural log-transformed. RunPCA function was run with default parameters and 2,000 highly variable genes (found using the FindVariableFeatures function). The RunUMAP function was run to obtain the uniform manifold approximation and projections, keeping default parameters and the number of PCA components, that is, ndim=1:12 was used.

**scRNA-seq copy number changes using InferCNV.** The InferCNV R package (v.1.1.0)[39] was used to call copy number changes. The UMI counts encompassing 17,187 genes and 21,055 cells were used as input to infer copy number changes. RPE control cells were considered as control samples. An inferCNV object was created using the CreateInfercnvObject object with raw UMI counts and hg19 genomic annotations as input. Run function parameters were set as follows: cutoff=0.1, cluster_by_groups=FALSE, denoise=TRUE, tumour_subcluster_partition_method = "qnorm", HMM = TRUE, HMM_type = "i6", analysis_mode = "subcluster", HMM_report_by ="subcluster". The residuals matrix generated from InferCNV containing the copy number status was used to perform hierarchical clustering using the fastcluster (v.1.2.3) Python package[84]. The Elbow method was used to determine the optimal number of clusters[85], in which the distance between the clusters was plotted against each threshold. Matplotlib (v.3.4.2) library[86] was used for data visualization.

**Differential expression analysis using scRNA-seq data.** Differentially expressed genes between RPE *TP53*[−/−] control and 20-week post-WGD tumours were detected using MAST[87] from the Seurat package. All the genes with an adjusted *P* value below 0.001 and absolute $\log_2$(FC) greater than 0.3 were considered as differentially expressed.

Transcriptional regulator scores were then computed following the SCENIC workflow[88,89], using the pyscenic package v.0.11.2 as follows: the gene regulatory network was generated using the grn command, then the regulons (transcription factors and their target genes) were identified with using the ctx command using the motif list motifs-v9-nr.hgnc-m0.001-o0.0.tbl (downloaded from cisTarget database: https://resources.aertslab.org/cistarget/). The regulons for each single cell were scored using the aucell command.

## Enrichment of differentially expressed genes on CoREs
The percentage of upregulated and downregulated genes overlapping with activating and inactivating CoREs were computed separately and

used as background the complete set of genes. Effect sizes were then computed as $\log_2(FC)$ for the four possible scenarios $\sigma_{Up}^{Activating}$, $\sigma_{Down}^{Activating}$, $\sigma_{Up}^{Inactivating}$ and $\sigma_{Down}^{Inactivating}$ where, for example,

$$\sigma_{Up}^{Activating} = \log_2 \frac{\text{\% upregulated genes overlapping an activating CoRE}}{\text{\% all genes overlapping an activating CoRE}}$$

and the other effect sizes were similarly computed.

Statistical significance was assessed by randomly selecting 100,000 times a set of genes with the same number of genes as the total number of differentially expressed genes and computing the percentage of those genes overlapping a CoRE. An empirical $P$ value was calculated as the probability that a random set of genes has a percentage of overlapping genes higher than the real set of differentially expressed genes.

## WGS

**Library preparation.** DNA was extracted from cells using a DNeasy Blood & Tissue kit (Qiagen, 69504) following the manufacturer's protocol. Library preparation was performed using TruSeq DNA PCR-Free (Illumina, 2001596) or TruSeq DNA Nano (Illumina, 2001596) kits. Libraries were sequenced on a NovaSeq 6000 system (Illumina) with a PE150 configuration.

**Mapping and processing.** Paired-end fastq files for each sample were aligned jointly to human_g1k_hs37d5 from the 1000 Genomes Phase 3 using bwa mem (v.0.7.17) and sorted with samtools (v.1.10)[90] using the sort command. Mismatches in read pairing were fixed using the fixmate command. Duplicate reads were identified using the genome analysis toolkit (GATK)[91,92] with the MarkDuplicatesSpark command. Base quality scores were corrected using BaseRecalibrator and ApplyBQSR from the GATK suite.

**CNV calling.** CNVs were identified using the Control-FREEC software[93] (v.11.6). CNVs for each sample were called with respect to the RPE *TP53*$^{-/-}$ control sample by taking as input the sample and the control bam files that were previously generated. The software was run using the following parameters:
ploidy = 2,3,4;
breakPointThreshold = .08;
intercept = 0;
window = 10000;
mateOrientation = FR;
sex = XX.

All other parameters were kept as default.

Statistical significance for each detected CNV was then computed using the assess_significance.R script provided with the Control-FREEC software.

**Finding a consensus CNV set in RPE *TP53*$^{-/-}$ 20-weeks post-WGD tumours.** The CNVs obtained using Control-FREEC having both a Kolmogorov–Smirnov $P$ value and a Wilcoxcon $P$ value below 0.01 were selected. Copy number changes of the same type (gain or loss) were merged on the basis of their overlap (±50 kb) across the three tumour samples using the merge function of bedtools (v.2.30.0)[64,65].

**Mutation calling.** SNVs were called using the Mutect2 (ref. [91]) algorithm from GATK. Blacklisted regions were derived from ref. [94] and excluded using the -XL option. F1R1 counts calculated in the pre-processing phase were provided with the option --f1r2-tar-gz. Germline variants were filtered out using the --germline-resource option with the gnomAD database (b37 version)[95]. Filtering of the mutations was performed using the FilterMutectCalls command setting --contamination-estimate to 0 and using the read orientation model computed at pre-processing with the --ob-priors option. VCF files were then converted to MAF using vcf2maf[96] and assembled into a single file. Only single-nucleotide polymorphism variants were considered in the analysis.

**Mutations having the following FILTER values were removed: normal_artifact, germline, multiallelic and clustered_events.** Then, all mutations having a gnomAD_AF value above 0.01 or were already reported in dbSNP (dbSNP_RS == 'novel') were removed. The VAF in the tumour sample and in the control sample, respectively, were calculated as t_vaf = t_alt_count/t_depth and n_vaf = n_alt_count/n_depth. Mutations such that t_vaf < 2n_vaf were removed. For the remaining mutations, the overlap between samples was checked (n_samples). Mutations found in only one sample and having FILTER != 'PASS' or t_depth < 6 were removed. Conversely, if a mutation is found in only one sample but FILTER == 'PASS' and t_depth >= 6, it was kept.

Oncogenicity of the variants was assessed using OncoKB[97].

To study how many mutations were gained after WGD and to compare them to the ones accumulated in the same time frame without WGD induction, the number of mutations found in 6 weeks post-WGD and 6 weeks control samples were counted, with the following exclusion criteria: mutations already detected in the *TP53*$^{-/-}$ RPE control sample; mutations shared between 6-weeks post-WGD and 6-weeks control samples, as these mutations were most likely already present in the *TP53*$^{-/-}$ RPE control sample and went undetected; mutations shared between 6-weeks control and WGD samples for the same reason.

## ChIP–seq and analysis

**ChIP–seq.** ChIP was performed using a SimpleChIP Enzymatic Chromatin IP kit (Cell Signaling Technology, 91820S) following the manufacturer's protocol. In brief, 2–4 million cells per condition were fixed in 1% formaldehyde for 10 min at room temperature. The reaction was quenched with 1× glycine and the cell pellet was washed with PBS. Cells were lysed, nuclei were digested with 1,000 U micrococcal nuclease for 20 min at 37 °C and briefly sonicated at 80 V peak incidence power, 200 cycles per burst and 5% duty factor for 90 s on an E220 focused-ultrasonicator (Covaris) to disrupt the nuclear membrane. Next 1–5 µg digested chromatin, with or without 0.1–0.5 µg digested mouse chromatin for spike-in normalization, was incubated at 4 °C overnight with one of the following antibodies, at the recommended dilutions: anti-acetyl-histone H3 (Lys27) (Cell Signaling Technology, 8173), anti-trimethyl-histone H3 (Lys9) (Cell Signaling Technology, 13969), anti-CTCF (active motif, 61311), anti-trimethyl-histone H3 (Lys4) (Cell Signaling Technology, 9751) or trimethyl-histone H3 (Lys27) (Cell Signaling Technology, 9733). Antibody-bound chromatin was precipitated using protein G magnetic beads and crosslink reversal was performed using NaCl and proteinase K. Resulting DNA was purified using spin columns. Library preparation for sequencing from the chromatin immunoprecipitated DNA was performed using a NEBNext Ultra II DNA Library Prep kit for Illumina (New England Bio-Labs, E7645) following the manufacturer's protocol. DNA fragments were processed for end repair, followed by stubby adaptor ligation. DNA was size-selected using AMPure XP beads (Beckman-Coulter, A63881). Adaptor-ligated DNA was PCR-amplified using indexed NEBNext Multiplex Oligos for Illumina (New England Biolabs, E7335). The resulting libraries were sequenced on a NextSeq 500 system (Illumina) in a PE37/38 configuration.

**ChIP–seq data analysis.** Fastq files were processed using the nfcore/chipseq pipeline (v.1.2.2)[98,99] (https://nf-co.re/chipseq/) with default parameters, aligning the reads to the hg19 genome. For each sample, the fold change against the input experiment was then computed using the bamComapre command from the deepTools package (v.3.5.1)[100] setting –scaleFactorsMethod readCount, --extendReads, --operation ratio and --binSize 100). MACS3 (ref. [101]) was used for peak calling. Peaks with a $q$ value lower than 0.1 and a minimum fold change of 1.5 were retained. A consensus set of peaks for CTCF and H3K9me3

was then created from control and WGD samples from the RPE $TP53^{-/-}$ and CP-A $TP53^{-/-}$ cell lines separately by aggregating peaks that were closer than 10 kb for H3K9me3 and that directly overlapped for CTCF in the two samples. The maximum ChIP–seq signal (fold change over input) in control and WGD was associated for each consensus peak in the collection.

## Spike-in normalization
Spike-in normalization on the CTCF signal was performed as previously described[102].

**CTCF peaks at Hi-C boundaries.** Insulation boundaries shared between control and WGD samples from RPE $TP53^{-/-}$ samples identified from Hi-C were associated to CTCF peaks by taking all the peaks lying in on the genomic bin of the boundary (±10 kb). Boundaries were then associated to a CTCF score representing the sum of all the CTCF peak signals at the boundary.

**Comparing ChIP–seq signal differences to Hi-C compartment differences.** Fold change signals computed at the previous step were scaled for each 10/50 bp bin (having value $x$) computing the $\log_2(x+1)$ value. Scaled values were then binned at 50 kb resolution, taking the average signal in the bin. Finally, the 50 kb binned signal was normalized by dividing each bin by the chromosome median value. For each RPE $TP53^{-/-}$ 20-weeks post-WGD tumour Hi-C sample, the binned Calder ranks at 50 kb were matched to the binned ChIP–seq values for each histone modification. Finally, differences in Calder rank ($\Delta$rank) were compared with differences in ChIP–seq signal ($\Delta hm_1$, $\Delta hm_2$, ...) for each tumour against the RPE $TP53^{-/-}$ control sample.

**Correlation between CoRE regions and differences in ChIP–seq signal.** To study the relationship between subcompartment repositioning events and epigenetic changes in RPE $TP53^{-/-}$ post-WGD tumours, the average ChIP–seq signal difference for each histone mark was assigned to each CoRE (average($\Delta hm_1$), average($\Delta hm_2$), ...), where $\Delta hm_1 = hm_1^{Tumour} - hm_1^{Control}$ for each 50 kb bin in the CoRE region. The Spearman correlation was computed between subcompartment repositioning scores and the average epigenetic differences of the CoREs. To estimate the significance of the observed correlations, an empirical $P$ value was computed by randomly sampling a number of regions of the same size of the observed CoREs across the genome 1,000 times and recomputing the correlation value. A $P$ value was obtained, corresponding to the number of times the expected correlation was greater or equal in absolute value to the one observed, divided by the total number of random trials.

## Reporting summary
Further information on research design is available in the Nature Portfolio Reporting Summary linked to this article.

## Data availability
All raw data have been deposited into the NCBI's Gene Expression Omnibus and are accessible through GEO Series accession number GSE222391. Processed Hi-C data together with compartment domain calls by Calder are available at https://doi.org/10.5281/zenodo.7351767. ChIP-seq data, scRNA-seq matrices and copy number profiles are available at https://doi.org/10.5281/zenodo.7351776. Source data are provided with this paper.

## Code availability
Analysis code regarding LCS and the identification of CoREs is available at https://github.com/CSOgroup/WGD.

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

**Acknowledgements** We thank J. Korbel for sharing the RPE cells and RPE *TP53*−/− cells used in this study; and the staff at the EPFL facilities (the Flow Cytometry Core Facility, the Bioimaging and Optics Platform, the Center of PhenoGenomics), in particular the team at the Gene Expression Core Facility led by B. Mangeat. This work was supported by the EPFL (to E.O.) and UNIL (to G.C.) internal funds, the Swiss National Science Foundation (to E.O. and G.C.) and the Swiss Cancer League (to E.O. and G.C.).

**Author contributions** R.A.L. designed and performed all the experiments. L.N. designed and developed new methods to analyse the data and performed all the computational analyses. Y.L. contributed to Hi-C data analyses. J.D.-M. contributed to the analyses of genomic data. A.I. and D.T. contributed to scRNA-seq data analyses. N.K. contributed to sample preparation. G.C. and E.O. designed and supervised the study. R.A.L., L.N., G.C. and E.O. wrote the manuscript with input from all the authors.

**Funding** Open access funding provided by EPFL Lausanne.

**Competing interests** The authors declare no competing interests.

**Additional information**
**Correspondence and requests for materials** should be addressed to Giovanni Ciriello or Elisa Oricchio.

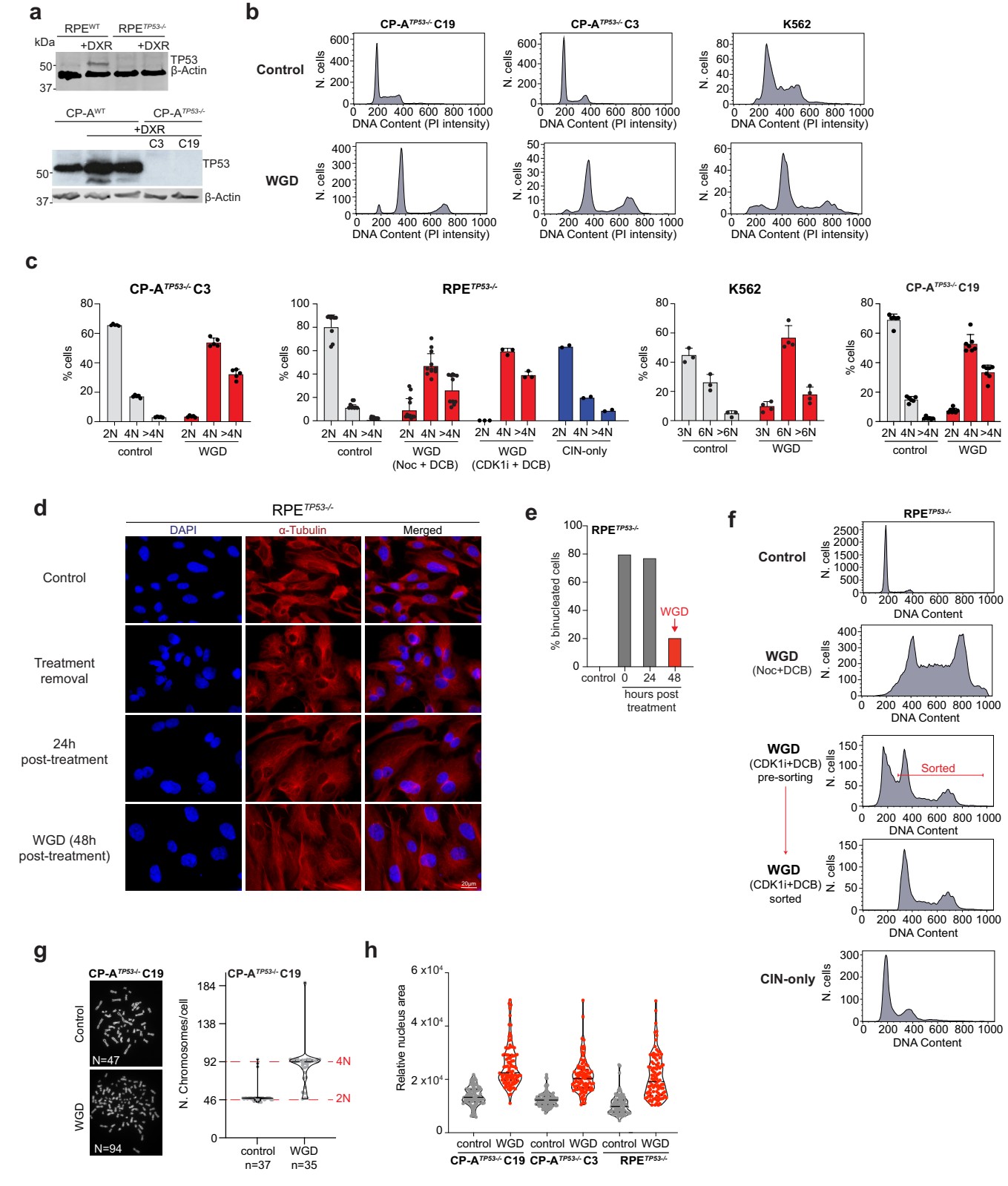

**Extended Data Fig. 1** | See next page for caption.

**Extended Data Fig. 1 | Whole-genome doubling induction. a**, Immunoblot presenting TP53 and beta-actin protein levels in RPE and CP-A, *TP53* WT and *TP53*$^{-/-}$ cell lines. Samples treated with the TP53-pathway inducer doxorubicin (DXR) were included. (n = 1 experiment) **b**, Example of PI-based cell cycle staining histograms for control and WGD populations of CP-A$^{TP53-/-}$ (clone 19 and clone 3) and K562 cells. **c**, Percentage of cells of different ploidies for CP-A$^{TP53-/-}$ C3 control (n = 5) and WGD (n = 5); RPE$^{TP53-/-}$ control (n = 10), WGD (n = 11), WGD CDK1i+DCB (n = 3), and CIN-only (n = 2); K562 control (n = 3) and WGD (n = 4); and CP-A$^{TP53-/-}$ clone 19 control (n = 6) and WGD (n = 6) populations. Data are mean ± s.d.; all data points are shown; n indicates the number of independent experiments. **d**, Immunofluorescence images of RPE$^{TP53-/-}$ cells showing transition from mononucleated diploids to mononucleated tetraploids via binucleated intermediates during WGD induction. Nuclei are shown in blue (DAPI) and cytoskeleton (alpha-tubulin) in red (n = 100 cells from 2 WGD inductions per condition were considered). **e**, Percentage of binucleated cells in RPE$^{TP53-/-}$ control, at WGD-induction treatment removal, and 24 and 48 h (WGD point) post-treatment removal populations. **f**, Example of PI-based cell cycle staining histograms for RPE$^{TP53-/-}$ control, WGD induced via nocodazole (Noc) + DCB, WGD induced via CDK1i+DCB treatment before and after cell sorting, and CIN-only populations. **g**, Karyotyping of CP-A$^{TP53-/-}$ clone 19 control and WGD cells. Example images of metaphase spreads and quantification of chromosomes per cell are shown. **h**, Relative nucleus area measured for control and WGD CP-A$^{TP53-/-}$ (clone 19 and clone 3), and RPE$^{TP53-/-}$ cells (n = 100, from 2 independent WGD inductions).

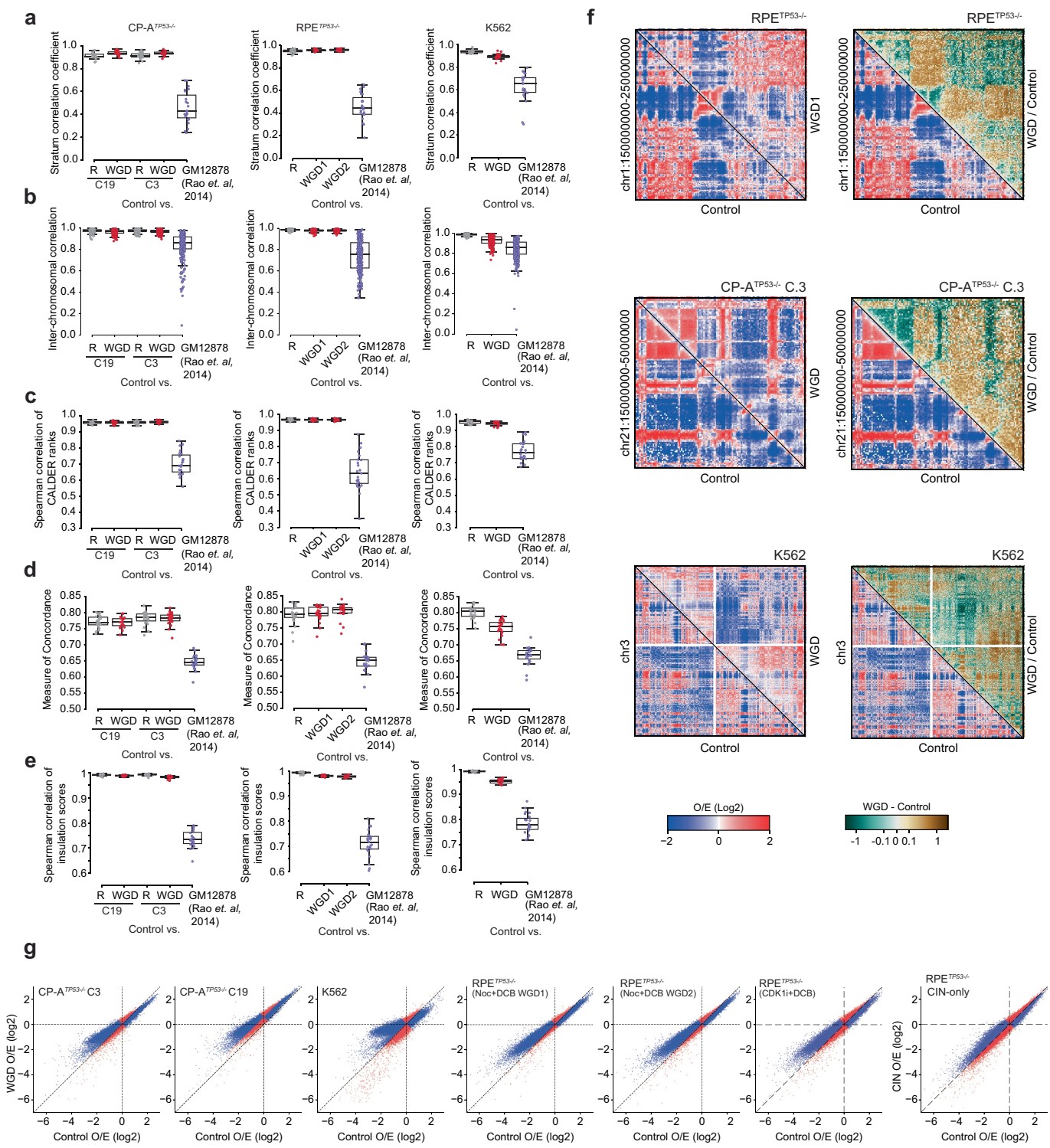

**Extended Data Fig. 2 | Chromatin topological similarities between control and WGD. a-e**, Comparisons of Hi-C contact maps between control diploid cells and replicate (R) or WGD cells of the indicated cell line, or between control diploid cells of the indicated cell line and GM12878 cells. **a**, Stratum correlation coefficient of matching intra-chromosomal maps (n = 23 per boxplot), **b**, inter-chromosomal Hi-C contact correlation of matching inter-chromosomal maps (n = 253), **c**, Spearman's correlation of compartment domain ranks computed by Calder of matching intra-chromosomal maps (n = 23), **d**, measure of concordance of TADs for each matching intra-chromosomal maps (n = 23), **e**, Spearman's correlation of insulation scores of matching intra-chromosomal maps (n = 23). Boxplots: central line is the median, bounding box corresponds

to the 25th–75th percentiles, and the whiskers extend up to 1.5 times the interquartile range. **f**, Examples of intrachromosomal loss of chromatin segregation in RPE$^{TP53-/-}$ (top), CP-A$^{TP53-/-}$ C3 (centre) and K562 (bottom) models. Intrachromosomal observed/expected maps in Control and WGD samples are shown on the right in log2 scale, with Control in the lower and WGD (left) or Observed/Expected (right) in the upper triangular matrices. **g**, Relationship between observed/expected intra-chromosomal values in control vs. WGD samples. Each dot represents an inter-chromosomal Hi-C interaction. Red dots denote higher, whereas blue dots lower interaction signals in WGD compared to control.

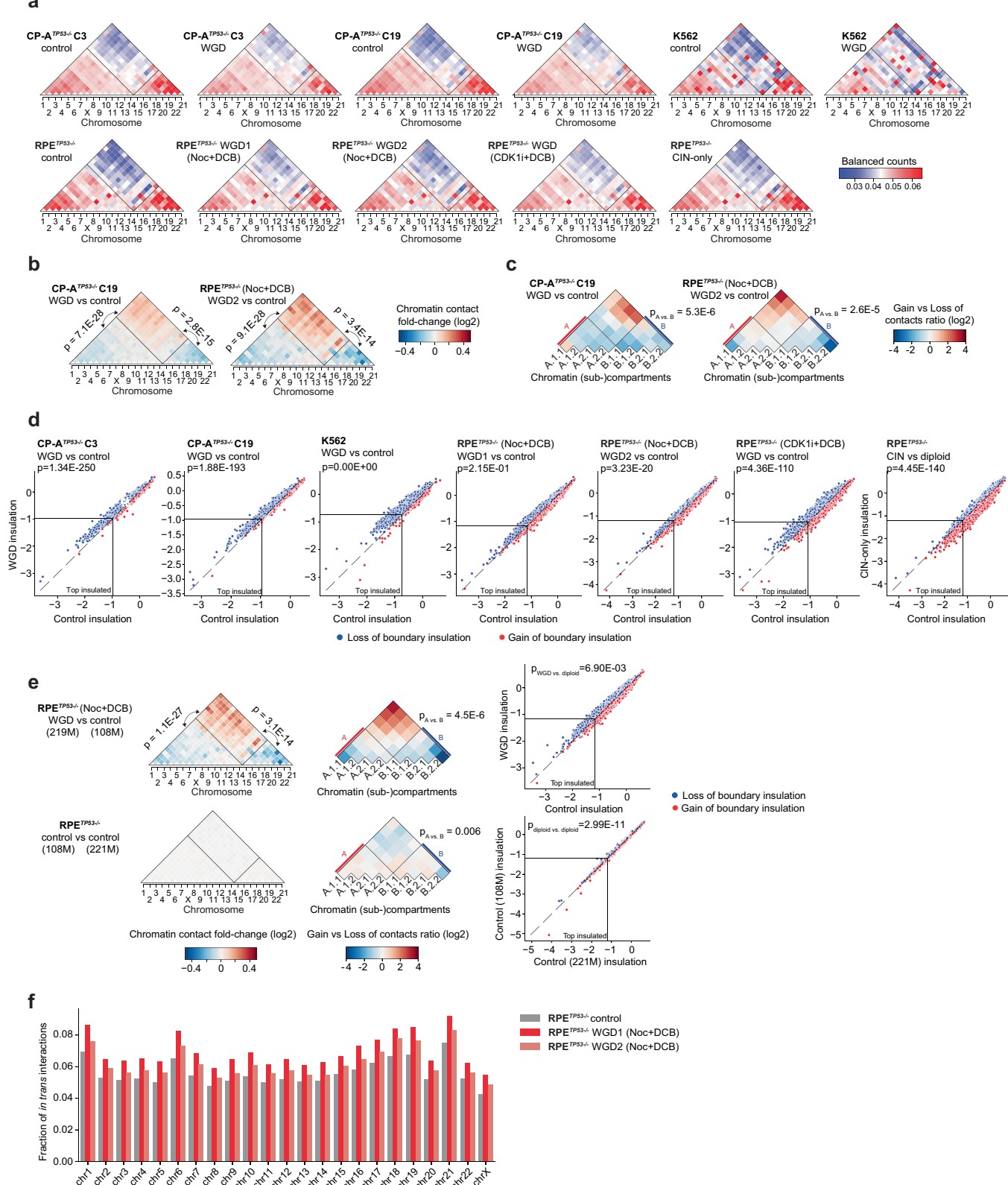

**Extended Data Fig. 3** | See next page for caption.

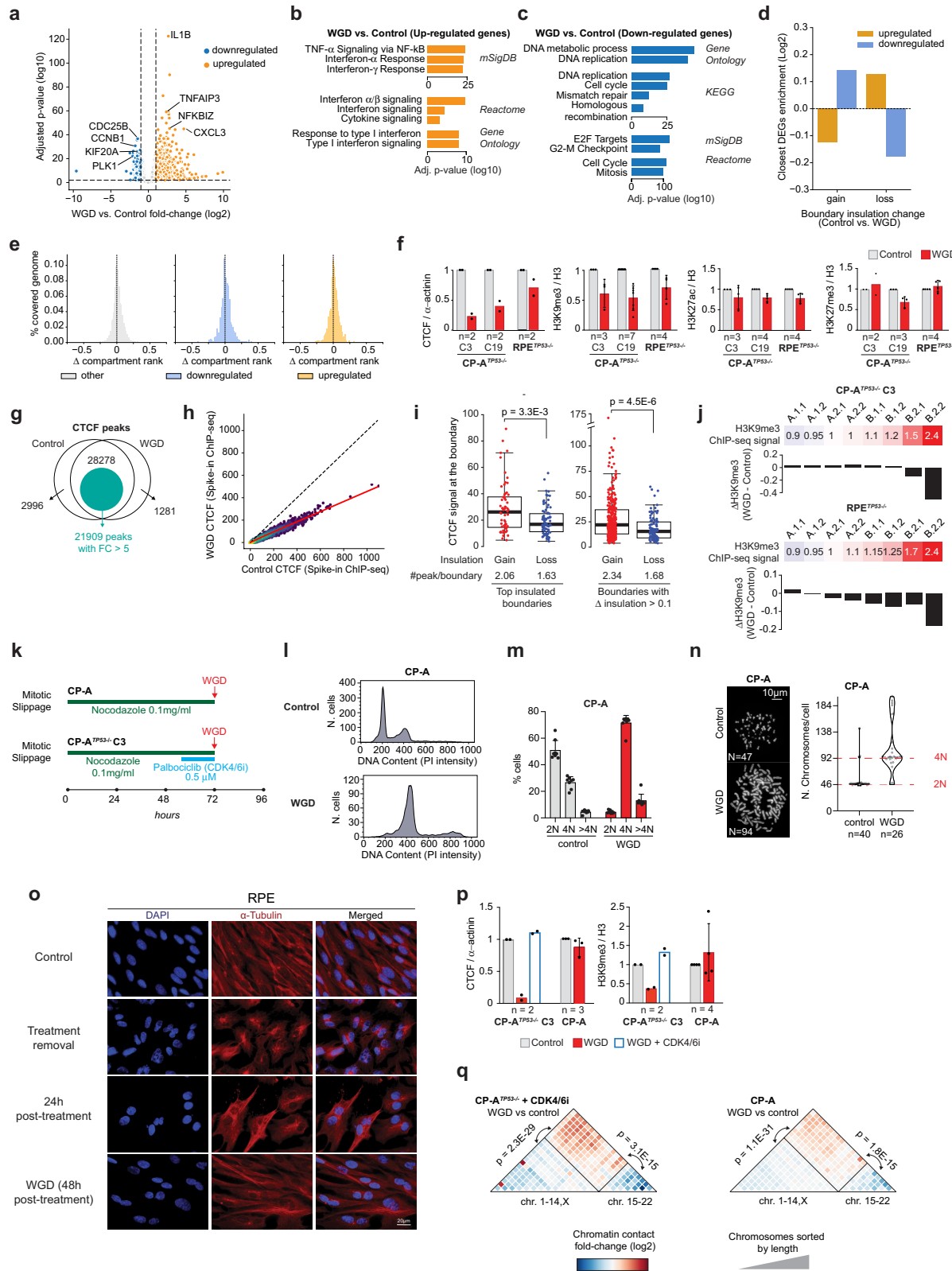

**d**, Boundary insulation scores in Control (x-axis) and WGD or CIN-only (y-axis) samples. Only boundaries shared between the two conditions are considered. *P*-values are calculated by two-tailed Wilcoxon test. **e**, Ratios of inter-chromosomal contact enrichments (left), genomic bins belonging to the indicated sub-compartments that gain vs. lose contacts (middle), and boundary insulation scores (right) between samples with different sequencing coverage to account for coverage per genome size discrepancies between the control and WGD conditions. Conditions considered were RPE$^{TP53-/-}$ WGD (219 million reads) vs control (108 million reads), and RPE$^{TP53-/-}$ control (108 million reads) versus control (221 million reads). All *P*-values are calculated by two-tailed Wilcoxon test. **f**, Fraction of interactions between homologous copies of each chromosome in RPE$^{TP53-/-}$ control and WGD cells.

**Extended Data Fig. 4** | See next page for caption.

**Extended Data Fig. 4 | Effect of WGD on the transcriptome and on CTCF and H3K9me3-mediated chromatin compaction. a**, Volcano plot showing the fold change and its associated adjusted *P*-value calculated by MAST for each differentially expressed genes between WGD condition and control in RPE$^{TP53-/-}$ cells. Significantly upregulated genes are shown in yellow, whereas significantly downregulated genes in blue. **b, c**, Gene set enrichment analysis for the significantly upregulated **(b)** and significantly downregulated **(c)** genes in RPE$^{TP53-/-}$ WGD versus control cells. Databases utilized for the analysis for each gene set are indicated. Adjusted *p*-value calculated by hypergeometric test. **d**, Differentially expressed genes enrichment at boundaries that gain or lose insulation post-WGD. Only boundaries having an insulation score in Control below −0.1 were considered. The enrichment is calculated using as background the complete set of genes. **e**, Distribution of compartment rank differences between Control and WGD RPE$^{TP53-/-}$ cells estimated by Calder (Δcompartment rank: negative values indicate a shift towards less active compartment, positive values indicate a shift towards a more active compartment) in bins overlapping gene promoters. Distributions relative to genes not changing expression (grey), downregulated genes (blue) and upregulated genes (orange) are shown. **f**, Quantification of indicated proteins in diploid control and WGD CP-A$^{TP53-/-}$ (clones 3 and 19) and RPE$^{TP53-/-}$ cells. Data is mean ± s.d. **g**, Number of CTCF peaks identified in RPE$^{TP53-/-}$ control and WGD samples and their overlap. **h**, ChIP-seq signal of CTCF shared peaks between control and WGD conditions in CP-A$^{TP53-/-}$ clone 3 cells. Dots are coloured by point density in log10 scale, and the regression line (red) is calculated using locally weighted scatterplot smoothing (LOWESS). **i**, Aggregated CTCF peak signals in RPE$^{TP53-/-}$ control cells at boundaries that gain (*red*) or lose (*blue*) insulation in WGD cells. Top-300 boundaries (*left*, gain n = 62, lose n = 81) and boundaries having an absolute change of insulation greater than 0.1 (*right*, gain n = 340, lose n = 101) are considered. *p*-values are computed with a two-sided Mann-Whitney U test. Boxplots: central line is the median, bounding box corresponds to the 25th–75th percentiles, and the whiskers extend up to 1.5 times the interquartile range **j**, H3K9me3 enrichment across chromatin sub-compartments and changes in H3K9me3 signal, based on ChIP-Seq data, between WGD and control conditions in CP-A$^{TP53-/-}$ clone 3 (above) and RPE$^{TP53-/-}$ (below) cells. **k**, Schematic representation of WGD induction in *TP53* wild type CP-A cells and in CP-A$^{TP53-/-}$ treated with 0.5 μM of the CDK4/6 inhibitor Palbociclib. **l**, Example of PI-based cell cycle staining histograms for control and WGD populations of CP-A cells. **m**, Percentage of cells of distinct ploidies as of CP-A control (n = 8) and WGD (n = 8) populations. Data are mean ± s.d.; n indicates the number of independent experiments. **n**, Karyotyping of CP-A control and WGD cells. Example images of metaphase spreads and quantification of chromosomes per cell are shown. **o**, Immunofluorescence images of RPE cells, Control and WGD, illustrating the maintenance of binucleated cells post-WGD. Nuclei are shown in blue (DAPI) and cytoskeleton (alpha-tubulin) in red (n = 100 cells from 2 WGD inductions per condition were tested). **p**, Quantification of indicated proteins in diploid control and WGD CP-A cells and in WGD CP-A$^{TP53-/-}$ cells treated with CDK4/6 inhibitor. Data is mean ± s.d. **q**, Ratios of inter-chromosomal contact enrichments (observed vs. expected) between CP-A and CP-A$^{TP53-/-}$ treated with CDK4/6i WGD and control samples. Chromosomes were sorted by length. Long chromosomes (chr1 to chr14 and chrX) and short chromosomes (chr15 to chr22) are highlighted. *p*-values are calculated by two-tailed Wilcoxon test.

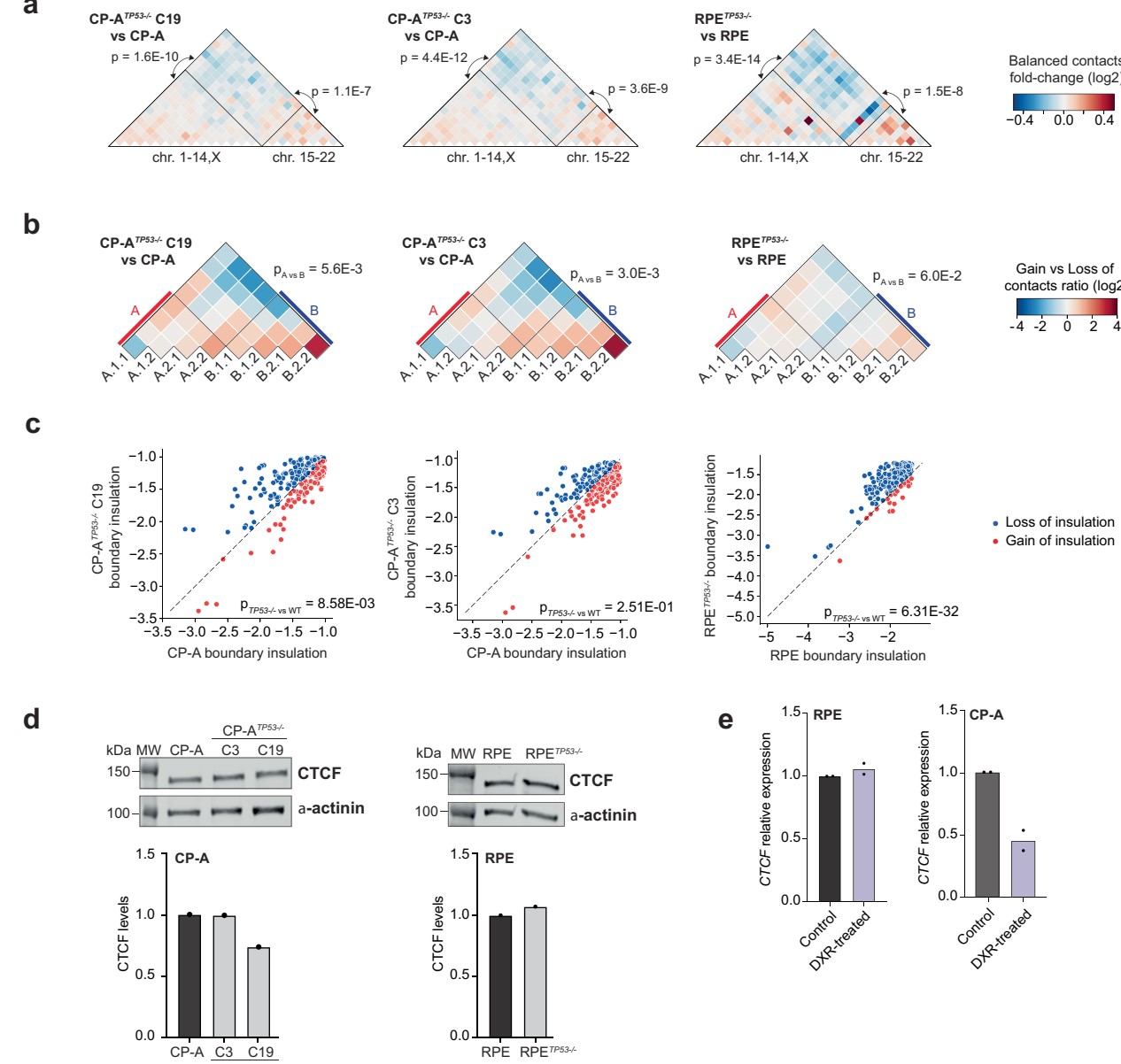

**Extended Data Fig. 5 | *TP53* loss is not the cause of LCS. a**, Ratios of inter-chromosomal contact enrichments (observed vs. expected) between *TP53*⁻/⁻ and *TP53* WT samples for CP-A and RPE cells. Chromosomes were sorted by length. Long chromosomes (chr1 to chr14 and chrX) and short chromosomes (chr15 to chr22) are highlighted. *p*-values are calculated by two-tailed Wilcoxon test. **b**, Heatmap of ratios of genomic bins belonging to the indicated sub-compartments that gain vs. lose contacts between *TP53*⁻/⁻ and TP53 WT samples for CP-A and RPE cells. **c**, Boundary insulation scores in *TP53* WT and *TP53*⁻/⁻ for CP-A and RPE samples. Only boundaries shared between the two conditions are considered. **b-c** *p*-values calculated by two-tailed Wilcoxon test. **d**, Representative image of immunoblot (above) and quantification (below) of CTCF protein expression in *TP53* WT and *TP53*⁻/⁻ CP-A (left) and RPE (right) cells (n = 1 experiment). **e**, *CTCF* mRNA expression levels in RPE and CP-A *TP53* WT cell lines after treatment with 3 μM doxorubicin (n = 2 independent experiments).

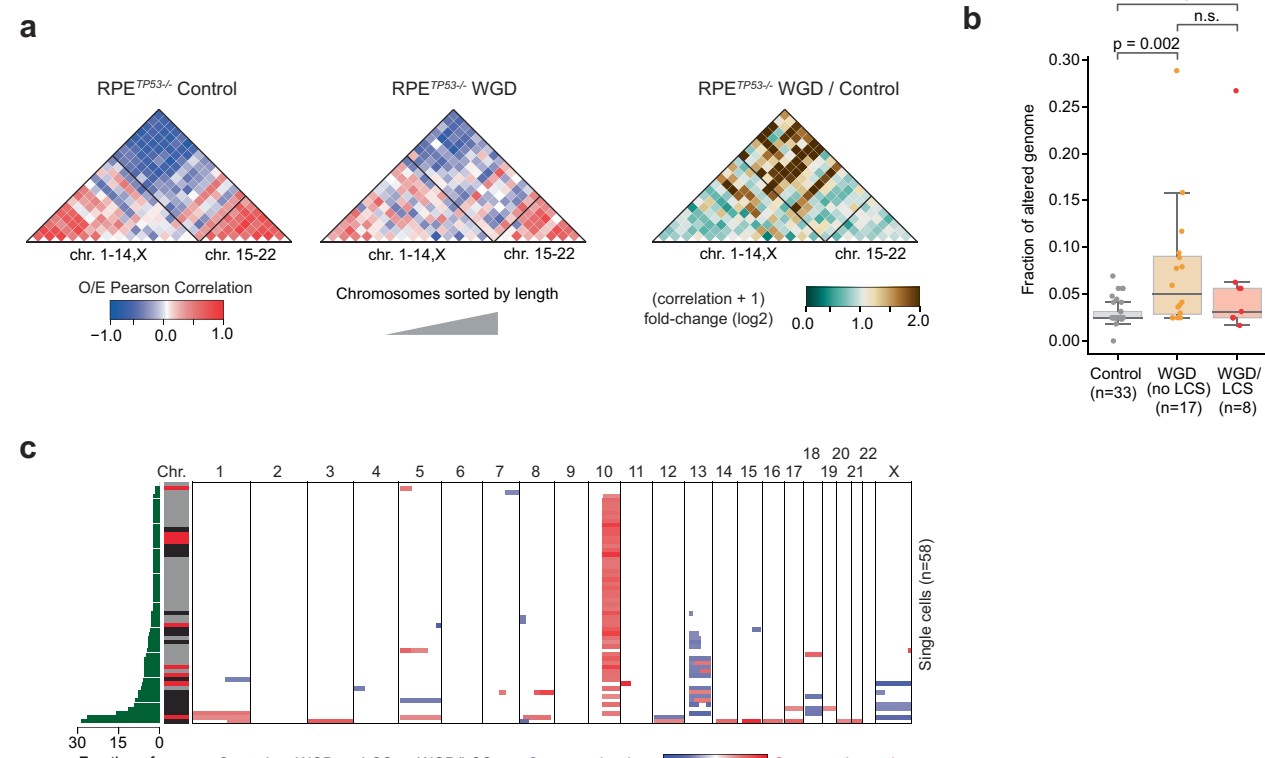

**Extended Data Fig. 6 | scHi-C recovers loss of chromosome segregation and chromosomal instability features. a**, Chromosome interactions in pseudo-bulk Hi-C data derived from the aggregation of single-cell Hi-C data for RPE^TP53−/− control and WGD, and Log2 fold-changes between WGD and Control (right) conditions. For individual conditions, interactions are shown as the Pearson correlation of the chromosome level Observed/Expected values for each pair of chromosomes. For WGD/Control condition, control and WGD correlation values were increased by 1 before computing the ratio. **b**, Fraction of altered genome in RPE^TP53−/− control, WGD with no LCS, and WGD/LCS single cell groups. *p*-value calculated by two-tailed Wilcoxon test. Boxplots: central line is the median, bounding box corresponds to the 25th–75th percentiles, and the whiskers extend up to 1.5 times the interquartile range. **c**, Relative copy number changes in each of the single cells queried by scHi-C. Cells are sorted by fraction of altered genome.

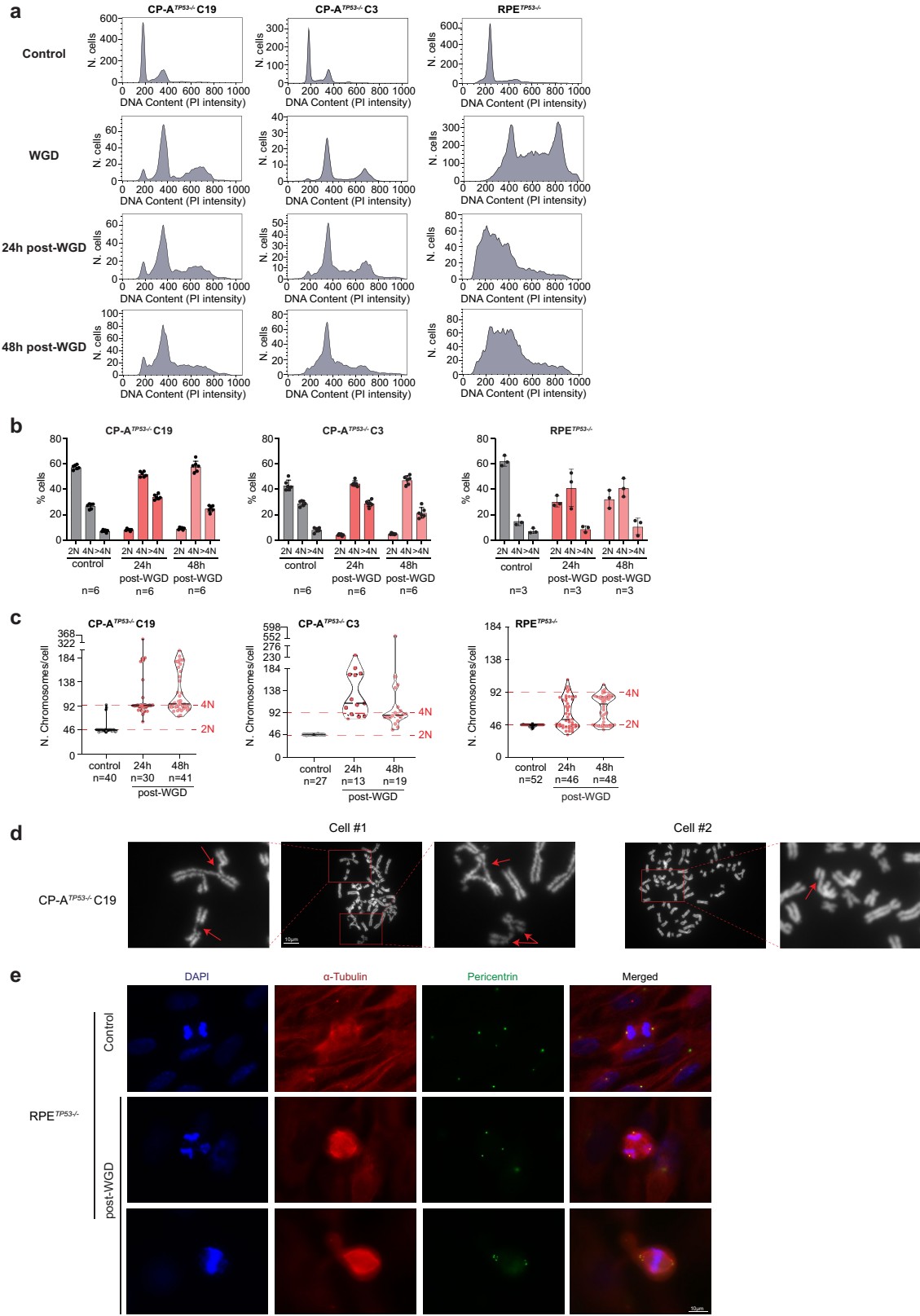

**Extended Data Fig. 7 | Chromosomal instability and rapid aneuploidization post-WGD.** Changes in ploidy short term post-WGD (24 and 48 h) in CP-A$^{TP53-/-}$ (clone 19 and clone 3), and RPE$^{TP53-/-}$ as shown by **a,b**, PI-based cell cycle staining **(a)**, quantification of percentage of cells of distinct ploidies (2N, 4N, >4N) **(b)** and **c**, Quantification of chromosomes per cell via karyotyping. Data are mean ± s.d.; Number of independent experiments is indicated. Violin plots: dashed line is median **d**, Examples of chromosomal instability markers (telomere fusion and chromosome breakages) in CP-A$^{TP53-/-}$ clone 19 cells 24 h post-WGD. n = 3 out of 30 total karyotypes presented such markers. Red arrows indicate affected chromosomes. **e**, Examples of bipolar and multipolar division in RPE$^{TP53-/-}$ cells (Control and 24 h post-WGD). n = 5 dividing cells were detected for each phenotype out of ~100 total cells. Nuclei are stained in blue (DAPI), cytoskeleton (alpha-tubulin) in red, and centrosomes (pericentrin) in green.

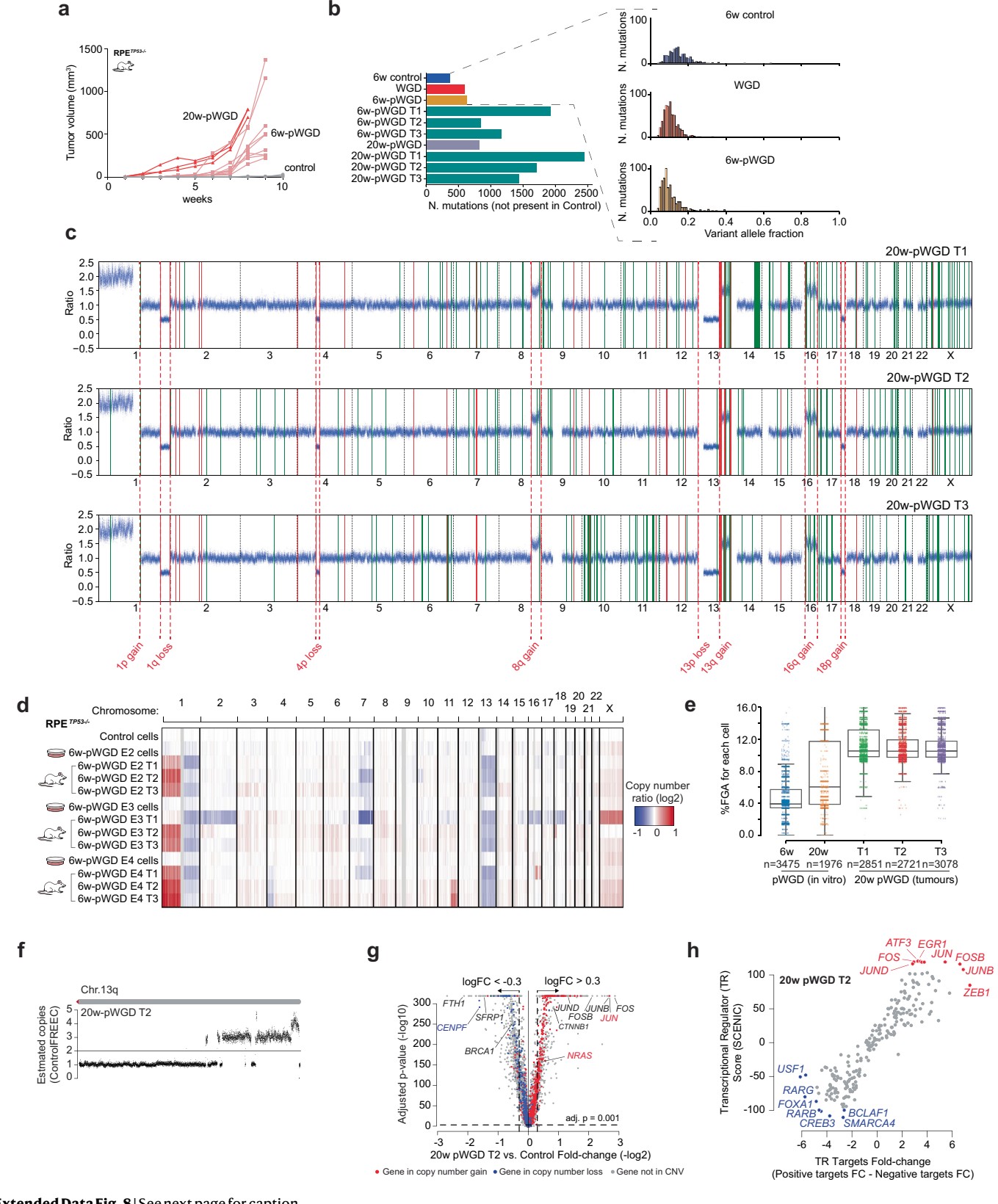

**Extended Data Fig. 8** | See next page for caption.

**Extended Data Fig. 8 | Tumour progression, ploidy, copy number changes, and gene expression changes post-WGD. a**, Evolution of the tumour volumes from the time of subcutaneous injection in NSG mice. Tumours were derived from RPE$^{TP53-/-}$ Control (n = 3), 6 weeks post-WGD (n = 9), and 20 weeks post-WGD (n = 3) cells. **b**, Mutational burden of RPE$^{TP53-/-}$ *in vitro* and *in vivo* samples defined as the number of SNPs compared to control. Control cells maintained in cell culture for 6 weeks were included (6w control). Variant allele frequency is displayed for 6w control, WGD, and 6w-pWGD samples. **c**, Breakpoints associated with copy number variants across the RPE$^{TP53-/-}$ 20 weeks post-WGD tumour samples. Copy number profile is shown as coverage ratio between tumour samples and control. Red lines mark shared breakpoints, green lines mark sample-specific breakpoints. Breakpoints defining the main CNVs shared by all tumour samples are highlighted with dashed red lines. **d**, Copy number alterations determined via WGS in RPE$^{TP53-/-}$ Control and post-WGD cells (in vitro and in vivo) derived from distinct WGD induction experiments (E2, E3, E4) **e**, Fraction of altered genome calculated for each single cell, using copy number changes derived by the InferCNV pipeline applied to the single-cell RNA-seq data for the RPE$^{TP53-/-}$ post-WGD samples (in vivo and in vitro). Boxplots: central line is the median, bounding box corresponds to the 25th–75th percentiles, and the whiskers extend up to 1.5 times the interquartile range. **f**, Estimated chromosomal copy number alterations of chromosome 13q from WGS in RPE$^{TP53-/-}$ 20-weeks post-WGD tumour 2. **g**, Volcano plot showing differential expression of significant genes in RPE$^{TP53-/-}$ control versus 20 weeks post-WGD derived Tumour 2 sample. Log2 fold-change of gene expression and adjusted *p*-value were calculated with the MAST algorithm. Genes overlapping a CNV are highlighted in blue (losses) and red (gains). **h**, Highest scoring transcription regulators of differentially expressed genes in the 20 weeks post-WGD derived Tumour 2 identified using the SCENIC workflow. Expression changes of the transcription factors' targets are presented on the x-axis, while on the y-axis shows the SCENIC score for each regulon.

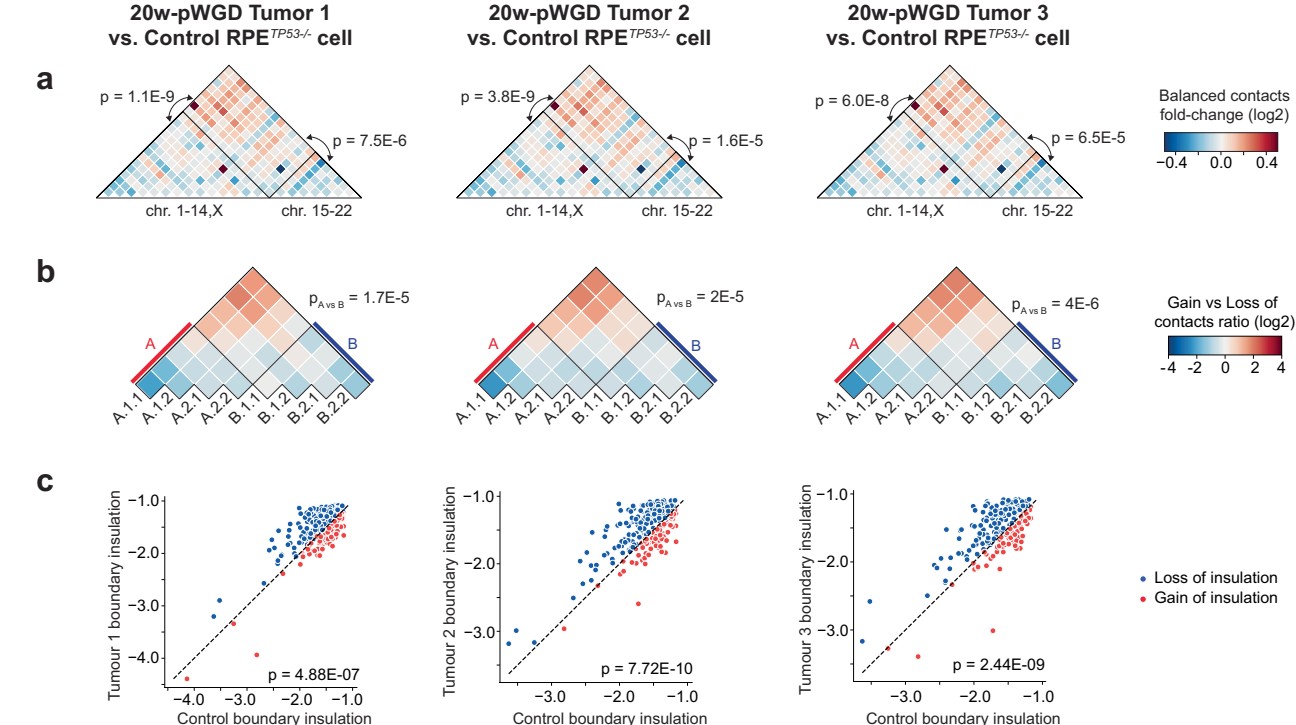

**Extended Data Fig. 9 | Loss of chromatin segregation in post-WGD tumours.** **a**, Ratios of inter-chromosomal contact enrichments (observed vs. expected) between RPE*TP53−/−* post-WGD tumours and control. Chromosomes were sorted by length. Long chromosomes (chr1 to chr14 and chrX) and short chromosomes (chr15 to chr22) are highlighted. **b**, Heatmap of ratios of genomic bins belonging to the indicated sub-compartments that gain vs. lose contacts between RPE*TP53−/−* post-WGD tumours and control. **c**, Boundary insulation scores in RPE*TP53−/−* control and post-WGD tumours. Only boundaries shared between the two conditions are considered. **a**–**c** *p*-values are calculated by two-tailed Wilcoxon test.

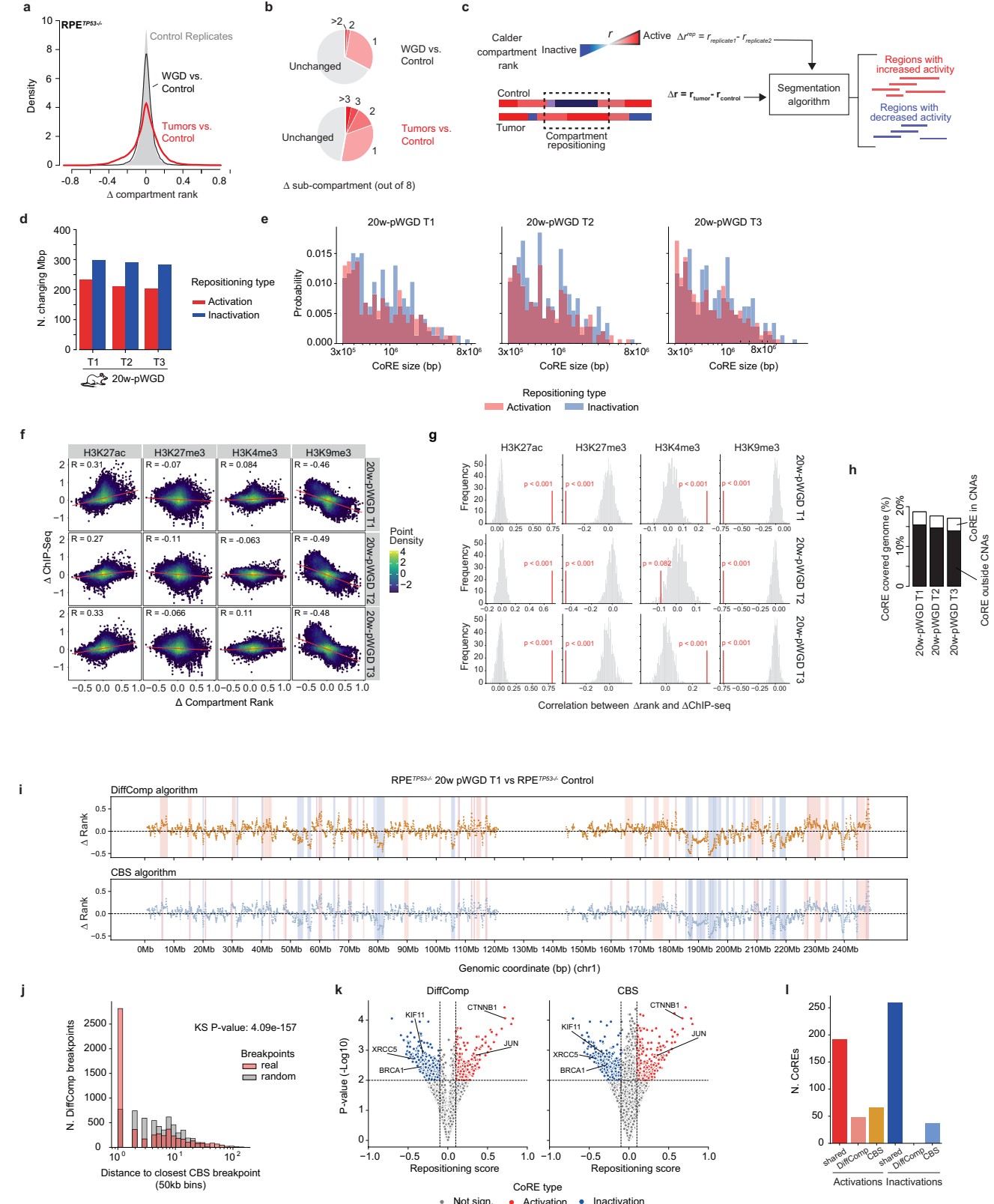

**Extended Data Fig. 10** | See next page for caption.

**Extended Data Fig. 10 | Properties of compartment repositioning events in post-WGD tumours and their consistency across different change-point detection algorithms. a**, Distribution of compartment rank differences estimated by Calder (Δcompartment rank: negative values indicate a shift towards less active compartment, positive values indicate a shift towards a more active compartment) between RPE$^{TP53-/-}$ control replicates (grey fill), WGD vs control (black line), and 20 weeks post-WGD tumours vs control (red line). **b**, Distribution of sub-compartment differences estimated by Calder (Δsub-compartment) in WGD and 20 weeks post-WGD tumours, compared to control. **c**, Schematic representation of the CoRE detection algorithm. The genome is partitioned in segments having the same Calder rank (blue representing Inactive, red representing Active regions). The differential rank values (Δr) between the tumour and the control samples were segmented using a recursive strategy. A null hypothesis of compartment repositioning from replicate comparisons (Δr$^{rep}$) was derived and tested to compute the significance of each computed segment. The output of the algorithm is a list of activating and in-activating regions. **d**, Number of Mbp covered by activating and inactivating CoRE regions in the three 20 weeks post-WGD tumours. **e**, Distribution of CoRE region sizes (in bp) between activating and inactivating regions in the three 20 weeks post-WGD tumours. **f**, The relationship between differential compartment rank (x-axis) and differential ChIP-seq signal (y-axis) between the tumour and control sample for each tumour and each histone mark. Points represent 50-kb genomic bins. **g**, Expected distributions of correlations (grey) between changes in sub-compartment rank and changes in histone modification signal across different histone marks and different 20 weeks post-WGD tumours in RPE$^{TP53-/-}$ cells. Observed correlations are highlighted as red lines. Empirical $p$-values are shown. $p$-values calculated by data permutation (n = 1000) **h**, Percentage of the genome covered by a CoRE in the three RPE$^{TP53-/-}$ 20 weeks post-WGD derived tumour samples. **i**, Example of CoREs detected using the DiffComp or CBS algorithm for delta-rank segmentation. **j**, Distribution of distances between DiffComp and CBS breakpoints (red) and between DiffComp breakpoints and randomly generated ones by shuffling the CBS breakpoints across the genome. $p$-value calculated by Kolmogorov-Smirnov test. **k**, Volcano plot representation of CoREs detected by DiffComp (left) and CBS (right). CoREs involving the selected oncogenes and tumour suppressors discussed in the manuscript are labelled and detected by both approaches. $p$-values calculated by DiffComp. **l**, Number of activating and inactivating CoREs detected by both algorithm (shared) or by only one of them.

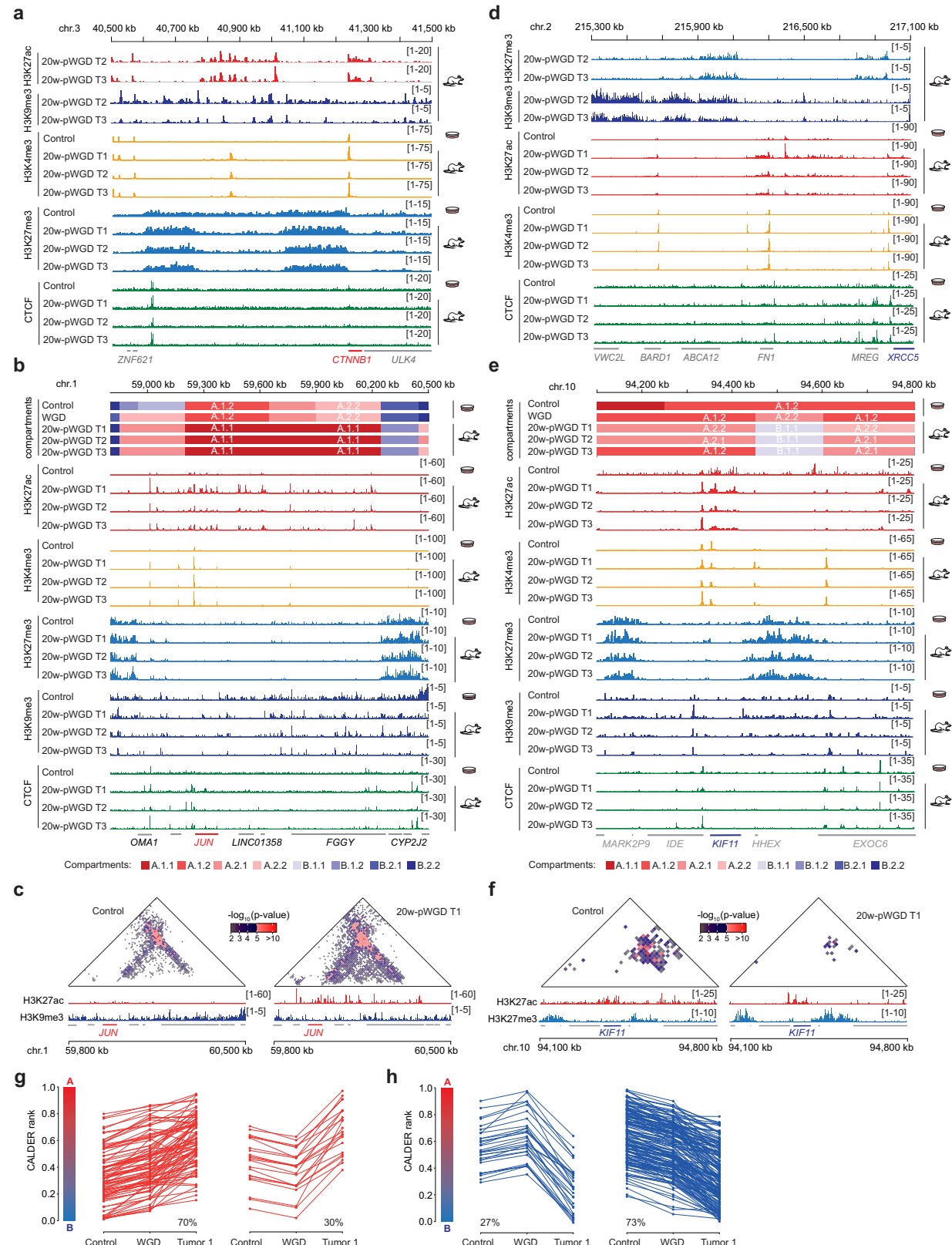

**Extended Data Fig. 11 | Examples of compartment repositioning events associated with gene expression and epigenetic changes. a, d**, Detailed characterization of histone modification and CTCF changes in the indicated genomic region in RPE$^{TP53-/-}$ control and tumour samples. **b, e**, Detailed characterization of the compartment and histone modification and CTCF changes in the indicated genomic region in RPE$^{TP53-/-}$ control and tumour samples. Top: sub-compartment assignments inferred by Calder. Bottom:

histone mark intensities **c, f**, Significant interactions within the indicated regions in RPE$^{TP53-/-}$ control and tumour 1 samples. Histone mark intensities for the corresponding sample are shown at the bottom. **g,h**, Percentage of activating (*red*, consistent n = 91, inconsistent n = 22) and inactivating (*blue*, consistent n = 170, inconsistent n = 30) CoREs that were detected by CBS and that could be traced back to compartment changes occurring already in WGD cells.

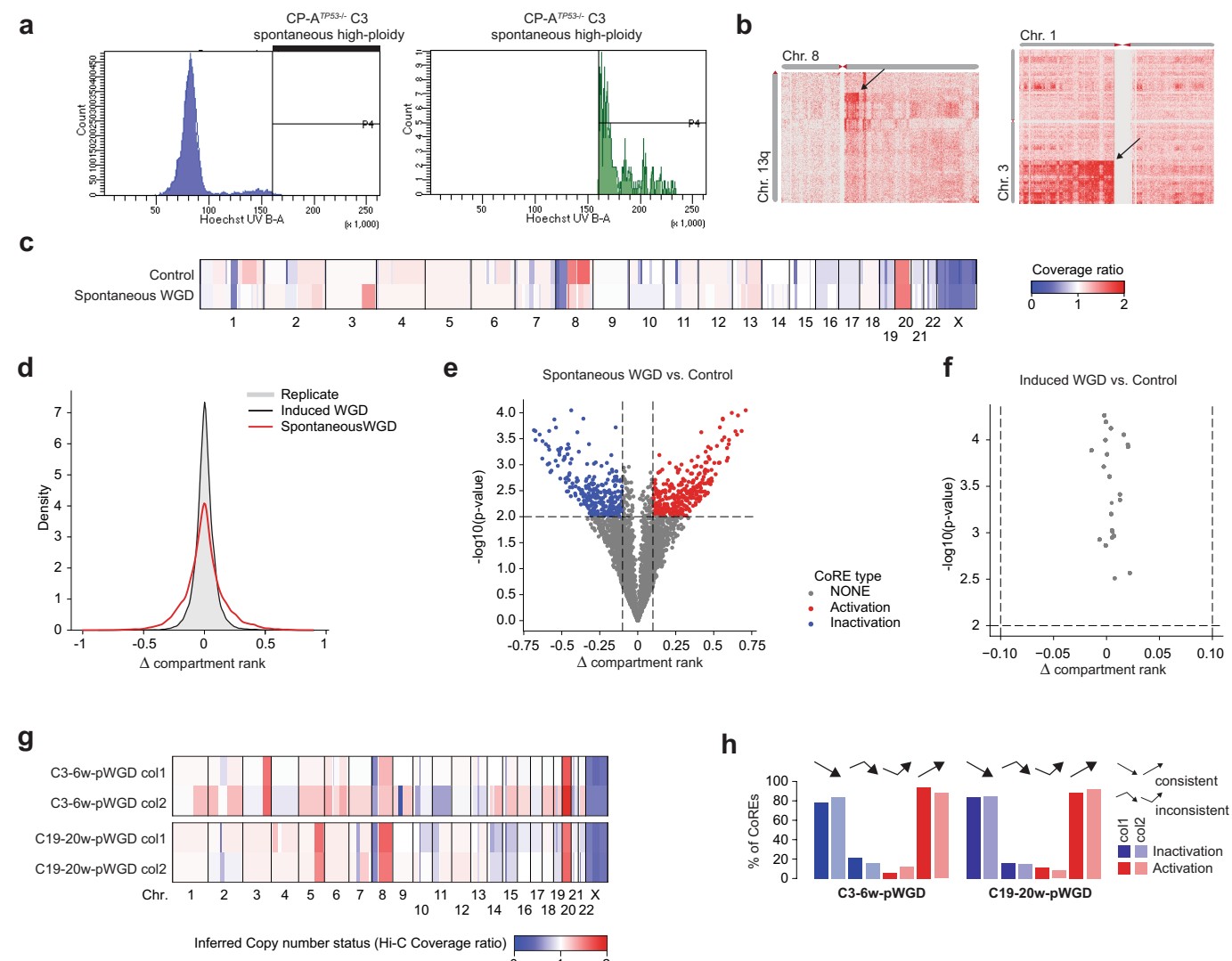

**Extended Data Fig. 12 | CP-A$^{TP53-/-}$ cells acquiring spontaneous high ploidy show chromosomal instability and significant compartment repositioning. a**, Flow-cytometry based sorting of the spontaneous high-ploidy cells from the CP-A$^{TP53-/-}$ population. **b**, Inter-chromosomal Hi-C maps of spontaneous high-ploidy CP-A$^{TP53-/-}$ cells show contact patterns consistent with chromosomal translocations (arrows). **c**, Relative copy number profile of control and spontaneous high-ploidy CPA$^{TP53-/-}$ cells (red: gains, blue: losses). **d**, Density distribution of the difference (delta) of compartment ranks of matching genomic bins between of spontaneous high-ploidy and control cell

(red), induced WGD and control cells (black), and replicates of control cells (gray). **e,f**, Volcano plots showing the delta compartment rank values for candidate CoREs (x-axis) and associated p-values (y-axis) obtained from the comparison of spontaneous high-ploidy and control cells (**e**) and induced WGD and control cells (**f**). Significant CoREs are in red and blue, non-significant are in gray. **e-f** p-values calculated by DiffComp. **g**, Copy number profiles for each colony isolated from soft agar for CP-A$^{TP53-/-}$ samples. **h**, Percentage of consistent and inconsistent inactivating (blue) and activating (red) CoREs for each colony.

Giovanni Ciriello

# Reporting Summary

## Statistics

For all statistical analyses, confirm that the following items are present in the figure legend, table legend, main text, or Methods section.

| n/a | Confirmed | |
|---|---|---|
| ☐ | ☒ | The exact sample size (*n*) for each experimental group/condition, given as a discrete number and unit of measurement |
| ☐ | ☒ | A statement on whether measurements were taken from distinct samples or whether the same sample was measured repeatedly |
| ☐ | ☒ | The statistical test(s) used AND whether they are one- or two-sided *Only common tests should be described solely by name; describe more complex techniques in the Methods section.* |
| ☒ | ☐ | A description of all covariates tested |
| ☐ | ☒ | A description of any assumptions or corrections, such as tests of normality and adjustment for multiple comparisons |
| ☐ | ☒ | A full description of the statistical parameters including central tendency (e.g. means) or other basic estimates (e.g. regression coefficient) AND variation (e.g. standard deviation) or associated estimates of uncertainty (e.g. confidence intervals) |
| ☐ | ☒ | For null hypothesis testing, the test statistic (e.g. *F*, *t*, *r*) with confidence intervals, effect sizes, degrees of freedom and *P* value noted *Give P values as exact values whenever suitable.* |
| ☒ | ☐ | For Bayesian analysis, information on the choice of priors and Markov chain Monte Carlo settings |
| ☒ | ☐ | For hierarchical and complex designs, identification of the appropriate level for tests and full reporting of outcomes |
| ☐ | ☒ | Estimates of effect sizes (e.g. Cohen's *d*, Pearson's *r*), indicating how they were calculated |

*Our web collection on statistics for biologists contains articles on many of the points above.*

## Software and code

Policy information about availability of computer code

| Data collection | Next generation sequencing was performed on Illumina Hi-SeqX, Illumina NextSeq 500, Illumina NovaSeq 6000, or Illumina HiSeq 4000. Single cell RNA sequencing was performed on 10X Genomics Chromium. Flow cytometry was performed on Guava easyCyte™ (Luminex) and Galios (BeckmanCoulter) cytometers. Microscopy imaging was performed on a Zeiss Axioplan upright microscope. Western blots were imaged with the Odyssey CLx Imaging System (LICOR) or FUSION FX6 EDGE Imaging System (Witec). |
|---|---|
| Data analysis | Analysis code to reproduce the results presented in the manuscript is available at https://github.com/CSOgroup/WGD

The following software has been used for data processing and analysis:

- STAR, version 2.7.10a
- RSEM, version 1.3.1
- DESeq2, version 1.34.0
- MAST, version 1.18.0
- Mutect2, version 4.2.2.0
- gnomAD, version 2 (b37)
- FlowJo v10.8
- Fiji v2.9.0
- Python, version 3.9.0
- R, version 4.1.0
- bwa, version 0.7.17 (alignement of reads)
- samtools, version 1.10 (analysis of reads)
- Juicer 1.6 (processing of Hi-C reads)
- hicrep.py, version 0.2.6 (calculation of the SCC score) |

- FANC, version 0.9.21 (Hi-C insulation score and boundaries)
- Freebayes, version v1.3.2-46-g2c1e395-dirty (identification of SNVs from Hi-C reads for phasing)
- SHAPEIT2, version v2.904.3.10.0-693.11.6.el7.x86_64 (population-based phasing)
- HapCUT2, version 1.3.3 (second round of phasing)
- scHiCExplorer, version 7 (analysis of Single-cell Hi-C reads)
- hicexplorer, version 3.5.1 (analysis of Hi-C data)
- cooler, version 0.8.11 (analysis of Hi-C data)
- cooltools, version 0.5.0 (analysis of Hi-C data)
- cell-ranger, version 3.1.1 (analysis of scRNAseq data)
- Seurat, version v3.1.5 (analysis of scRNAseq data)
- InferCNV, version v1.1.0 (analysis of scRNAseq data)
- fastcluster, version v1.2.3 (clustering)
- matplotlib, version v3.4.2 (plotting)
- seaborn, version 0.11.2 (plotting)
- pyscenic, version 0.11.2 (analysis of scRNAseq data)
- GATK, version v4.2.2.0 (marking of duplicates, reads analysis)
- Control-FREEC, version 11.6 (CNV calling)
- bedtools, version v2.30.0 (bed file management)
- vcf2maf, version 1.6.21 (conversion from vcf to maf)
- nfcore/chipseq v.1.2.2 (ChIP-seq reads processing)
- deeptools, version 3.5.1 (analysis of ChIP-seq data)
- MACS3, version 3.0.0a6 (calling peaks from ChIP-seq data)
- nfcore/rnaseq v.3.8 (RNA-seq reads processing)
- HiC-DC, downloaded from https://bitbucket.org/leslielab/hic-dc/src/master/

For manuscripts utilizing custom algorithms or software that are central to the research but not yet described in published literature, software must be made available to editors and reviewers. We strongly encourage code deposition in a community repository (e.g. GitHub). See the Nature Portfolio guidelines for submitting code & software for further information.

# Data

Policy information about availability of data

All manuscripts must include a data availability statement. This statement should provide the following information, where applicable:

- Accession codes, unique identifiers, or web links for publicly available datasets
- A description of any restrictions on data availability
- For clinical datasets or third party data, please ensure that the statement adheres to our policy

Reference genome hg19 was downloaded from http://hgdownload.cse.ucsc.edu/goldenpath/hg19/chromosomes/.
Reference genome human_g1k_hs37d5 was downloaded from http://ftp.1000genomes.ebi.ac.uk/vol1/ftp/technical/reference/phase2_reference_assembly_sequence/
The gnomAD database was downloaded from https://storage.googleapis.com/gatk-best-practices/somatic-b37/af-only-gnomad.raw.sites.vcf.
The dbSNP database was dowloaded from https://ftp.ncbi.nlm.nih.gov/snp/organisms/human_9606/VCF/00-common_all.vcf.gz
Cellranger reference data were downloaded and processed as reported at https://support.10xgenomics.com/single-cell-gene-expression/software/release-notes/build#hg19_3.0.0.
Hi-C data of the GM12878 cell line were downloaded from GSE63525

Processed Hi-C data together with compartment domain calls by Calder are available at https://doi.org/10.5281/zenodo.7351767.
ChIP-seq data, single-cell RNA-seq matrices and copy number profiles of RPE TP53-/- samples are available at https://doi.org/10.5281/zenodo.7351776.
Raw files are deposited at: GSE222390

Data available:
Hi-C (includes Hi-C maps and following features: compartment domains, insulation boundaries, insulation scores):
CP-A control
CP-A WGD
CP-A TP53-/- clone 19 control
CP-A TP53-/- clone 19 WGD
CP-A TP53-/- clone 19 post-WGD colony 1
CP-A TP53-/- clone 19 post-WGD colony 2
CP-A TP53-/- clone 3 control
CP-A TP53-/- clone 3 WGD
CP-A TP53-/- clone 3 WGD + CDK4/6i
CP-A TP53-/- clone 3 post-WGD colony 1
CP-A TP53-/- clone 3 post-WGD colony 2
CP-A TP53-/- clone 3 spontaneous high-ploidy
RPE
RPE TP53-/- control
RPE TP53-/- WGD R1
RPE TP53-/- WGD R2
RPE TP53-/- WGD (CDK1i+DCB)
RPE TP53-/- CIN-only
RPE TP53-/- 20wk Tumor 1
RPE TP53-/- 20wk Tumor 2
RPE TP53-/- 20wk Tumor 3
K562 Control
K562 WGD

ChIP-Seq (for each of the following targets: CTCF, H3K27ac, H3K27me3, H3K9me3, H3K4me3):
RPE TP53-/- Control
RPE TP53-/- 20wk Tumor 1
RPE TP53-/- 20wk Tumor 2
RPE TP53-/- 20wk Tumor 3

ChIP-Seq (for CTCF and H3K9me3):
RPE TP53-/- WGD
CP-A TP53-/- clone 3 Control
CP-A TP53-/- clone 3 WGD

WGS (CNVs):
merged file for RPE TP53-/- control, WGD, 20wk Tumor 1, 20wk Tumor 2, 20wk Tumor 3

WGS (mutations):
merged files (all mutations and filtered mutations) for RPE TP53-/- control, WGD, 6wk in vitro, 20wk in vitro, 20wk Tumor 1, 20wk Tumor 2, 20wk Tumor 3

scRNASeq (differential expression files, gene expression matrix (cell x gene), infercnv data):
RPE TP53-/- control
RPE TP53-/- 6wk in vitro
RPE TP53-/- 20wk in vitro
RPE TP53-/- 20wk Tumor 1
RPE TP53-/- 20wk Tumor 2
RPE TP53-/- 20wk Tumor 3

RNA-Seq:
RPE TP53-/- control (3 replicates)
RPE TP53-/- WGD (3 replicates)

# Field-specific reporting

Please select the one below that is the best fit for your research. If you are not sure, read the appropriate sections before making your selection.

☒ Life sciences        ☐ Behavioural & social sciences        ☐ Ecological, evolutionary & environmental sciences

For a reference copy of the document with all sections, see nature.com/documents/nr-reporting-summary-flat.pdf

# Life sciences study design

All studies must disclose on these points even when the disclosure is negative.

| Sample size | No power analysis was performed to determine the correct sample size. However, three cell lines of distinct origins and distinct induction methods were used, which is sufficient for validating WGD effects on chromatin. Six to nine animals per group were used for in vivo experiments, which is sufficient for proof of tumorigenesis and downstream analyses, while also respecting 3R principles for animal research. For all other experiments, sample size was chosen in order to provide sufficient statistical power. |
|---|---|
| Data exclusions | No data or sequence were excluded, except for the low pass quality reads. |
| Replication | The number of independent replicate for each experiment is reported in the figure legends and/or in the text. |
| Randomization | No randomization was applied as it was not necessary in this study. |
| Blinding | For sequencing experiments, as well as for immunoblots, comparisons were performed between controls and WGD or pWGD conditions, thus blinding was not possible. For karyotyping and immunofluorescence experiments, control and WGD cells have clear distinctions (increased number of chromosomes or nucleus size), which also impedes blinding. |

# Reporting for specific materials, systems and methods

We require information from authors about some types of materials, experimental systems and methods used in many studies. Here, indicate whether each material, system or method listed is relevant to your study. If you are not sure if a list item applies to your research, read the appropriate section before selecting a response.

## Materials & experimental systems

| n/a | Involved in the study |
|-----|------------------------|
| ☐ | ☒ Antibodies |
| ☐ | ☒ Eukaryotic cell lines |
| ☒ | ☐ Palaeontology and archaeology |
| ☐ | ☒ Animals and other organisms |
| ☒ | ☐ Human research participants |
| ☒ | ☐ Clinical data |
| ☒ | ☐ Dual use research of concern |

## Methods

| n/a | Involved in the study |
|-----|------------------------|
| ☐ | ☒ ChIP-seq |
| ☐ | ☒ Flow cytometry |
| ☒ | ☐ MRI-based neuroimaging |

## Antibodies

| Antibodies used | For immunofluorescence:<br>anti-Pericentrin (Abcam, catalogue no. ab4448), anti-α-Tubulin (Sigma-Aldrich, catalogue no. T6074), anti-Mouse IgG-Alexa Fluor 594 (Thermo Fisher Scientific, catalogue no. A-11005), anti-Rabbit IgG-Alexa Fluor 488 (Thermo Fisher Scientific, catalogue no. A-11034).<br>For immunoblotting:<br>anti-TP53 (Santa Cruz Biotechnology, catalogue no. sc-126), anti-β-Actin (Cell Signaling Technology, catalogue no. 4967), anti-CTCF (Active Motif, catalogue no. 61311), anti-RAD21 (Abcam, catalogue no. ab992), anti-α-Actinin (Cell Signaling Technology, catalogue no. 6487), anti-Tri-Methyl-Histone H3 (Lys9) (Cell Signaling Technology, catalogue no. 13969), anti-Acetyl-Histone H3 (Lys27) (Cell Signaling Technology, catalogue no. 8173), anti-Tri-Methyl-Histone H3 (Lys27) (Cell Signaling Technology, catalogue no. 9733), anti-Histone H3 (Cell Signaling Technology, catalogue no. 4499), Goat anti-Mouse (LI-COR Biosciences, catalogue no. 926-68070), Goat anti-Rabbit (LI-COR Biosciences, catalogue no. 926-32211), HRP-conjugated Goat Anti-Mouse Antibody (Merck, catalogue no. AP308P), Goat Anti-Rabbit Antibody (Merck, catalogue no. AP307P).<br>For ChIP-Seq:<br>anti-Acetyl-Histone H3 (Lys27) (Cell Signaling Technology, catalogue no. 8173), anti-Tri-Methyl-Histone H3 (Lys9) (Cell Signaling Technology, catalogue no. 13969), anti-CTCF (Active Motif, catalogue no. 61311), anti-Tri-Methyl-Histone H3 (Lys4) (Cell Signaling Technology, catalogue no. 9751), Tri-Methyl-Histone H3 (Lys27) (Cell Signaling Technology, catalogue no. 9733). |
|-----------------|------|
| Validation | All antibodies used in this study are commercially available and validated for the application they were used for in this study, as stated on the manufacturer's websites. Additionally, the antibodies were previously validated and used in numerous studies. |

## Eukaryotic cell lines

Policy information about cell lines

| Cell line source(s) | hTERT RPE-1 (originally from ATCC, from the laboratory of Dr. Jan Korbel, EMBL Heidelberg, Germany)<br>hTERT RPE-1 TP53-/- (from the laboratory of Dr. Jan Korbel, EMBL Heidelberg, Germany)<br>CP-A (KR-42421) (ATCC)<br>CP-A TP53-/- (this study)<br>K562 (DSMZ) |
|---------------------|------|
| Authentication | None of the cell lines were authenticated. |
| Mycoplasma contamination | All cell lines tested negative for mycoplasma contamination. |
| Commonly misidentified lines (See ICLAC register) | No commonly misidentified cell lines were used. |

## Animals and other organisms

Policy information about studies involving animals; ARRIVE guidelines recommended for reporting animal research

| Laboratory animals | All animals used in the study were NOD scid gamma (NSG) female mice maintained at the EPFL animal facilities in a 12h-light 12h-dark cycle, at 18-23°C with 40-60% humidity, as recommended and in accordance with the regulations of the Animal Welfare Act (SR 455) and Animal Welfare Ordinance (SR 455.1).<br>Experiment timelines are described in the manuscript. |
|--------------------|------|
| Wild animals | This study did not involve wild animals. |
| Field-collected samples | The study did not involved samples collected from the field. |
| Ethics oversight | Animal experiments were performed in accordance with the Swiss Federal Veterinary Office guidelines and as authorized by the Cantonal Veterinary Office (animal license VD2932.1). |

Note that full information on the approval of the study protocol must also be provided in the manuscript.

# ChIP-seq

## Data deposition

☒ Confirm that both raw and final processed data have been deposited in a public database such as GEO.

☒ Confirm that you have deposited or provided access to graph files (e.g. BED files) for the called peaks.

**Data access links**
*May remain private before publication.*

GSE222390
https://doi.org/10.5281/zenodo.7351767
https://doi.org/10.5281/zenodo.7351776

**Files in database submission**

Bigwig files of ChIP-seq experiments generated in this study. Files are log2 input normalized tracks. The "bw_signals_attributes.txt" file can be used to add metadata to the tracks for visualization in IGV.

Files available:

RPE_TP53_Ctrl_H3K27ac_vs_RPE_TP53_Ctrl_input.bigWig
RPE_TP53_Ctrl_H3K27me3_vs_RPE_TP53_Ctrl_input.bigWig
RPE_TP53_Ctrl_H3K4me3_vs_RPE_TP53_Ctrl_input.bigWig
RPE_TP53_Ctrl_H3K9me3_vs_RPE_TP53_Ctrl_input.bigWig
RPE_TP53_Ctrl_CTCF_vs_RPE_TP53_Ctrl_input.bigWig

RPE_TP53_WGD_H3K9me3_vs_RPE_TP53_WGD_input.bigWig
RPE_TP53_WGD_CTCF_vs_RPE_TP53_WGD_input.bigWig

RPE_TP53_20w0T1_CTCF_vs_RPE_TP53_20w0T1_input.bigWig
RPE_TP53_20w0T1_H3K27ac_vs_RPE_TP53_20w0T1_input.bigWig
RPE_TP53_20w0T1_H3K27me3_vs_RPE_TP53_20w0T1_input.bigWig
RPE_TP53_20w0T1_H3K4me3_vs_RPE_TP53_20w0T1_input.bigWig
RPE_TP53_20w0T1_H3K9me3_vs_RPE_TP53_20w0T1_input.bigWig

RPE_TP53_20w0T2_CTCF_vs_RPE_TP53_20w0T2_input.bigWig
RPE_TP53_20w0T2_H3K27ac_vs_RPE_TP53_20w0T2_input.bigWig
RPE_TP53_20w0T2_H3K27me3_vs_RPE_TP53_20w0T2_input.bigWig
RPE_TP53_20w0T2_H3K4me3_vs_RPE_TP53_20w0T2_input.bigWig
RPE_TP53_20w0T2_H3K9me3_vs_RPE_TP53_20w0T2_input.bigWig

RPE_TP53_20w0T3_CTCF_vs_RPE_TP53_20w0T3_input.bigWig
RPE_TP53_20w0T3_H3K27ac_vs_RPE_TP53_20w0T3_input.bigWig
RPE_TP53_20w0T3_H3K27me3_vs_RPE_TP53_20w0T3_input.bigWig
RPE_TP53_20w0T3_H3K4me3_vs_RPE_TP53_20w0T3_input.bigWig
RPE_TP53_20w0T3_H3K9me3_vs_RPE_TP53_20w0T3_input.bigWig

CPA-TP53-Clone3_Ctrl_CTCF_vs_CPA-TP53-Clone3_Ctrl_input.bigWig
CPA-TP53-Clone3_Ctrl_H3K9me3_vs_CPA-TP53-Clone3_Ctrl_input.bigWig
CPA-TP53-Clone3_WGD_CTCF_vs_CPA-TP53-Clone3_WGD_input.bigWig
CPA-TP53-Clone3_WGD_H3K9me3_vs_CPA-TP53-Clone3_WGD_input.bigWig

**Genome browser session**
(e.g. UCSC)

No longer applicable.

## Methodology

**Replicates**

One replicate for each condition.

**Sequencing depth**

Total paired-end reads after trimming, alignment, and PCR duplicates removal for samples generated for this study:

RPE_TP53_Ctrl_CTCF: 90,500,994
RPE_TP53_Ctrl_H3K27ac: 64,342,884
RPE_TP53_Ctrl_H3K27me3: 76,116,438
RPE_TP53_Ctrl_H3K4me3: 69,075,642
RPE_TP53_Ctrl_H3K9me3: 70,791,476

RPE_TP53_WGD_CTCF: 78,708,974
RPE_TP53_WGD_H3K9me3: 40,490,233

RPE_TP53_20w0T1_CTCF: 82,792,296
RPE_TP53_20w0T1_H3K27ac: 81,453,282
RPE_TP53_20w0T1_H3K27me3: 82,291,046
RPE_TP53_20w0T1_H3K4me3: 61,718,866
RPE_TP53_20w0T1_H3K9me3: 85,119,828

RPE_TP53_20w0T2_CTCF: 76,721,118

nature portfolio | reporting summary

RPE_TP53_20w0T2_H3K27ac: 107,571,008
RPE_TP53_20w0T2_H3K27me3: 89,503,348
RPE_TP53_20w0T2_H3K4me3: 80,108,224
RPE_TP53_20w0T2_H3K9me3: 75,368,694

RPE_TP53_20w0T3_CTCF: 72,833,790
RPE_TP53_20w0T3_H3K27ac: 78,101,386
RPE_TP53_20w0T3_H3K27me3: 92,531,496
RPE_TP53_20w0T3_H3K4me3: 86,117,812
RPE_TP53_20w0T3_H3K9me3: 94,177,908

CPA-TP53-Clone3_Ctrl_CTCF: 29,735,520
CPA-TP53-Clone3_Ctrl_H3K9me3: 29,798,985
CPA-TP53-Clone3_WGD_CTCF: 29,683,395
CPA-TP53-Clone3_WGD_H3K9me3: 23,051,295

**Antibodies**

anti-Acetyl-Histone H3 (Lys27) (Cell Signaling Technology, catalogue no. 8173), anti-Tri-Methyl-Histone H3 (Lys9) (Cell Signaling Technology, catalogue no. 13969), anti-CTCF (Active Motif, catalogue no. 61311), anti-Tri-Methyl-Histone H3 (Lys4) (Cell Signaling Technology, catalogue no. 9751), Tri-Methyl-Histone H3 (Lys27) (Cell Signaling Technology, catalogue no. 9733). All antibodies were used at the concentrations recommended by the manufacturer.

**Peak calling parameters**

Peak calling for CTCF was done with MACS3 callpeak, version 3.0.0a6 with the following parameters:
--format BAMPE
--gsize hs
--nomodel
and the rest of the parameters left as default.

**Data quality**

- PCR duplicates were removed, FDR threshold of 0.05, minimum fold-change for a peak was set to 5

**Software**

- nfcore/chipseq pipeline 1.2.2
- deeptools, version 3.5.1
- MACS3, version 3.0.0a6

# Flow Cytometry

## Plots

Confirm that:

☒ The axis labels state the marker and fluorochrome used (e.g. CD4-FITC).

☒ The axis scales are clearly visible. Include numbers along axes only for bottom left plot of group (a 'group' is an analysis of identical markers).

☒ All plots are contour plots with outliers or pseudocolor plots.

☒ A numerical value for number of cells or percentage (with statistics) is provided.

## Methodology

**Sample preparation**

Cells were collected and washed with PBS (Thermo Fisher Scientific, catalogue no. 10010023). Permeabilization was performed in 0.01% Triton-X100 (AppliChem, catalogue no. A1388) in PBS for 30-60 minutes at 4°C. Following PBS washes, the cells were fixed and stained with FxCycle™ PI/RNase Staining Solution (Thermo Fisher Scientific, catalogue no. F10797) overnight at 4°C in the absence of light.

**Instrument**

Guava® easyCyte™ (Luminex) and Galios (Beckman Coulter)

**Software**

FlowJo (BD).

**Cell population abundance**

All cells in the population were queried for PI-based DNA content.

**Gating strategy**

FSC/SSC used for exclusion of debris and FSC-H/FSC-W for the exclusion of doublets. No other gating strategy was necessary.

☒ Tick this box to confirm that a figure exemplifying the gating strategy is provided in the Supplementary Information.

