## [Peer Review File · Nature]

Manuscript Title: Whole genome doubling drives oncogenic loss of chromatin segregation

Reviewer Comments & Author Rebuttals

Reviewer Reports on the Initial Version:

Referee expertise:

Referee #1: chromatin, computational expertise

Referee #2: genomics, evolution, WGD

Referee #3: WGD, aneuploidy

Referee #4: chromatin, Hi-C

Referees' comments:

Referee #1 (Remarks to the Author):

This manuscript explores the 3D conformation changes that accompany whole genome duplication. Using an impressive array of genomic assays, the authors explore WGD in three different systems. They show that WGD is associated with systematic changes in the strength of compartments and domains, and that this phenomenon arises in a subset of cells. They also explore the effects of WGD on subsequent evolution of the cells, showing that WGD is associated with repositioning of some regions of the chromosome to different subcompartments. I found the evidence showing that WGD leads to chromosomal instability and complex re-arrangements particularly convincing. Overall, the analysis is generally accompanied by well described and rigorous statistical analysis to back up each primary claim.

Major

There seems to be conceptual problem with the way the manuscript talks about tetraploid genome structure. For example, in line 74, it's not clear what it would be mean for tetraploid cells to "faithfully reconstitute the chromatin 3D organization of their diploid counterparts." Similarly, at lines 102-103, I think it simply doesn't makes sense to say that "The overall chromatin 3D organization in control and WGD cells remained largely unchanged." Doubling the number of chromosomes by definition is a huge change. What is surprising is that, despite the doubling, many aspects of the 3D architecture in WGD cells are similar to those of diploid cells, including the aspects listed in this sentence. In particular, all of the aspects measured in Suppl Fig 2 relate to cis contacts. The text should make this clear. For example, one could say "the overall *intra-chromosomal* chromatin 3D organization was largely preserved."

An important, unacknowledged confounder in the analysis of intra-chromosomal contacts is the presence of contacts between homologous chromosomes. These are trans contacts that will be interpreted as cis contacts, and their rate is expected to increase by a factor of three when changing from diploid to tetraploid (each chromosome has one homolog or three homologs). If there is significant homolog pairing, then the rate of such contacts will be increased. This is relevant, e.g., to the discussion in lines 131-141.

I can't make any sense of ED Figure 6a. Why are we interested in the Pearson correlation of O/E between different chromosomes? Indeed, how is that even computed, since the chromosomes are of different lengths? I feel like I'm missing something fundamental here. In general, why is Pearson of O/E used, rather than SCC as in ED Fig 2a? To me, SCC seems better.

The observation in Figure 3b is striking, but I'm not sure I buy the argument that this phenomenon indicates "that these cells intrinsically exhibited LCS." As mentioned earlier, the changes in interaction frequencies between long/short chromosomes could best be explained as resulting from the change in the number of homologous copies of each chromosome. Indeed, what I don't understand are the cells that seem to not exhibit such changes. I would suggest that the authors adopt a more precise terminology than "LCS" and "cells w/o inter-chr LCS" to refer to these two groups.

I was confused by the summarization of results on lines 266-267: "Overall, these results indicate that LCS observed at the population level is an intrinsic feature of individual cells that underwent WGD." The hypothesis was framed in Figure 3a as a distinction between "intrinsic" and "emergent." Doesn't Figure 3b clearly show that there are two classes of cells, and subsequent analysis support this idea? The analysis in lines 263-265 suggests that one of those groups has compartment structure that is statistically indistinguishable from the control cells. I thus thought that we were clearly in the "emergent" scenario in Figure 3a, with two distinct groups of cells. Why is the conclusion that this is "intrinsic"? Maybe it's just that the figure is misleading. I think the data shows that a subset of the cells look like wild type and a subset show significant changes in both long-vs-short chromosome contacts and in compartment structure. The text (and figure) should do a better job of conveying this. The choice of terminology (emergent versus intrinsic) should be reconsidered.

The algorithm for segmenting the delta compartment vectors (line 816-825) is, on the face of it, reasonable but pretty arbitrary. This is an example of a changepoint detection problem, for which there are many existing algorithms (including standard libraries in Python and R). I think the authors should demonstrate that their results do not depend on the particulars of this new changepoint detection algorithm by recapitulating at least some of their results using a commonly used changepoint detection algorithm.

Minor

Line 25: I don't understand what it means to "upscale protein production of the chromatin architectural protein CTCF." Grammatically, this is incorrect, but even if we interpret the phrase to mean that the cells are failing to increase CTCF production, it's not clear in what context this increase is expected to happen.

Line 41: "imputed" is the wrong word here.

Line 70: It seems to me that this should be the start of a new paragraph.

Line 78-80: Grammatically, I couldn't make sense of this sentence.

In Extended Data Fig 2, "CALDER" has not yet been defined. Give the citation here.

Line 121: In the caption to the Extended Data Figure 3, one sentence refers to inter-chromosomal contacts and the subsequent one refers to intra-chromosomal contacts. Which is correct? "Inter-" makes no sense (The O/E transformation is intended to correct for the genomic distance effect, which is a property of cis contacts), so I am guessing it is supposed to be "Intra-." I don't understand why lower O/E values imply that "chromatin structural elements were less segregated."

Line 211-213: This observation -- that contacts between long and short chromosomes increase even p53 wild type WGD cells -- does not seem surprising to me, and emphasizes what was already mentioned above (line 143-144) that the increased contact frequency is simply explained by the doubled number of homologous chromosomes.

Line 218-221: This distinction is now clear, but the mention of it in the abstract was not.

Line 224: I think, somewhere around here, you should mention the mean number of obtained contacts per cell. I would like to see a supplementary table or figure that shows what this distribution looks like.

Line 250-257: I did not find Figure 3d or the associated text very informative. I'm not sure what this analysis tells us. On the other hand, Figure 3e is quite compelling.

I did not find ED Fig 9c convincing. I am not sure what message we are supposed to glean from this figure. And ED Fig 9d is not well described. What is this a distribution over? I understand that the x-axis in each plot is correlation, but I can't figure out what the vertical axis is a count/density over.

Line 400-401: How were the 6000 regions selected for testing?

Supplement:

Lines 55-56: It seems a bit funny to critique the standard method for not providing a quantification of domain strength. An easy way to do this is to examine the eigenvalues associated with each compartment region. But the authors are not even obliged to make this argument here, since they are using a previously described algorithm (Calder) to get a continuous value for each compartment region.

In general, I don't see why these two supplementary notes are necessary. If the authors think this information is critical, please just include it (briefly) within Methods. I think the first note, in

particular, is not necessary at all.

Referee #2 (Remarks to the Author):

Lambuta et al., explore the impact of WGD in cancer with regards to 3D organization of chromatin and its contribution to oncogenic phenotypes.

This is a very interesting and thorough manuscript. It is dense and contains a lot of information. Overall, I think it is well written and clear. However, there are a few areas where additional clarity could improve the manuscript (particularly towards the end). I have two main concerns (outlined in more detail below): 1. Are the results a result of WGD rather than CIN? 2. Are the same results observed in the context of naturally occurring WGD (rather than induced WGD)?

Comments:

- The authors explore WGD in three distinct cellular models. However, in all cases they induce WGD through treatment with nocodazole. I think it would be beneficial to also consider naturally occurring cell-lines with WGD, to the nocodazole treated cell-lines can be considered equivalent. While I appreciate this would not permit a direct comparison of before and after treatment, I think this is a crucial analysis to ensure this can be considered equivalent to the WGD observed in tumours.

- Relatedly, the authors could consider a naturally occurring isogenic genome doubling system - in many cell-lines a naturally occurring doubled population can be isolated. This would ensure confidence that this is not related to the treatment.

- Likewise, if the authors induce CIN, but not WGD, is the increase in chromatin segregation observed? Or is it truly specific to WGD? In general, I think this is a critical question throughout the manuscript - how much of what is observed can be linked directly to WGD rather than simply increased instability and aneuploidy?

- Can the authors demonstrate that their results for various experiments (e.g. ChIP-seq, HiC etc) do not relate to differences in coverage per haploid copy. Are the results the same if they effectively down-sample the diploid control?

- Relatedly, are the results preserved if the post-WGD diploid tumours are considered? (i.e. are these differences between WGD and nWGD cells maintained following significant loss of chromosomes).

- The authors state that:

"The overall tumour mutation burden was similar across samples and did not significantly increase in RPETP53^{-/-} cells that underwent WGD compared to RPETP53^{-/-} cells that were kept in culture for the same amount of time (Extended Data Fig. 8c)." Given that there is twice the amount of DNA it seems surprising that there isn't twice the number of mutations - even if the same number of mutations per haploid copy. It would be interesting to explore this further - does this relate to the sequencing coverage? Or, is there a lower mutation rate?

- The authors observe that "tumours from 20-week post-WGD cells exhibited remarkably similar CNV profiles, indicating the selection and expansion of the same clone in vivo, which was characterized by copy number gains of chromosomes 1p, 13q, and 8q, and losses within chromosomes 1q, 4p, 13p, and 18p (Fig. 4d)." To confirm this can the authors evaluate whether indeed the same allele is gained or lost (are there sufficient heterozygous SNPs?), and further, whether there is evidence for the same breakpoint? (i.e. is this certainly selection and expansion of the same clone rather than convergent/parallel evolution?)

- I'm afraid I find Figure 4d and e difficult to follow. I think it would be easier to interpret if the different timepoints were more clearly separated, and a heterogeneity score assessed.

- Given the various comparisons in 'chromatin evolution of WGD cells' it is easy to get lost - the authors are comparing RPE_tp53^{-/-} WGD cells, with 20 week post WGD cells and control (non-WGD cells). But, given there are multiple instances of WGD (in tumour and cell-line), simply describing a cell as WGD and tumour is not terribly informative.

- The authors conclude that 'Overall, our study demonstrates that WGD promotes changes of the chromatin 3D organization contributing to deregulating oncogenes and tumour suppressors, in parallel and largely independently of chromosomal instability.' I think more is needed to demonstrate this is truly independent of WGD

- I wonder if the manuscript could benefit from a cartoon (perhaps in Supplementary) summarising the key findings and how they link together? I think something like this could make the take-home message clearer.

Referee #3 (Remarks to the Author):

In this manuscript, Lambuta & Nanni et al. explore the chromatin organization in cells that have undergone whole-genome doubling (WGD). The authors induced WGD in three human cell lines and performed Hi-C of the matched diploid/tetraploid pairs. They observed reduced segregation of chromatin structural elements in WGD+ cells, increased short-range and long-range interactions between chromosomes and chromatin sub-compartments, and repositioning of sub-compartments that were associated with oncogenic transcriptional changes. The authors propose that these chromatin changes occurred independently of mutations and chromosomal aberrations, and are therefore complementary mechanisms that contribute to tumor formation/evolution following WGD.

Overall, this is an interesting study. There are surprisingly few published papers that have evaluated the effect of ploidy changes (both polyploidy and aneuploidy) on chromatin organization and nuclear architecture. The manuscript presents rigorous experiments, and data analysis also appears to be sound. The novelty and conceptual advance are borderline for publication in Nature in my mind. In

any case, to justify publication in Nature, I believe that more work needs to be done in order to: (a) further link the observed chromatin organization changes to gene expression changes and tumorigenicity; (b) convincingly demonstrate that the effects of WGD are indeed independent of chromosomal instability (CIN); and (c) generalize the findings by using additional methods for WGD induction, another cell line for the in vivo experiments, and (potentially) analyzing available clinical data from cancer patients.

Major comments

Robustness:

(1) The chromatin organization changes were partly dependent on p53 status – p53-WT WGD+ cells neither exhibited enriched contacts between A and B sub-compartments, nor loss of insulation at TAD boundaries, but still exhibited increased contact frequencies between long and short chromosomes. These data suggest that much of the observed loss of chromatin segregation (LCS) depends on p53. However, this comparison was only done in a single cell line, and should be repeated using additional cell lines. In addition, can the authors add a control of p53+/+ vs. p53-/- cells to determine whether p53 can lead (in and of itself) to any changes in chromatin organization? In particular: does p53 inactivation have any effect on CTCF levels and TAD boundaries?

(2) Relatively little data is provided to link the observed changes in chromatin organization to perturbed transcription and altered gene expression. What would be the transcriptional consequence of the increased interactions between short and long chromosomes, increased contacts between A and B sub-compartments, and reduced insulation at TAD boundaries? The authors analyzed the effect of compartment repositioning events (CoREs) on gene expression, but what about the other described chromatin alterations (i.e., the increased chromosome interactions and the perturbed TAD boundaries)? This should be experimentally addressed at least at the population level. The scRNA-seq data that was generated from the cells post-WGD is somewhat under-exploited in this regard – can these data be used to address the effect of loss of chromatin segregation (beyond CoREs) on gene expression? To assess the importance of the observed phenomenon, it would be important to quantify the fraction of gene expression changes in the WGD+ populations that are driven by CNVs vs. the fraction that can be linked to the chromatin changes.

(3) Related to the previous point: the authors show interesting examples of compartment repositioning events associated with gene expression and epigenetic changes. In what fraction of differentially expressed genes between the tumors and the parental cell lines this mechanism can be identified? In other words, is this a common mechanism for activating oncogenic pathways and silencing tumor suppressive pathways?

(4) A key claim of this study is that loss of chromatin segregation and sub-compartment repositioning are CIN-independent mechanisms of WGD-driven tumorigenesis. The authors argue that the evidence for that is that most compartment repositioning events occur in regions unaffected by copy number changes or chromosomal re-arrangements. However, this does not really prove a CIN-independent mechanism. Notably, the magnitude of the observed changes in the

tumors was considerably higher than in the WGD cells from which they were formed (Fig. 5j,k), and at the tumor stage CIN was already prevalent. So it is not clear that this is indeed CIN-independent. To assess whether this is indeed the case, CIN should be inhibited in the WGD+ cells using MCAK overexpression or the inhibitor UMK57, and the effect on chromatin segregation and its associated gene expression changes should be evaluated.

Generalizability:

(5) The authors initially used 3 cell lines (CP-A, RPE1 and K562) for WGD induction. However, WGD was always induced by nocodazole treatment (either with and without subsequent DCB treatment). Additional methods for WGD induction (by mitotic slippage, cytokinesis failure or endoreduplication) must be used to ensure that the observed chromatin changes are indeed associated with WGD and not with the nocodazole treatment.

(6) Along the same lines: the in vivo experiments were done using a single cell line (RPE1). Given their central role in the manuscript as a whole and for some of the key novel claims, these experiments need to be repeated in another cell line.

(7) The WGD status of tumors has been determined in many cancer genomic studies, including TCGA (see Taylor et al. Cancer Cell 2018 for example). Can the authors analyze available Hi-C data from previous studies to further address the generalizability and clinical relevance of the phenomenon that they describe?

Referee #4 (Remarks to the Author):

Comments:

Lambuta et al. present a manuscript studying the impact of whole genome duplications (WGD) on 3D genome organization in the nucleus. In general, this is an understudied problem so the general topic in my mind will be of significant interest. They use three cell line models to induce WGD and examine the effects of genome doubling on nuclear structure. They observe reduced chromatin compartment and TAD separation. This is also accompanied by a reduction in the levels of the CTCF protein, suggesting that this may play a causal role. They then use the WGD cells to generate tumors in mice and examine the genetic and genomic changes that occur in the tumor samples that grow out. They observe several copy number alterations and genetic rearrangements, some of which they say are recurrent. They also observe changes in 3D genome structure, including changes in chromatin compartments, in the tumors compared with the original cell lines. While I find the general idea of the manuscript interesting, there are several aspects that I think are currently underdeveloped. This includes the role of p53 and CTCF in contributing to changes in genome organization during WGD, as well as linking changes observed in chromatin organization in WGD with the final genetic or genomic changes observed in the final tumor samples. Due to these limitations, I don't think I can recommend publication at this time. I have organized my comments into major and minor comments below.

Major comments:

Mechanistically, it is hard to fully accept the changes in 3D genome organization after WGD as being the consequence of reduced levels of CTCF given the current data. Specifically, the authors indicate that the reduction in CTCF levels underlies the changes in compartment strength and TAD insulation seen after WGD. One issue that I see with this is that the loss of compartment strength in CTCF-AID cells (2d) seems to be less than the loss observed in WGD (1e), which to me raises the issue of to what extent these changes are being driven by CTCF. To me a really good control experiment would be to overexpress CTCF in the context of WGD to determine if rescuing CTCF levels would alter the changes in 3D genome structure they observe in WGD.

While the authors show that there are numerous CNVs, rearrangements, and compartment changes that occur in the tumors from mice, it isn't clear from the data how this is related to any of the observed changes in 3D genome organization that occur during WGD. In other words, the link between the first and second half of the manuscript isn't well developed. For example, does the loss of chromatin separation observed in WGD lead to increased likelihood of DNA damage or mitotic errors?

Can the authors rule out that extended Nocodazole + dihydrocytochalasin B induces heritable changes in 3D genome structure independent of WGD? For example, its not apparent in the methods, but do the control cells undergo any kind of similar extended nocodazole treatment?

Very few plots show the actual Hi-C contact maps, and most instead show aggregated signals. I think it is important to show at least some examples to give the reader a better idea of the magnitude of the effects.

Minor points:

They say that the WGD occurs due to "mitotic slippage" in the introduction. I'm not familiar with this term. Can they be more precise in describing what actually occurs in "mitotic slippage".

I think that to really distinguish the reduced CTCF binding in Figure 2b, they need to do some kind of spike in control.

Although the mechanism is different, during the cell cycle cells naturally double their DNA content, albeit temporarily and with sister chromatid cohesion. I believe there are existing Hi-C data in G2 synchronized cell publicly available. Is there any similar loss of 3D genome segmentation in G2 cells? This might be a nice control to evaluate whether simply doubling DNA content in the nucleus can have a similar effect.

From the figures (Figure 4 and Ex. Fig. 8), it is hard to tell which tumors are derived from independent WGD experiments and which are from the same. Seemingly, the most interesting cases of CNV or other rearrangements would be events that are recurrent across independent WGD experiments. The authors mention they identify such events but don't really go into detail about what they are or what role they may be playing in contributing to tumorigenesis.

The observation that CTCF is reduced in WGD cells but only in a p53 null background is pretty interesting, but it really begs the question of how this is regulated. Is CTCF a direct p53 target, and is it being regulated somehow in a stress responsive way (i.e. do non-WGD cells show changes in CTCF levels in response to p53 loss?).

Author Rebuttals to Initial Comments:

Referee expertise:

Referee #1: chromatin, computational expertise

Referee #2: genomics, evolution, WGD

Referee #3: WGD, aneuploidy

Referee #4: chromatin, Hi-C

We thank all four reviewers for finding the results reported in our study interesting and supported by robust experiments. We have appreciated their suggestions to further corroborate our key findings and in the revised version of the manuscript, we have included several new experiments and analyses.

Briefly, in this study, we show that WGD in p53^{-/-} cells results in loss of chromatin segregation (LCS), defined by increased contacts between normally well segregated structural elements of the chromatin such as TADs and compartments. To further elucidate the mechanisms underlying this phenotype and the connection between LCS and regions undergoing significant compartment repositioning in tumors, we have included several new experiments addressing the reviewers' questions. Here, we would like to summarize the major experiments and findings introduced in this revised version of the manuscript. A more detailed description is provided in response to the specific comments of the reviewers.

1. *First, we have expanded our analyses to investigate which factors regulate LCS.* In two independent TP53^{-/-} models, we now show that WGD cells exhibit reduced levels of **CTCF**, which is required for boundary insulation, and **H3K9me3**, which drives heterochromatin compaction and chromatin compartmentalization (see new **Figure 2**). Expression of other chromatin modifiers and histone marks were either unchanged or only moderately altered. Interestingly, loss of compartment segregation was most evident in the B.2.2 compartment, which was enriched for H3K9me3 and exhibited the greatest loss of this histone mark in WGD cells (see new **Figure 2j** and **ED Figure 5c,d**). Importantly, this effect is specific of TP53^{-/-} cells, and it is not observed in their TP53 wild-type counterparts (**Figure 2**). Moreover, we could rescue CTCF and H3K9me3 levels by stalling the cells in G1, and, with restored protein levels, cells no longer exhibited LCS. These results indicated that WGD cells failed to upscale the protein levels of key chromatin regulators, which resulted in LCS. Notably, these results are in line with recent findings showing that tetraploid cells exhibit reduced levels of proteins involved in DNA replication thus increasing DNA damage and chromosomal instability (CIN) (Gemble et al., 2022). Hence, in WGD cells, shortage of distinct sets of proteins seems to be the underlying cause of CIN and LCS, with both contributing to cell malignant transformation.

2. *Next, we have clarified the relationship between LCS and CIN in WGD cells.* We have induced CIN without WGD in RPE^{TP53^{-/-}} and analyzed chromatin architecture in these cells. In this condition, we did not detect loss of insulation at TAD boundaries or loss of segregation of chromatin compartments (see the updated **Figure 1g,h**). Moreover, we have now included a series of new analyses to support the conclusion that CIN alone does not induce LCS, including CNV inference in single cells (new **Figure 3e**) and systematic comparisons of copy number

variants and compartment repositioning events (new **Figure 5c-f**). As mentioned above, WGD induces both CIN and LCS and both contribute to cell malignant transformation.

3. Lastly, we have introduced a new tumorigenic model using *CP-A^{TP53-/-}* cells post-WGD to link LCS and CIN with the acquisition of oncogenic traits and validate our results on tumors derived from *RPE^{TP53-/-}* cells. We performed Hi-C analyses of 4 independent soft agar colonies formed from 2 independent *CP-A^{TP53-/-}* clones that underwent WGD. In these colonies, we confirmed the acquisition of copy number alterations and chromatin compartment repositioning. Most importantly, compartment repositioning events in *CP-A^{TP53-/-}* derived soft agar colonies could be traced back to chromatin changes occurring in WGD cells, in two independent WGD experiments (new **Figure 6**). These results not only validate our findings in an independent model but further support the link between loss of chromatin segregation in WGD cells and compartment repositioning occurring in tumors.

We have discussed comprehensively all the new data and analyses in the point-by-point response below. We hope that the reviewers will find this study compelling and ready to be published.

Referees' comments:

Referee #1 (Remarks to the Author):

This manuscript explores the 3D conformation changes that accompany whole genome duplication. Using an impressive array of genomic assays, the authors explore WGD in three different systems. They show that WGD is associated with systematic changes in the strength of compartments and domains, and that this phenomenon arises in a subset of cells. They also explore the effects of WGD on subsequent evolution of the cells, showing that WGD is associated with repositioning of some regions of the chromosome to different subcompartments. I found the evidence showing that WGD leads to chromosomal instability and complex re-arrangements particularly convincing. Overall, the analysis is generally accompanied by well described and rigorous statistical analysis to back up each primary claim.

We thank the reviewer for the overall positive assessment of our work and below we have responded to their comments point-by-point.

Major

There seems to be conceptual problem with the way the manuscript talks about tetraploid genome structure. For example, in line 74, it's not clear what it would be mean for tetraploid cells to "faithfully reconstitute the chromatin 3D organization of their diploid counterparts." Similarly, at lines 102-103, I think it simply doesn't makes sense to say that "The overall chromatin 3D organization in control and WGD cells remained largely unchanged." Doubling the number of chromosomes by definition is a huge change. What is surprising is that, despite the doubling, many aspects of the 3D architecture in WGD cells are similar to those of diploid cells, including the aspects listed in this sentence. In particular, all of the aspects measured in

Suppl Fig 2 relate to cis contacts. The text should make this clear. For example, one could say "the overall *intra-chromosomal* chromatin 3D organization was largely preserved."

We thank the reviewer for this suggestion, and we amended these sentences, which helped us clarify that the overall organization of intra-chromosomal structures is maintained even after the entire genome has been duplicated. Moreover, we added one additional comparison in the supplementary figure to specifically compare inter-chromosomal contacts in these experiments. This new analysis further confirmed a similar degree of contacts similarity between diploid and tetraploid cells and between replicates of the same condition (see new **ED Figure 2b**).

An important, unacknowledged confounder in the analysis of intra-chromosomal contacts is the presence of contacts between homologous chromosomes. These are trans contacts that will be interpreted as cis contacts, and their rate is expected to increase by a factor of three when changing from diploid to tetraploid (each chromosome has one homolog or three homologs). If there is significant homolog pairing, then the rate of such contacts will be increased. This is relevant, e.g., to the discussion in lines 131-141.

The reviewer is correct and, in fact, the rates of both *in cis* and *in trans* contacts are expected to increase. Precisely, putative "*in cis*" contacts should increase by a factor of 3 upon WGD and "*in trans*" contacts should increase by a factor of 4. Hence, we should expect that the ratio of in trans (T) vs. in cis (C) contacts increases upon WGD as described below:

$$r_{control} = \frac{T}{2C} ; \quad r_{WGD} = \frac{4T}{6C} ; \quad \frac{r_{WGD}}{r_{control}} = \frac{4}{3} = 1.33$$

To verify this prediction, we phased the Hi-C reads for the RPE^{TP53-/-} control and WGD samples, obtaining one Hi-C map for each haplotype in the two conditions (phased haplotypes were estimated by phasing whole genome sequencing reads, see the updated Methods section for a description of the procedure). We then compared the ratio of *in trans* vs. *in cis* contacts in control ($r_{control}$) and WGD (r_{WGD}) cells. The results clearly show an increase of *in trans* interactions in WGD cells (**Rebuttal Fig. 1**) leading to a ratio $\frac{r_{WGD}}{r_{control}} = 1.25$ on average, which is close to the predicted value of 1.33. (We should note that the slight difference between the predicted and measured values could be due to the phasing algorithm itself, which intrinsically aims at minimizing the fraction of "*in trans*" contacts, hence potentially leading to an underestimation of their actual value.)

Finally, even though contacts among duplicated homologous chromosome will be erroneously considered "*in cis*", we do not think this affects our results on loss of chromatin segregation (LCS) estimation (lines 131-141 of the originally submitted version of the manuscript). Indeed, these "false *in cis*" contacts are expected to be quite rare, of the same magnitude of in trans contact (see **Rebuttal Fig. 1**). Moreover, chromatin compartments and domains are largely preserved between Control and WGD cells and, based on our comparisons, highly similar between haplotypes. Thus, it is unlikely that small differences in chromatin topology between duplicated homologous chromosomes could lead to the observed extent of LCS. Lastly, our WGD induction experiments in TP53 wild type cells or cells treated with a CDK4/6 inhibitor do not exhibit LCS (see new **Figure 2**).

Rebuttal Fig. 1: Fraction of interactions between homologous chromosomes (Haplotype 1 and Haplotype 2) for each chromosome. In gray are control cells, in shades of red are two independent WGD experiments in RPE TP53^{-/-} cells.

In summary, a change in the rates of *in cis* and *in trans* interactions do occur as the reviewer correctly anticipated, but this is modest and, based on our results, unlikely to impact our conclusions.

I can't make any sense of ED Figure 6a. Why are we interested in the Pearson correlation of O/E between different chromosomes? Indeed, how is that even computed, since the chromosomes are of different lengths? I feel like I'm missing something fundamental here. In general, why is Pearson of O/E used, rather than SCC as in ED Fig 2a? To me, SCC seems better.

The Pearson correlation coefficient of observed vs. expected contacts is often used to visualize contact enrichment of Hi-C contact maps. This technique has been frequently adopted to visualize chromatin compartments (Lieberman-Aiden et al., 2009; Rao et al., 2014) where it leads to the popular “plaid pattern” associated with intra-compartment enrichment vs. inter-compartment depletion of contacts (e.g., Fig. 3d,e in *Lieberman-Aiden et al., 2009*).

Here, we used the same strategy to visualize the enrichment of contacts between different chromosomes. The way it is computed is rather simple:

1. compute the matrix of observed vs. expected contacts (O/E matrix) between each pair of chromosomes (the expected number of contacts accounts for the chromosome length and the total number of contacts made by that chromosome);
2. compute the Pearson correlation between each pair of rows of the O/E matrix.

As a result, we can test if each pair of chromosomes interacts more or less frequently than expected and visualize the clusters of long and short chromosomes that are commonly observed in human cells. The stratum correlation coefficient is instead a correlation measure between two different matrices. We have used this metric to compare two intra-chromosomal contact matrices.

The observation in Figure 3b is striking, but I'm not sure I buy the argument that this phenomenon indicates “that these cells intrinsically exhibited LCS.” As mentioned earlier, the changes in interaction frequencies between long/short chromosomes could best be explained

as resulting from the change in the number of homologous copies of each chromosome. Indeed, what I don't understand are the cells that seem to not exhibit such changes. I would suggest that the authors adopt a more precise terminology than "LCS" and "cells w/o inter-chr LCS" to refer to these two groups.

In this comment and the one below the reviewer indeed raises a valid concern regarding the phrasing of this section of the manuscript and in the figure. Overall, based on contacts among distinct chromosomes and compartments, it appears that WGD cells exhibit a variable extent of LCS, with a subpopulation of WGD cells exhibiting values comparable to control cells. This can be explained by several factors including the experimental resolution, which is low in single-cell Hi-C compared to bulk experiments, heterogeneity of the cell cycle phase (early G1 vs. late G1/S), and presence of aneuploidies. Indeed, upon estimating copy number status in single cells from Hi-C coverage, we found that WGD cells with low or no LCS exhibited a higher number of aneuploidies (see new **Figure 3e**). We have amended the text to better describe the reported results (see also the response to the comment below).

I was confused by the summarization of results on lines 266-267: "Overall, these results indicate that LCS observed at the population level is an intrinsic feature of individual cells that underwent WGD." The hypothesis was framed in Figure 3a as a distinction between "intrinsic" and "emergent." Doesn't Figure 3b clearly show that there are two classes of cells, and subsequent analysis support this idea? The analysis in lines 263-265 suggests that one of those groups has compartment structure that is statistically indistinguishable from the control cells. I thus thought that we were clearly in the "emergent" scenario in Figure 3a, with two distinct groups of cells. Why is the conclusion that this is "intrinsic"? Maybe it's just that the figure is misleading. I think the data shows that a subset of the cells look like wild type and a subset show significant changes in both long-vs-short chromosome contacts and in compartment structure. The text (and figure) should do a better job of conveying this. The choice of terminology (emergent versus intrinsic) should be reconsidered.

As mentioned in the response above, indeed the description in the text required improvements. We followed the reviewer's suggestion, and we have now better described the results presented in **Figure 3**, avoiding the "emergent versus intrinsic" terminology.

As observed by the reviewer, there is quite a variability in the extent of LCS among WGD cells with possibly two populations: one exhibiting intrinsically LCS (i.e., LCS is detected in each single cell) and one that doesn't. Thus, we cannot completely discriminate between the intrinsic or emergent scenario. We have clarified this in the text.

We want to point out, that we are not in a "clearly emergent" scenario where LCS is just the result of admixing multiple cells with different chromatin conformation. If that was the case, all WGD cells would have inter-chromosome and inter-compartment contact values undistinguishable from control cells. In other words, none of the single cells would exhibit LCS, but LCS would just be the product of admixing cells with heterogenous (but well segregated) chromatin topologies.

As the reviewer mentioned, demonstrating LCS within single cells is a striking result, especially given the sparsity of single cell Hi-C data, as it further supports a cell intrinsic mechanism

(protein shortage, which we describe in **Figure 2**). However, we might have over-emphasized and inaccurately described these findings in the first version of the manuscript. We have now revised the description of these results.

The algorithm for segmenting the delta compartment vectors (line 816-825) is, on the face of it, reasonable but pretty arbitrary. This is an example of a changepoint detection problem, for which there are many existing algorithms (including standard libraries in Python and R). I think the authors should demonstrate that their results do not depend on the particulars of this new changepoint detection algorithm by recapitulating at least some of their results using a commonly used changepoint detection algorithm.

Indeed, several changepoint detection algorithm have been proposed in the literature. To address the reviewer's concern, we selected one of the most used changepoint detection algorithms in bioinformatics, *circular binary segmentation* (CBS), which was initially proposed to segment copy number ratios and is implemented in the R package DNACopy (Olshen et al., 2004). Using CBS, we repeated the analysis of compartment repositioning events (CoREs) in tumors derived from 20-weeks post-WGD RPE^{TP53-/-} cells and we detected a largely overlapping set of CoREs, either sharing the same boundaries or with boundaries in high proximity of those detected with our approach (here termed DiffComp), including but not limited to, the most significant ones discussed in the manuscript. Moreover, even when using CoREs detected with CBS, the majority of significant CoREs could be traced back to compartment changes occurring already in WGD cells confirming the results obtained with our approach. These new results are included in the **Supplementary Figure 4**, which we also report below for the reviewer's perusal.

Supplementary Fig. 4: **a)** Example of CoREs detected using the DiffComp or CBS algorithm for delta-rank segmentation. **b)** Distribution of distances between DiffComp and CBS breakpoints (red) and between DiffComp breakpoints and randomly generated ones. **c)** Volcano plot representation of CoREs detected by DiffComp (left) and CBS (right). CoREs involving the selected oncogenes and tumor suppressors discussed in the manuscript are labeled and detected by both approaches. **d)** Number of activating and inactivating CoREs detected by both algorithm (shared) or by only one of them. **e-f)** Percentage of activating (red) and inactivating (blue) CoREs that were detected by CBS and that could be traced back to compartment changes occurring already in WGD cells. Trends and percentages closely resemble those observed for CoREs detected with DiffComp (see **Figure 5**).

Minor

Line 25: I don't understand what it means to "upscale protein production of the chromatin architectural protein CTCF." Grammatically, this is incorrect, but even if we interpret the phrase to mean that the cells are failing to increase CTCF production, it's not clear in what context this increase is expected to happen.

In WGD cells, we observed that protein levels of CTCF (and now also H3K9me3) are relatively lower than in diploid cells, considering that WGD cells have doubled the amount of genomic material (see new **Figure 2**). This can be attributed to the inability of WGD cells to upscale/increase protein synthesis of specific chromatin regulators after WGD induction. A similar observation has been recently made for DNA replication proteins (Gemble et al., 2022), the shortage of which was shown to increase DNA damage in WGD cells. Overall, both studies indicate that WGD cells fail to increase production of specific proteins and protein

shortage is an underlying mechanism of WGD cell phenotypes. We have amended the manuscript to clarify this concept.

Line 41: "imputed" is the wrong word here.

We amended the text and substituted "imputed" with "explained by".

Line 70: It seems to me that this should be the start of a new paragraph.

We reformatted the text.

Line 78-80: Grammatically, I couldn't make sense of this sentence.

We rephrased this sentence.

In Extended Data Fig 2, "CALDER" has not yet been defined. Give the citation here.

We included the citation.

Line 121: In the caption to the Extended Data Figure 3, one sentence refers to inter-chromosomal contacts and the subsequent one refers to intra-chromosomal contacts. Which is correct? "Inter-" makes no sense (The O/E transformation is intended to correct for the genomic distance effect, which is a property of cis contacts), so I am guessing it is supposed to be "Intra-." I don't understand why lower O/E values imply that "chromatin structural elements were less segregated."

Indeed, what was before **ED Figure 3a** (currently **ED Figure 2g**) corresponds to intra-chromosomal interactions. We corrected the typo.

In the O/E computation, expected counts indeed account for the genomic distance effect, but not for the underlying topological elements (e.g., TADs and compartments), hence the *expected* distribution of contacts is uniform at the same genomic distance (see Suppl. Fig 9 in Liebermann-Aiden et al., 2009). Conversely, the more segregated the topological elements are, the more skewed / non-uniform the observed distribution of contacts will be. Skewed observed distributions of contacts lead to high O/E values: highly positive O/E values correspond to the enrichment of contacts within the same TAD or compartment, and highly negative O/E values correspond to the depletion of contacts among different TADs or compartments. Hence, lower O/E absolute values in WGD cells than in Control cells, indicate that chromatin elements are less segregated in WGD cells than they are in Control cells.

Line 211-213: This observation -- that contacts between long and short chromosomes increase even p53 wild type WGD cells -- does not seem surprising to me, and emphasizes what was already mentioned above (line 143-144) that the increased contact frequency is simply explained by the doubled number of homologous chromosomes.

We agree with the reviewer that at the chromosome levels the increased contact frequency in WGD cells is *simply* associated with the double number of chromosomes. Indeed, we

observed this in all WGD experiments (**Figure 1 and 2**) and that is why in this sentence we mention that this is independent of CTCF or p53 status. We amended this sentence to clarify this observation.

Line 218-221: This distinction is now clear, but the mention of it in the abstract was not.

We have now substantially edited the abstract and, as mentioned above, improved the description of the data in **Figure 3**.

Line 224: I think, somewhere around here, you should mention the mean number of obtained contacts per cell. I would like to see a supplementary table or figure that shows what this distribution looks like.

We included a supplementary table (**Supplementary Table 2**) with summary statistics for each single cell and mentioned the average number of contacts in the text (*mean number of filtered interactions per cell: 565,324*).

Line 250-257: I did not figure Figure 3d or the associated text very informative. I'm not sure what this analysis tells us. On the other hand, Figure 3e is quite compelling.

While in the previous panels of **Figure 3** we showed that a subset of WGD cells increases the number of contacts between long and short chromosomes, in what is now **Figure 3c** (formerly **3d**) we showed that specific pairs of long and short chromosomes interact more frequently than others in WGD cells. This phenomenon might favor chromosomal translocations. For example, this analysis showed that chr. 1 and chr. 16 are among the long-short chromosomes that interact more frequently in WGD cells. This is quite intriguing because in **Figure 4g** we show that tumors originated from WGD cells exhibit a translocation between chr. 1 and chr. 16. Importantly, in WGD cells we could not detect a single read supporting this translocation, indicating that it was acquired at a later timepoint. Although correlative, we thought it was an interesting observation linking increased contact frequency among specific long and short chromosomes (**Figure 3c**) and acquired chromosomal translocations (**Figure 4g**).

I did not find ED Fig 9c convincing. I am not sure what message we are supposed to glean from this figure. And ED Fig 9d is not well described. What is this a distribution over? I understand that the x-axis in each plot is correlation, but I can't figure out what the vertical axis is a count/density over.

The former **ED Figure 9c** (which in the revised version of the manuscript is **ED Figure 9f**) shows scatterplots comparing the difference (delta) in compartment rank (X-axis) and ChIP-seq signal (Y-axis) at each genomic bin (50kb). A plot is shown for each tumor (rows) and histone mark (columns). The message here is that even though there is not always a perfect correlation between the two, we typically see an association between compartment changes (delta compartment rank > or < than 0) and histone mark changes, especially for H3K27ac and H3K9me3. In what is now **ED Figure 9g**, we show the distribution of expected correlation values between delta compartment rank and delta ChIP-seq signal for the CoREs that we identified. To estimate the significance of the observed correlations, we computed an empirical p-value by randomly sampling a number of regions of the same size of the observed

CoREs across the genome and re-computing the correlation value. The distributions of correlation values obtained over 1000 random samplings are shown in **ED Figure 9g**, with the histograms corresponding to the number of times a specific correlation value was found in 1000 random trials (correlation values are automatically binned by the histplot function). The red line indicates the observed correlation value, and we report the associated empirical p-value corresponding to the number of times the expected correlation was greater or equal in absolute value to the one observed, divided by the total number of random trials (when the empirical p-value is equal to 0, we report $p < 0.001$). We amended the figure legend to clarify the figure content and we included a detailed description in the method section.

Line 400-401: How were the 6000 regions selected for testing?

In each tumor, the changepoint detection algorithm that we adopted identifies approximately 6000 segments, each of which was tested. We amended the sentence in the text to clarify that that number simply indicates the total number of segments detected by our approach.

Supplement:

Lines 55-56: It seems a bit funny to critique the standard method for not providing a quantification of domain strength. An easy way to do this is to examine the eigenvalues associated with each compartment region. But the authors are not even obliged to make this argument here, since they are using a previously described algorithm (Calder) to get a continuous value for each compartment region.

Maybe this was a misunderstanding, we didn't mean to critique the standard method but simply to provide some background and cite the literature. We removed that sentence.

In general, I don't see why these two supplementary notes are necessary. If the authors think this information is critical, please just include it (briefly) within Methods. I think the first note, in particular, is not necessary at all.

We agree with the reviewer, and we have now integrated the first supplementary note in the methods section. The second note goes together with the Supplementary Data Figure describing the single cell selection for scHi-C experiments. In the revised version of the manuscript, we have restructured the extended data figures and material to include all necessary information while trying to respect the journal format requirements.

Referee #2 (Remarks to the Author):

Lambuta et al., explore the impact of WGD in cancer with regards to 3D organization of chromatin and its contribution to oncogenic phenotypes. *This is a very interesting and thorough manuscript*. It is dense and contains a lot of information. Overall, I think it is well written and clear. However, there are a few areas where additional clarity could improve the manuscript (particularly towards the end). I have two main concerns (outlined in more detail below): 1. Are the results a result of WGD rather than CIN? 2. Are the same results observed in the context of naturally occurring WGD (rather than induced WGD)?

We appreciated the positive assessment of the reviewer. Here, we would like to first summarize the new experiments and analyses that we did to address the two main concerns that the reviewer raised. A more detailed description is provided in response to the specific comments.

1. To disentangle the effect of WGD and CIN, we have induced CIN independently of WGD in RPE^{TP53-/-} using the Mps1-inhibitor (Mps1i) Reversine, as previously described (Santaguida et al., 2017). We performed Hi-C in these cells after CIN induction and, importantly, in this condition, we did not detect loss of insulation at TAD boundaries or loss of segregation of chromatin compartments (see the updated **Figure 1g,h**). We still observe some changes in contact frequency between long and short chromosomes. However, as we discussed in the manuscript, these seem independent of chromatin architectural proteins and, in this case, due to newly acquired aneuploidies in Mps1i treated cells.

In addition, we have performed new analyses:

- a) to infer copy number changes from single Hi-C data and showed that the amount of copy number alterations (CNAs) does not correlate with loss of chromatin segregation in these cells (new **Figure 3e**),
- b) to determine the association between significant compartment repositioning events (CoREs) in tumors and acquired CNAs. Here, we confirmed that while chromosome regions affected by CNAs also undergo compartment repositioning, several chromosomes without CNA exhibit multiple CoREs (new **Figure 5c,d**).

In summary, with these new data we can conclude that WGD induces both CIN and LCS, possibly through shortage of different proteins (as discussed at the beginning of this response), but CIN in itself is neither the cause nor a necessary condition for LCS.

2. To study LCS in spontaneously occurring WGD was more difficult due to the limited number of cells that spontaneously undergo WGD (compared to the number required for a Hi-C experiment) and the impossibility of synchronizing these cells. Nevertheless, we monitored CPA^{TP53-/-} cells and sorted a small subpopulation that spontaneously acquired high ploidy, which could be indicative that these cells underwent to WGD (~1% of the cells, **Rebuttal Fig. 2a**). We then performed Hi-C of this population and examined chromatin segregation as well as copy number changes and compartment repositioning. Indeed, these cells could have undergone WGD at a different time before sorting (they are not synchronized) and acquired both chromatin changes and copy number alterations, representing a heterogenous high-ploidy population. Indeed, we found that spontaneous high ploidy CPA^{TP53-/-} cells exhibited two new chromosomal translocations and acquired copy number changes (**Rebuttal Fig. 2b,c**). Importantly, while tetraploid cells obtained by induction of WGD in CPA^{TP53-/-} cells exhibited loss of compartment segregation (**Figure 1**) but no significant changes of compartments and (CoREs), spontaneous WGD CPA^{TP53-/-} cells exhibited moderate loss of compartment segregation but a high number of significant CoREs (**Rebuttal Fig. 2d,e**), similar to what we observed in tumors and soft-agar colonies evolving from WGD cells (see **Figure 5** and **6**). Finally, because of the limited number of spontaneous WGD cells obtained, it was not possible to assess histone marks and CTCF protein levels in these cells. Hence, we think that these samples are not truly representative of the WGD state (unlike the cells where WGD was

pharmacologically induced, and Hi-C immediately performed) but rather of a potentially heterogeneous cell population that has evolved for an imprecise amount of time after a spontaneous WGD event.

In parallel, we compared stable tetraploid and diploid cell lines derived from different Barrett esophagus patients: CP-D cells which are stable near-tetraploid cells and diploid CP-A^{TP53-/-} cells. Overall, we could confirm that also CP-D cells show a certain extent of LCS (see **Rebuttal Fig. 2f,g**), but this is milder than LCS observed in our syngeneic experiments (see **Figure 1**). Overall, while interesting, we believe that introducing this model would just introduce new conditions and possible confounders (such as new non-syngeneic models, non-synchronized and potentially heterogeneous cells) rather than strengthen the results already in the manuscript. For these reasons, we would prefer not to include these results in the manuscript, but we will do so if the reviewer thinks otherwise.

Finally, in this revised manuscript, we further confirmed that the LCS phenotype does not depend on our WGD induction protocol (as detailed below), but it is driven by reduced CTCF and H3K9me3 levels in the first cell cycles after WGD (see new **Figure 2**). Indeed, we have now induced WGD in RPE^{TP53-/-} by cytokinesis failure using two different protocols and we observed LCS in both cases (see updated **Figure 1**). We then induced WGD in CP-A *TP53* wild-type and knock-out cells using the same protocol (**Figure 1 and 2**), but *TP53* wild-type cells after WGD showed similar CTCF and histone marks levels and, as a consequence, LCS was not detectable in these models (see new **Figure 2**). Lastly, we demonstrated that stalling the cells in G1 (by treatment with the CDK4/6 inhibitor Palbociclib) restores CTCF and H3K9me3 levels and chromatin segregation (see new **Figure 2**). We believe that these experiments convincingly show that LCS does not depend on our induction protocol, but it is driven by reduced levels of chromatin regulators in WGD cells.

Rebuttal Fig. 2: a) Flow-cytometry based sorting of the spontaneous WGD cells from the CP-A^{TP53-/-} population. b) Inter-chromosomal Hi-C maps show contact patterns consistent with chromosomal translocations (arrows). c) Relative copy number profile of control and spontaneous WGD CPA^{TP53-/-} cells (red: gains, blue: losses). d) Density distribution of the difference (delta) of compartment ranks of matching genomic bins between of spontaneous WGD and control cell (red), induced WGD and control cells (black), and replicates of control cells (gray). e) Volcano plots showing the delta compartment rank values for candidate CoREs (X-axis) and associated p-values (Y-axis) obtained from the comparison of induced WGD and control cells (left) and spontaneous WGD and control cells (right). Significant CoREs are in red and blue, non-significant are in gray. f-g) Ratio of inter-chromosomal O/E contact values (f) and gain vs. loss compartment changes (g) obtained by comparing CP-D and CPA^{TP53-/-} cells.

Below we described more extensively the results of all the experiments included in the revised version of the manuscript to support these points and respond to all the other requests of the reviewer.

Comments:

- The authors explore WGD in three distinct cellular models. However, in all cases they induce WGD through treatment with nocodazole. I think it would be beneficial to also consider naturally occurring cell-lines with WGD, to the nocodazole treated cell-lines can be considered equivalent. While I appreciate this would not permit a direct comparison of before and after treatment, I think this is a crucial analysis to ensure this can be considered equivalent to the WGD observed in tumours.

As already discussed above, LCS in spontaneously emerging WGD cells was challenging given the extremely low number of cells that could be isolated from asynchronous cell population in syngeneic condition. Nonetheless, in CP-A^{TP53-/-} cells that spontaneously underwent WGD,

we do observe several significant compartment changes (**Rebuttal Fig. 2**) compared to the parental diploid cell line, as well as the acquisition of new translocations, consistent with WGD-induced LCS and CIN. Moreover, as an alternative to syngeneic tetraploid and diploid cells, we used tetraploid and diploid cells with the same tissue of origin. Indeed, we compared Barret esophagus cells, CP-D, that are stable near-tetraploid cells with our diploid CP-A^{TP53^{-/-}} clones. However, as already anticipated, this comparison is sub-optimal since we cannot really assess acquisition of CNAs or compartment repositioning events (CoREs) post-WGD in non-syngeneic models, and it is difficult to compare CTCF and H3K9me3 levels between different cell lines. Nevertheless, in this comparison, we could still detect a modest LCS, with increased interactions among log and short chromosomes and different A and B sub-compartment (**Rebuttal Fig. 2**), but no difference in boundary insulation as most probably these cells have compensated CTCF levels.

As already mentioned above, we can exclude that the observed LCS phenotype is dependent on our induction protocol. Indeed, 1) we induced WGD with and without nocodazole in RPE^{TP53^{-/-}} cells and in both cases LCS was detected, and 2) we induced WGD with nocodazole in TP53 wild type CP-A cells, but we did NOT observe LCS in those models (see new **Figure 2**).

- Relatedly, the authors could consider a naturally occurring isogenic genome doubling system in many cell-lines a naturally occurring doubled population can be isolated. This would ensure confidence that this is not related to the treatment.

Following the reviewer's suggestion, we sorted naturally occurring WGD cell population in CP-A^{TP53^{-/-}} cells, although this corresponded to ~1% of the population (**Rebuttal Fig. 2a**). The results and conclusions drawn from these models have already been discussed above.

- Likewise, if the authors induce CIN, but not WGD, is the increase in chromatin segregation observed? Or is it truly specific to WGD? In general, I think this is a critical question throughout the manuscript - how much of what is observed can be linked directly to WGD rather than simply increased instability and aneuploidy?

This is indeed a critical point. We were able to follow the reviewer's suggestion to disentangle the contribution of CIN to LCS. For this purpose, we induced CIN in RPE^{TP53^{-/-}} cells independently of WGD using an Mps1-inhibitor (Mps1i) as was previously done in the RPE parental cell line (Santaguida et al., 2017). Mps1i treatment led to accumulation of aneuploidies with cells exhibiting highly variable karyotypes (see **Figure 1e**). Next, we performed Hi-C in RPE^{TP53^{-/-}} cells after Mps1i treatment and compared chromatin segregation in these cells and control RPE^{TP53^{-/-}} cells. Importantly, in this condition, we did not detect loss of insulation at TAD boundaries or loss of segregation of chromatin compartments (see the updated **Figure 1g,h**). We still observe some changes in contact frequency between long and short chromosomes. However, as we discussed in the manuscript, these seem independent of chromatin architectural proteins and, in this case, can be attributed to newly acquired aneuploidies in Mps1i treated cells.

- Can the authors demonstrate that their results for various experiments (e.g. ChIP-seq, HiC etc) do not relate to differences in coverage per haploid copy. Are the results the same if they effectively down-sample the diploid control?

The reviewer raises here an interesting possibility. Since, for each Hi-C experiment, we aggregated multiple replicates, we can easily test this hypothesis by comparing aggregated and single replicates (which is equivalent to downsampling the diploid control).

We aggregated the replicates of WGD RPE^{TP53-/-} cells and compared it to individual replicates of control diploid cells. In this manner, we obtained twice the number of reads in WGD cells compared to diploid cells, corresponding to a similar coverage per haploid copy. In this condition, we can still observe LCS at the three topological levels (**Rebuttal Fig. 3a**).

Next, we compared one diploid control replicate vs aggregated diploid control replicates to simulate the same imbalance of coverage per haploid copy that we have in our WGD vs. control analyses. However, LCS was not detected in this comparison at any scale of the topological hierarchy (**Rebuttal Fig. 3b**). Overall, these results show that the Hi-C sample coverage does not determine LCS.

Rebuttal Fig. 3: Loss of chromosome segregation (top), compartment segregation (middle), and boundary insulation (bottom) between WGD and control cells (a) after downsampling control cells to maintain the same coverage per haplotype, and between one replicate and 2 aggregated replicates of control cells to simulate the different coverage per haplotype that we have in WGD and control cells (b).

- Relatedly, are the results preserved if the post-WGD diploid tumours are considered? (i.e. are these differences between WGD and nWGD cells maintained following significant loss of chromosomes).

The assessment of chromatin segregation in tumors with respect to RPE^{TP53-/-} control cells should be taken with a grain of salt since tumors were collected up to 6 weeks after engraftment of cells that underwent WGD 20 weeks before. During this time, these cells accumulated multiple aneuploidies and translocations, and changed their epigenome and transcriptome, all of which could contribute to alter chromatin topology. Nonetheless, we performed this comparison and found that LCS was significantly reduced in RPE^{TP53-/-} WGD cells (**Rebuttal Fig. 4**), similar to what observed in CP-D vs. CP-A cells (**Rebuttal Fig. 2**).

Although potentially interesting, we would prefer to not include this analysis in the manuscript for the reasons described above.

Rebuttal Fig. 4: Loss of chromosome segregation (top), compartment segregation (middle), and boundary insulation (bottom) between RPE-derived tumors and control cells.

- The authors state that: "The overall tumour mutation burden was similar across samples and did not significantly increase in RPE^{TP53-/-} cells that underwent WGD compared to RPE^{TP53-/-} cells that were kept in culture for the same amount of time (Extended Data Fig. 8c)." Given that there is twice the amount of DNA it seems surprising that there isn't twice the number of mutations - even if the same number of mutations per haploid copy. It would be interesting to explore this further - does this relate to the sequencing coverage? Or, is there a lower mutation rate?

We thank the reviewer for raising this point and indeed this analysis required further scrutiny. To improve the detection of bona fide newly acquired mutations in WGD and control cells after 6 weeks in culture, we applied additional filters and excluded from the analysis:

- mutations that were already present in the RPE^{TP53-/-} control diploid cells,
- mutations that were detected in both RPE^{TP53-/-} WGD cells after 6 weeks in culture and in RPE^{TP53-/-} diploid cells after 6 weeks in culture, since these are likely missed variants in the original control sample,
- and mutations that were detected in both RPE^{TP53-/-} WGD cells and in RPE^{TP53-/-} diploid after 6 weeks in culture, since these are also likely missed variants in the original control sample.

After applying these filters, we found an increased number of mutations (~1.8 fold-change) in 6 weeks post-WGD cells compared to diploid cells maintained in culture for 6 weeks (new **ED Figure 8b**), consistent with the prediction of the reviewer. Moreover, in WGD cells, we noticed an increased fraction of mutation with a low variant allele frequency (VAF), which was

even more accentuated in 6-weeks post-WGD cells. This is likely associated with the increased amount of DNA of the WGD population and a higher heterogeneity of the cell population at 6 weeks post-WGD, suggesting that the number of acquired mutations may be underestimated (new **ED Figure 8b**).

- The authors observe that "tumours from 20-week post-WGD cells exhibited remarkably similar CNV profiles, indicating the selection and expansion of the same clone in vivo, which was characterized by copy number gains of chromosomes 1p, 13q, and 8q, and losses within chromosomes 1q, 4p, 13p, and 18p (Fig. 4d)." To confirm this can the authors evaluate whether indeed the same allele is gained or lost (are there sufficient heterozygous SNPs?), and further, whether there is evidence for the same breakpoint? (i.e. is this certainly selection and expansion of the same clone rather than convergent/parallel evolution?)

To test whether tumors derived from RPE^{TP53-/-} 20-weeks post-WGD cells originated from the selection of the same clone or whether they converged towards similar genomic profiles, we compared copy number breakpoints detected by whole genome sequencing (WGS) and determined whole-chromosome phased haplotypes using both WGS and Hi-C data. Using this data, we determined a large overlap of copy number breakpoints among the 3 tumors, in particular all the major CNV that we detected in tumors (at 1q, 4p, 13p, and 18p) exhibit the same breakpoints in all three tumors (new **ED Figure 8c**). Consistently, haplotype-resolved genomic profiles showed that losses and gains always occur in the same haplotype (new **Figure 4e** and **ED Figure 8d**). Overall, these results indicate that in this experiment a specific clone was selected post-WGD and was able to originate tumors in mice.

- I'm afraid I find Figure 4d and e difficult to follow, I think it would be easier to interpret if the different timepoints were more clearly separated, and a heterogeneity score assessed.

We have improved the labeling of **Figure 4** and to clearly indicate to which sample each data panel refers and we re-formatted **Figure 4d** and what is now **4f**. Moreover, as heterogeneity score, we reported the fraction of genome altered (FGA), which had been previously defined as the percentage of the genome where the estimated copy number ratio in absolute number is > than 0.3 (Sanchez-Vega et al., 2018).

- Given the various comparisons in 'chromatin evolution of WGD cells' it is easy to get lost - the authors are comparing RPE_{tp53-/-} WGD cells, with 20-weeks post WGD cells and control (non-WGD cells). But, given there are multiple instances of WGD (in tumour and cell-line), simply describing a cell as WGD and tumour is not terribly informative.

As mentioned above, we have revised the labeling of the samples, to maintain consistency in the nomenclature of the samples across all figures and improve clarity. We hope this helps to better follow the described results.

- The authors conclude that 'Overall, our study demonstrates that WGD promotes changes of the chromatin 3D organization contributing to deregulating oncogenes and tumour suppressors, in parallel and largely independently of chromosomal instability.' I think more is needed to demonstrate this is truly independent of WGD

We have rephrased this sentence to better reflect the results and conclusion of the paper, in which we might have overstated our conclusion. Indeed, we think that LCS occurs in parallel of CIN and, although sometimes associated, LCS and CIN are both driven by WGD but through different mechanisms and both can contribute to the acquisition of oncogenic traits.

- I wonder if the manuscript could benefit from a cartoon (perhaps in Supplementary) summarising the key findings and how they link together? I think something like this could make the take-home message clearer.

We have now included a graphical summary to better contextualize the findings of this manuscript (**Figure 6h**).

Referee #3 (Remarks to the Author):

In this manuscript, Lambuta & Nanni et al. explore the chromatin organization in cells that have undergone whole-genome doubling (WGD). The authors induced WGD in three human cell lines and performed Hi-C of the matched diploid/tetraploid pairs. They observed reduced segregation of chromatin structural elements in WGD+ cells, increased short-range and long-range interactions between chromosomes and chromatin sub-compartments, and repositioning of sub-compartments that were associated with oncogenic transcriptional changes. The authors propose that these chromatin changes occurred independently of mutations and chromosomal aberrations, and are therefore complementary mechanisms that contribute to tumor formation/evolution following WGD.

Overall, *this is an interesting study*. There are surprisingly few published papers that have evaluated the effect of ploidy changes (both polyploidy and aneuploidy) on chromatin organization and nuclear architecture. *The manuscript presents rigorous experiments, and data analysis also appears to be sound*. The novelty and conceptual advance are borderline for publication in Nature in my mind. In any case, to justify publication in Nature, I believe that more work needs to be done in order to: (a) further link the observed chromatin organization changes to gene expression changes and tumorigenicity; (b) convincingly demonstrate that the effects of WGD are indeed independent of chromosomal instability (CIN); and (c) generalize the findings by using additional methods for WGD induction, another cell line for the in vivo experiments, and (potentially) analyzing available clinical data from cancer patients.

We thank the reviewer for the positive feedback and finding our study interesting and rigorous. The revised version of the manuscript includes several new experiments that corroborate our findings and shed new light on the mechanisms driving loss of chromatin segregation (LCS) in WGD cells. In particular, to follow the reviewer's main suggestions:

- a) We have now included more thorough analyses of transcriptional changes both in WGD cells and in tumor originated from these cells.
- b) To further disentangle the contribution of CIN and LCS, we performed new experiments inducing CIN using an Mps1 inhibitor in cells that do not undergo WGD. Hi-C analyses show that these cells do not exhibit LCS (see new **Figure 1f-h**). In

addition, we have performed new analyses to infer copy number alterations in single cells and show that CIN and LCS do not correlate in single cells, and we studied more rigorously the association between compartment repositioning events in tumors and copy number alterations. Overall, our analyses show that WGD induces CIN and LCS through different mechanisms and both contribute to WGD cell transformation.

- c) Lastly, we included an independent model using CP-A^{TP53-/-} cells post-WGD to demonstrate that LCS and CIN contribute to acquisition of oncogenic traits. In this model, we performed Hi-C analyses of colonies formed from CP-A^{TP53-/-} cells that underwent WGD. These cells also showed the accumulation of copy number alterations and chromatin compartment repositioning, confirming all the major findings that we obtained in RPE^{TP53-/-} cells. Most importantly, compartment repositioning events in CP-A^{TP53-/-}-derived colonies could be traced back to chromatin changes occurring already in WGD cells, in two independent WGD experiments. All these results are in the new **Figure 6** and described in depth in response to specific comments.

Below, we explain more extensively the new data and analyses included in the revised version of the manuscript to address all the specific points of the reviewer.

Major comments

Robustness:

(1) The chromatin organization changes were partly dependent on p53 status – p53-WT WGD+ cells neither exhibited enriched contacts between A and B sub-compartments, nor loss of insulation at TAD boundaries, but still exhibited increased contact frequencies between long and short chromosomes. These data suggest that much of the observed loss of chromatin segregation (LCS) depends on p53. However, this comparison was only done in a single cell line, and should be repeated using additional cell lines.

The reviewer is correct, p53 status is key in our experiments due to its role in the tetraploid checkpoint, i.e., preventing cell cycle progression in cells that entered G1 in a tetraploid status (Andreassen et al., 2017; Margolis et al., 2003). In the initial version of the manuscript, we indeed analyzed LCS in 3 *TP53*^{-/-} models (CP-A^{TP53-/-}, RPE ^{TP53-/-}, and K562) but only one *TP53* wild type model (CP-A). Following the reviewer's suggestion, we induced WGD in RPE *TP53* wild-type cells, however, the majority of the cells remained binucleated. To obtain a mononucleated tetraploid population that could be analyzed by Hi-C, we isolated tetraploid cells, but again the majority did not recover post-WGD, and we were able to obtain only one clone out of 384 sorted cells. Cells from this clone were larger and appeared quiescent, making it difficult to directly compare chromatin organization between these cells and proliferating RPE ^{TP53-/-} cells, as the difference could arise from different cell state.

Conversely we could demonstrate with a series of new experiments that LCS in *TP53*^{-/-} cells is driven by reduced levels of CTCF, which provides boundary insulation, and H3K9me3, which drives heterochromatin compaction and chromatin compartmentalization (Falk et al., 2019; Padeken et al., 2022). In *TP53* wild type cells, the activation of the tetraploid checkpoint stalls the cell in G1 for a longer period and allows to restore the levels of these chromatin regulators

and, as a consequence, chromatin segregation. To demonstrate that prolonging G1 phase in p53^{-/-} cells is sufficient to rescue CTCF and H3K9me3 levels and LCS, we induced WGD in CP-A^{TP53^{-/-}} cells while treating them with the CDK4/6 inhibitor Palbociclib. In this experiment, we showed that treatment with Palbociclib was sufficient to restore CTCF and H3K9me3 levels and, as a consequence, LCS was significantly reduced (new **Figure 2h-j**).

In addition, can the authors add a control of p53^{+/+} vs. p53^{-/-} cells to determine whether p53 can lead (in and of itself) to any changes in chromatin organization? In particular: does p53 inactivation have any effect on CTCF levels and TAD boundaries?

To address this point, we have compared the chromatin status in control *TP53^{-/-}* and *TP53* wild-type cells for both CP-A and RPE cell lines, without inducing WGD. In all comparisons, we did not observe LCS at any level of the chromatin topology hierarchy (**Rebuttal Fig. 5a**) and CTCF levels were unchanged (**Rebuttal Fig. 5b**).

Rebuttal Fig. 5: (a) Loss of chromosome segregation (top), compartment segregation (middle), and boundary insulation (bottom) between *TP53^{-/-}* and *TP53* wild-type cells. (b) Western-blot analysis of CTCF protein expression.

(2) Relatively little data is provided to link the observed changes in chromatin organization to perturbed transcription and altered gene expression. What would be the transcriptional consequence of the increased interactions between short and long chromosomes, increased contacts between A and B sub-compartments, and reduced insulation at TAD boundaries? The authors analyzed the effect of compartment repositioning events (CoREs) on gene expression, but what about the other described chromatin alterations (i.e., the increased

chromosome interactions and the perturbed TAD boundaries)? This should be experimentally addressed at least at the population level.

To study the consequences on gene expression of LCS in WGD cells, we generated and analyzed RNA-seq data in diploid and WGD RPE^{TP53^{-/-}} cells. Interestingly, among differentially expressed genes ($|\log_2FC| > 1$ and adjusted p-value < 0.01), the majority were upregulated (n = 1268 out of 1887) and, in general, up-regulated genes exhibited higher fold-changes and lower p-values (see new **ED Figure 4a**). This broad up-regulation is likely due to the doubling of genetic material. Among significantly upregulated genes we detected an enrichment for genes involved in interferon signaling (new **ED Figure 4b**), consistent with activation of this pathway mediated by cGAS/STING in response to chromosome segregation errors (Bakhoun and Cantley, 2018; Santaguida et al., 2017).

On the other hand, although mRNA downregulation was observed in fewer genes and was overall more moderate, downregulated genes exhibited highly significant enrichments for specific functional categories associated with DNA replication, DNA repair, and cell cycle progression (new **ED Figure 4c**). This is in line with recent findings showing that, in WGD cells, the expression of proteins associated with the replication machinery does not scale up with the doubled DNA material (Gemble et al., 2022). The protein shortage observed in that study is consistent with what we observed for CTCF (see **Figure 2a-c**). CTCF mRNA expression was also significantly downregulated in our experiments (adjusted p-value = $8.3E-6$) although with a more moderate fold-change ($\log_2FC = -0.6$).

Lastly, we sought to determine whether up-regulated and down-regulated genes were associated with loss of boundary insulation and/or compartment segregation. Here, we did not see a strong association between LCS features and differentially expressed genes. We saw a moderate enrichment for up-regulated genes in proximity of boundary losing insulation and, vice versa, of down-regulated genes in proximity of boundary gaining insulation, but the effect is very mild and difficult to relate with chromatin changes observed in WGD cells. (**Rebuttal Fig. 6**).

Rebuttal Fig. 6: Differentially expressed genes enrichment at boundaries that gain or lose insulation.

Overall, we believe that WGD triggers an immediate transcriptional response driven by mitotic errors and replication stress, which dominates the signal that we measured. Conversely, gene expression changes due to rewired chromatin interactions and sub-compartments, like those that we observed for *CTNNB1* or *BRCA1* (**Figure 5**), might require more time to emerge and consolidate.

The scRNA-seq data that was generated from the cells post-WGD is somewhat under-exploited in this regard – can these data be used to address the effect of loss of chromatin segregation (beyond CoREs) on gene expression? To assess the importance of the observed phenomenon, it would be important to quantify the fraction of gene expression changes in the WGD+ populations that are driven by CNVs vs. the fraction that can be linked to the chromatin changes.

Following up on the observations discussed in the previous point and the comment of the reviewer, we have more thoroughly investigated gene expression changes measured by scRNA-seq in tumors originated from post-WGD cells. Out of 1,396 genes which are differentially expressed between tumors and control cells, 14.5% are located within a CNV, 18.5% are associated to a CoRE, and 7% are in regions that exhibit both a CNV and a CoRE (new **Figure 5d**).

To trace concordant changes in gene expression and compartment organization to loss of compartment segregation occurring at WGD, we followed two strategies. First, we tested whether CoREs correspond to regions that already exhibited a concordant compartment change in WGD cells. In the vast majority of cases, we could indeed observe such concordant changes not only in tumors derived from RPE^{TP53-/-} WGD cells (**Figure 5n**), but also in 4 soft agar colonies originated from CP-A^{TP53-/-} WGD cells (2 colonies from Clone 3 and 2 colonies from Clone 19) (new **Figure 6e,f**). These results are derived from 3 independent WGD inductions (RPE^{TP53-/-}, CP-A^{TP53-/-} clone 3, CP-A^{TP53-/-} clone 19) and 7 biological replicates demonstrating remarkable consistency (additional details of this new experiment are provided in response to a more specific request of this reviewer below). Next, we assessed whether transcriptional changes were concordantly associated with compartment repositioning events. Here, we found that 42% of the CoREs were associated with differentially expressed genes, which is greater than expected (**Figure 5e**) and up-regulated genes are enriched in CoREs that shift towards more active compartments, while down-regulated genes are enriched in CoREs that shift towards a less active compartment (new **Figure 5f**). Overall, transcriptional changes occurred in the same direction of compartment repositioning events that, for the most part, can be traced back to compartment changes occurring already in WGD cells.

(3) Related to the previous point: the authors show interesting examples of compartment repositioning events associated with gene expression and epigenetic changes. In what fraction of differentially expressed genes between the tumors and the parental cell lines this mechanism can be identified? In other words, is this a common mechanism for activating oncogenic pathways and silencing tumor suppressive pathways?

We believe we already addressed most part of this question while answering to the previous point and the exact number and percentages are now reported in **Figure 5d**. We would like here to note that, overall, most differentially expressed genes seems to be associated with neither CNV nor CoREs. This is not surprising since most of these transcriptional changes reflect downstream consequences of (in)activation of different pathways during oncogenic transformation. Nonetheless, both CNVs and CoREs are enriched for differentially expressed genes, indicating that these are indeed mechanisms for activating oncogenes and silencing tumor suppressors.

(4) A key claim of this study is that loss of chromatin segregation and sub-compartment repositioning are CIN-independent mechanisms of WGD-driven tumorigenesis. The authors argue that the evidence for that is that most compartment repositioning events occur in regions unaffected by copy number changes or chromosomal re-arrangements. However, this does not really prove a CIN-independent mechanism. Notably, the magnitude of the observed

changes in the tumors was considerably higher than in the WGD cells from which they were formed (Fig. 5j,k), and at the tumor stage CIN was already prevalent. So it is not clear that this is indeed CIN-independent. To assess whether this is indeed the case, CIN should be inhibited in the WGD+ cells using MCAK overexpression or the inhibitor UMK57, and the effect on chromatin segregation and its associated gene expression changes should be evaluated.

We realized that we did not properly explain some of the results in the previous version of the manuscript, and some conclusions might have been perceived as overstatements and/or misinterpreted. We have now thoroughly revised the text to clarify our observations. To strengthen these conclusions and better assess the contribution of CIN to LCS and sub-compartment repositioning we have now performed several new experiments and analyses.

First, we induced CIN in RPE^{TP53-/-} cells without inducing WGD using a Mps1 inhibitor (Reversine, 0.5 μ M) as previously described (Santaguida et al., 2017). We confirmed by karyotype analyses the accumulation of aneuploidies in these cells (new **Figure 1e**). Importantly, RPE^{TP53-/-} cells treated with Mps1i did not exhibit LCS at any level of the chromatin topology hierarchy (see new **Figure 1f-h**). Hence, CIN by itself does not induce LCS.

In addition, we have performed new analyses:

- 1) to infer copy number changes from single Hi-C data and showed that the amount of copy number alterations (CNAs) does not correlate with loss of chromatin segregation in these cells (new **Figure 3e**),
- 2) to determine the association between significant compartment repositioning events (CoREs) and acquired CNAs in tumors. Here, we confirmed that while chromosome regions affected by CNAs also undergo compartment repositioning, several chromosomes without CNA exhibit nonetheless multiple CoREs (new **Figure 5c**).

Overall, WGD induced both CIN and LCS but through different mechanisms and, although sometimes associated, neither is the cause of the other.

Lastly, regarding the experiment proposed by the reviewer, we thought that this might not be feasible, and that is why we chose to induce CIN without WGD rather than induce WGD without CIN. Indeed, the reviewer correctly noticed that significant sub-compartment repositioning events occur in tumors originating from cells maintained for 20 weeks in culture (**Figure 5**), while compartment changes are smaller in magnitude in WGD cells (see, for example, the enhancer of *CTNFB1* in **Figure 5h**). Now, the reviewer suggests determining if compartment repositioning could occur independently of CIN in cells post-WGD, by treating the cells with UMK57 to prevent CIN. However, to assess the extent of compartment repositioning in this condition, we should: 1) induce WGD and at the same time treat the cells with UMK57, 2) wait for 6-to-20 weeks to inject post-WGD cells into mice, 3) wait for tumor formation and perform Hi-C on collected tumors. In this setting, short-term treatment with UMK57 would not prevent CIN in the long term, and the cells will still acquire CNAs over time. Conversely, long-term treatment with UMK57 would inevitably induce phenotypes associated with the treatment rather than WGD, making the results not comparable with those reported in our study. For these reasons, we proposed as an alternative the experiment with the Mps1-inhibitor.

Generalizability:

(5) The authors initially used 3 cell lines (CP-A, RPE1 and K562) for WGD induction. However, WGD was always induced by nocodazole treatment (either with and without subsequent DCB treatment). Additional methods for WGD induction (by mitotic slippage, cytokinesis failure or endoreduplication) must be used to ensure that the observed chromatin changes are indeed associated with WGD and not with the nocodazole treatment.

The reviewer raises here a valid point. However, both previous and new experiments now in the manuscript allow us to exclude the possibility that loss of chromatin segregation is associated with nocodazole treatment. Indeed, we have now induced WGD by cytokinesis failure in RPE $TP53^{-/-}$ using an independent protocol without nocodazole (CDK1 inhibitor + DCB), and we confirmed LCS in these cells (see updated **Figure 1f-h**). In addition, we used the same protocol utilizing nocodazole to induce WGD in CP-A $TP53$ wild type cells and in CP-A $TP53^{-/-}$ cells treated with Palbociclib: in both these experiments cells did NOT exhibit LCS (see new **Figure 2**). Our results show instead that in $TP53^{-/-}$ cells, a defective tetraploid checkpoint leads to reduced levels of both CTCF, which provides boundary insulation, and H3K9me3, which drives heterochromatin compaction and compartmentalization. Notably, CTCF and H3K9me3 levels were restored upon Palbociclib treatment and were unaffected in $TP53$ wild type cells. These results indicate that LCS is driven by insufficient protein synthesis rather than by nocodazole.

(6) Along the same lines: the *in vivo* experiments were done using a single cell line (RPE1). Given their central role in the manuscript as a whole and for some of the key novel claims, these experiments need to be repeated in another cell line.

We agree with the reviewer that demonstrating oncogenic transformation and emergence of sub-compartment repositioning in multiple cell lines is important. As anticipated at the beginning of the response to the reviewer's comments, we have now introduced a new tumor model using two independent CP-A $TP53^{-/-}$ clones (clone 3 and clone 19). Importantly, since WGD was induced separately in the two independent clones, this experiment allows to further assess the reproducibility of our results.

Similar to what we have observed in RPE $TP53^{-/-}$ cells post WGD (**Figure 4b**), both CP-A $TP53^{-/-}$ clones *in vitro* show mixed ploidy at 6 and 20 weeks post-WGD (**Supplementary Figure 5**). However, CP-A $TP53^{-/-}$ cells were not able to grow *in vivo*. This was not surprising since lack of xenograft engraftment has been observed with many cancer cells and PDX models and this has been linked to the absence of human-specific growth factors. Indeed, CP-A $TP53^{-/-}$ cells in culture require the addition of epidermal growth factor recombinant human protein, hydrocortisone, adenine, bovine pituitary extract, insulin-transferrin-sodium selenite, and other stimulatory factors. To cope with this issue, we grew CP-A $TP53^{-/-}$ cells in soft agar and monitored the formation of colonies mimicking the uncontrolled growth that is typically observed in tumor cells. In this condition, we could clearly show that only cells that underwent WGD were able to grow and form large colonies (new **Figure 6a,b**).

Next, we selected 2 colonies for each clone (4 colonies in total) and performed Hi-C experiments to infer copy number alterations (CNA) and compartment repositioning events

(CoREs). Importantly, in these independent models, we could recapitulate the molecular phenotype observed in RPE-derived tumors, characterized by the acquisition of multiple CNAs and CoREs (new **Figure 6c,d**). Strikingly, in all 4 colonies, the vast majority of CoREs could be traced back to concordant compartment changes occurring already at WGD, suggesting that WGD “primed” genomic regions for compartment repositioning (new **Figure 6e,f**). This was further confirmed when we look at the number of CoREs that were found in both colonies derived from the same clone. Indeed, in both clone 3 and clone 19, the number of shared CoREs between two colonies was higher for CoREs that were “primed” at WGD, suggesting these were early events present in most cells (new **Figure 6g**).

Hence, these results do not only validate our findings in an independent model but strengthen the link between loss of chromatin segregation occurring in WGD cells and compartment repositioning in tumors.

(7) The WGD status of tumors has been determined in many cancer genomic studies, including TCGA (see Taylor et al. Cancer Cell 2018 for example). Can the authors analyze available Hi-C data from previous studies to further address the generalizability and clinical relevance of the phenomenon that they describe?

Unfortunately, both LCS and sub-compartment repositioning can only be assessed through longitudinal analyses of matching samples, which represent a crucial and unique point of our study. Long after WGD has occurred, LCS would be hard to measure and confounded by multiple factors such as extensive chromosomal instability (which characterize WGD tumors) and clonal heterogeneity. Even when we compared our clonal RPE-derived tumors with their normal counterpart, we saw that, while still visible, LCS was significantly reduced (see **Rebuttal Fig. 4**). Lastly, by definition, compartment repositioning events can only be measured with respect to a proper reference, e.g., by comparing two syngeneic models before and after WGD, since chromatin sub-compartment frequently differ across different cell types and cell states (Dixon et al., 2015; Liu et al., 2021).

Referee #4 (Remarks to the Author):

Comments:

Lambuta et al. present a manuscript studying the impact of whole genome duplications (WGD) on 3D genome organization in the nucleus. In general, this is an understudied problem so the general topic in my mind will be of significant interest. They use three cell line models to induce WGD and examine the effects of genome doubling on nuclear structure. They observe reduced chromatin compartment and TAD separation. This is also accompanied by a reduction in the levels of the CTCF protein, suggesting that this may play a causal role. They then use the WGD cells to generate tumors in mice and examine the genetic and genomic changes that occur in the tumor samples that grow out. They observe several copy number alterations and genetic rearrangements, some of which they say are recurrent. They also observe changes in 3D genome structure, including changes in chromatin compartments, in the tumors compared with the original cell lines. While I find the general idea of the manuscript interesting, there are several aspects that I think are currently underdeveloped.

This includes the role of p53 and CTCF in contributing to changes in genome organization during WGD, as well as linking changes observed in chromatin organization in WGD with the final genetic or genomic changes observed in the final tumor samples. Due to these limitations, I don't think I can recommend publication at this time. I have organized my comments into major and minor comments below.

We thank the reviewer to find our study interesting and in the revised version of the manuscript we have included several new data to better explain the role of p53 and CTCF in genome organization (see new **Figure 2**) and we included a second independent model to demonstrate that the compartment repositioning events observed in tumors can be traced back to chromatin changes that emerge in WGD cells (**Figure 5** and new **Figure 6**). Below we describe the new experiments.

Major comments:

Mechanistically, it is hard to fully accept the changes in 3D genome organization after WGD as being the consequence of reduced levels of CTCF given the current data. Specifically, the authors indicate that the reduction in CTCF levels underlies the changes in compartment strength and TAD insulation seen after WGD. One issue that I see with this is that the loss of compartment strength in CTCF-AID cells (2d) seems to be less than the loss observed in WGD (1e), which to me raises the issue of to what extent these changes are being driven by CTCF.

We thank the reviewer for this comment which prompted us to further investigate whether changes in 3D genome organization after WGD, especially at chromatin compartments, could be solely attributed to reduced levels of CTCF.

First, we decided to remove the CTCF-AID example since that experiment was performed in mouse embryonic stem cells where, even in wild type / unperturbed conditions, compartments are less segregated than typically observed in human cell lines (actually, the distinct cell type and species is what explains the lower loss of compartment strength in CTCF-AID than in our WGD human cells).

Next, we expanded our analyses to investigate if other proteins or factors involved in chromatin organization changed in WGD cells and could explain the observed phenotype. Histone post-translational modifications have been associated with chromatin compartmentalization (Mirny et al., 2019); hence, we tested their levels in control and WGD cells. In both CP-A^{TP53^{-/-}} and RPE^{TP53^{-/-}} cells we confirmed reduced CTCF (but not RAD21) and we found downregulation of H3K9me3, but mild or absent changes in other histone marks such as H3K27ac and H3K27me3 (see new **Figure 2**). H3K9me3 is a marker of heterochromatin (enriched in the B.2.2 sub-compartment identified by Calder) and is considered a major driver of chromatin compartmentalization (Falk et al. 2019). Interestingly, both H3K9me3 loss and loss of compartment segregation are most evident in the B.2.2 compartment in our experiments (see new **Figure 2i,j** and **ED Figure 5c,d**). Importantly, this effect is specific of *TP53^{-/-}* cells, and it is not observed in the TP53 wild-type counterparts (**Figure 2**).

In line with our results showing downregulation of multiple proteins in WGD cells, a recently published paper showed that several DNA replication proteins are also insufficient in the first

G1 and S phases of newly WGD cells (Gemble et al., 2022). In their study, the authors showed that treatment with the CDK4/6 inhibitor Palbociclib stalls cells in G1 allowing them to scale up protein levels and rescue the phenotype. To rescue loss of chromatin segregation, we adopted the same strategy and induced WGD in cells treated with Palbociclib and performed Hi-C. Interestingly, in this condition, we rescued CTCF and H3K9me3 levels (**Figure 2f,g**) and, by rescuing protein levels, loss of boundary insulation (**Figure 2h**) and compartment segregation (**Figure 2i,j**) were significantly reduced.

Overall, these results confirm the reviewer's hypothesis: CTCF downregulation is only one part of the story. Experiments in our study demonstrate that *protein shortage* is a feature of WGD cells and this shortage translates in loss of chromatin segregation.

To me a really good control experiment would be to overexpress CTCF in the context of WGD to determine if rescuing CTCF levels would alter the changes in 3D genome structure they observe in WGD.

Indeed, rescuing CTCF (but not only) protein expression is a key experiment to demonstrate our hypothesis. As described in the previous response, we managed to rescue protein levels of both CTCF and H3K9me3 by treating the cells with Palbociclib. Importantly, Palbociclib rescued CTCF and H3K9me3 and restored boundary insulation and compartment segregation. We thus decided to adopt this approach rather than directly over-expressing CTCF, as we realized that loss of chromatin segregation was not driven by loss of CTCF alone and over-expressing CTCF would have not rescued H3K9me3. Moreover, constitutive over-expression of CTCF in the cells would influence all phases of the cell cycle in both diploid and tetraploid cells, and inducible CTCF expression would be difficult to timely regulate in cells undergoing WGD.

While the authors show that there are numerous CNVs, rearrangements, and compartment changes that occur in the tumors from mice, it isn't clear from the data how this is related to any of the observed changes in 3D genome organization that occur during WGD. In other words, the link between the first and second half of the manuscript isn't well developed. For example, does the loss of chromatin separation observed in WGD lead to increased likelihood of DNA damage or mitotic errors?

The reviewer asked an interesting but quite challenging question to address. Here we would like to briefly summarize our evidence connecting molecular changes in the tumors to those observed in WGD cells and, importantly, introduce new experiments and results that corroborated and strengthened this connection.

First, it has already been shown that WGD induces mitotic errors and chromosomal instability (Dewhurst et al., 2014). Here, for the first time, we have shown that WGD cells also fail to upscale CTCF and H3K9me3 levels, leading to what we termed loss of chromatin segregation (LCS). By keeping these cells in culture, we could see the emergence of both copy number alterations and chromatin compartment changes, which generated a highly heterogeneous cell population. While tracking the emergence and selection of these changes over time is extremely challenging, we were able to trace copy number changes in tumors originated from RPE^{TP53-/-} post-WGD cells to specific subclones that were already present in 6-weeks and 20-

weeks post-WGD cells (**Figure 4**). In addition, we found that most significant compartment repositioning events (CoREs) in these RPE-derived tumors could be traced back to compartment changes that emerged already in WGD cells, suggesting that LCS in WGD cells “primed” these regions to change compartment (**Figure 5n**).

To validate these observations, we have now introduced a new tumorigenic model using two independent CP-A^{TP53-/-} clones (clone 3 and clone 19). Importantly, WGD was induced separately in these two clones allowing to further assess the reproducibility of our results. Similar to what we have observed in RPE^{TP53-/-} cells post WGD, both CP-A^{TP53-/-} clones *in vitro* show mixed ploidy at 6 and 20 weeks post-WGD (**Supplementary Figure 5**). However, CP-A^{TP53-/-} cells were not able to grow *in vivo*. This was not surprising since lack of xenograft engraftment has been observed with many cancer cells and PDX models and this has been linked to the absence of human-specific growth factors. Indeed, CP-A^{TP53-/-} cells in culture require the addition of epidermal growth factor recombinant human protein, hydrocortisone, adenine, bovine pituitary extract, insulin-transferrin-sodium selenite, and other stimulatory factors. To cope with this issue, we decided to grow CP-A^{TP53-/-} cells in soft agar and assess the formation of colonies mimicking the uncontrolled growth that is typically observed in tumor cells. In this condition, we could clearly show that only cells that underwent WGD were able to grow and form large colonies (new **Figure 6a,b**).

Next, we selected 2 colonies for each clone (4 colonies in total) and performed Hi-C experiments to infer copy number alterations (CNA) and compartment repositioning events (CoREs). Importantly, all colonies acquired multiple CNAs and CoREs (new **Figure 6c,d**). Strikingly, in all 4 colonies, the vast majority of CoREs could be traced back to concordant compartment changes already occurring already at WGD, confirming our observations in RPE-derived tumors and suggesting that WGD primed genomic regions for compartment repositioning also in CP-A^{TP53-/-} clones (new **Figure 6e,f**). This result was further strengthened when we looked at the number of CoREs that were shared by the two individual colonies derived from each clone (shared CoREs). Indeed, the number of shared CoREs between two colonies was higher among CoREs that were “primed” at WGD compared to all CoREs (new **Figure 6g**). This result is consistent with compartment changes emerging early after WGD and retained in the majority of the cells and, hence, in multiple colonies.

These results not only validate our findings in an independent model but provide further evidence that link compartment repositioning occurring in tumors to loss of chromatin segregation in WGD cells.

Can the authors rule out that extended Nocodazole + dihydrocytochalasin B induces heritable changes in 3D genome structure independent of WGD? For example, its not apparent in the methods, but do the control cells undergo any kind of similar extended nocodazole treatment?

Dissecting the potential contribution of our protocol to induce WGD to the observed phenotype is indeed important. On the one hand, we need to clarify that we cannot treat control cells with nocodazole, as this treatment is used specifically to induce depolymerization of the mitotic spindle and favor WGD, i.e., treating control cells with nocodazole will in fact induce WGD.

On the other hand, both previous and new experiments now in the manuscript allow us to exclude the possibility that loss of chromatin segregation (LCS) is associated with nocodazole treatment. Indeed, we have now used the same protocol (nocodazole exposure) to induce WGD in CP-A *TP53* wild type cells, and in CP-A *TP53*^{-/-} cells treated with Palbociclib: in all these experiments we did NOT observe LCS (see new **Figure 2**).

In addition, we have induced WGD by cytokinesis failure in RPE^{*TP53*^{-/-}} cells using two different protocols, with and without nocodazole (nocodazole+DCB and CDK1i+DCB, see updated **Figure 1**) and in both cases, we observed LCS in WGD vs diploid cells.

Our results show that in *TP53*^{-/-} cells, a defective tetraploid checkpoint leads to reduced levels of both CTCF, which provides boundary insulation, and H3K9me3, which drives heterochromatin compaction and compartmentalization. These results clearly show that the LCS phenotype is NOT dependent on the protocol we adopted.

Very few plots show the actual Hi-C contact maps, and most instead show aggregated signals. I think it is important to show at least some examples to give the reader a better idea of the magnitude of the effects.

We have now included representative examples of Hi-C maps:

- 1) to visualize the effect of LCS on the contact maps (see new **ED Figure 2f**),
- 2) to show change of chromatin contacts in the CTNNB1 and XRCC5 loci in tumors derived from 20 weeks post-WGD RPE^{*TP53*^{-/-}} cells (see new **Figure 5i and 5l**).

Minor points:

They say that the WGD occurs due to “mitotic slippage” in the introduction. I’m not familiar with this term. Can they be more precise in describing what actually occurs in “mitotic slippage”.

Mitotic slippage is a common term used to describe cells that following mitotic arrest manage to survive and exit mitosis without dividing, hence without segregating the chromosomes (Sinha et al., 2019; Topham and Taylor, 2013).

I think that to really distinguish the reduced CTCF binding in Figure 2b, they need to do some kind of spike in control.

For the ChIP analyses, we adopted different methods of normalization, and for the CP-A experiments, we included mouse genome for spike-in normalization. Although, the results are consistent with the normalization over the input (**Rebuttal Fig. 7**), the spike-in experiment should be performed with different human/mouse genome ratios to obtain more robust scaling factors. However, we feel this is unnecessary at this point especially considering that downregulation of CTCF and H3K9me3 was also observed by western blot (**Figure 2**).

Rebuttal Fig. 7: CTCF ChIP-seq signal in control (X-axis) and WGD (Y-axis) cells either normalized over input (left) and with spike-in control (right)

Although the mechanism is different, during the cell cycle cells naturally double their DNA content, albeit temporarily and with sister chromatid cohesion. I believe there are existing Hi-C data in G2 synchronized cell publicly available. Is there any similar loss of 3D genome segmentation in G2 cells? This might be a nice control to evaluate whether simply doubling DNA content in the nucleus can have a similar effect.

While intriguing, we feel that comparing the chromatin of cells in G1 and G2 to assess LCS is not trivial. Indeed, H3K9me3 is known to be modulated during cell cycle and specifically in G2, among other things, to suppress histone gene expression before entry in mitosis (Alabert et al., 2015; Ito et al., 2012). Moreover, chromatin contacts distributions differ between G1 and G2 and compartmentalization has been shown to change in G1, S, and G2 phases before rapidly disappearing in mitosis (Nagano et al., 2017). Overall, G1 and G2 cells do not only differ for their DNA content, but also for specific cellular processes that alter their chromatin organization (even beyond those described here). For these reasons, this would not be a fair comparison and it cannot really be used to further validate the LCS phenotype that we characterized in multiple WGD cell models.

From the figures (Figure 4 and Ex. Fig. 8), it is hard to tell which tumors are derived from independent WGD experiments and which are from the same. Seemingly, the most interesting cases of CNV or other rearrangements would be events that are recurrent across independent WGD experiments. The authors mention they identify such events but don't really go into detail about what they are or what role they may be playing in contributing to tumorigenesis.

The 3 tumors presented in **Figure 4** are derived from the same experiment, while in what is now **ED Figure 8e**, 3 tumors are shown for each of 3 new experiments (labeled E2, E3, and E4). In the revised version of **Figure 4**, we have further improved the sample labeling to univocally identify the samples analyzed in this study. Regarding the potential selection of recurrent copy number alterations, we noticed in all samples an amplification of chromosome 1p where both the *JUN* and *NRAS* oncogenes are located, potentially explaining the selective advantage conferred by this event. While we discuss these events in the revised version of the manuscript, we preferred to maintain the main focus of the study on changes of chromatin 3D organization upon WGD. Nevertheless, the convergence of copy number profiles observed in these tumors are surely intriguing, even more in light of recent findings showing deterministic CNV pattern in *TP53*^{-/-} models (Baslan et al., 2022) and could be further explored in future studies.

The observation that CTCF is reduced in WGD cells but only in a p53 null background is pretty interesting, but it really begs the question of how this is regulated. Is CTCF a direct p53 target, and is it being regulated somehow in a stress responsive way (i.e. do non-WGD cells show changes in CTCF levels in response to p53 loss?).

We checked if CTCF could be a direct target of p53, but this was not the case. Indeed, by treating the cells with doxorubicin, which activates the p53 pathway, we could not induce CTCF expression (**Rebuttal Fig. 8**). As discussed already in the previous responses, *TP53*^{-/-} cells lack a tetraploid checkpoint and the rapid transition to G1/S limits the ability of the cells to upscale the level of several proteins including CTCF, H3K9me3, and other regulators of DNA replication. Indeed, stalling the cells in G1 allowed to rescue protein levels and chromatin segregation. We believe this explain the link between reduced levels of CTCF and lack of TP53.

Rebuttal Fig. 8: CTCF expression levels in RPE and CP-A *TP53* wild type cell lines after treatment with 3uM doxorubicin.

References

- Alabert, C., Barth, T.K., Reverón-Gómez, N., Sidoli, S., Schmidt, A., Jensen, O.N., Imhof, A., and Groth, A. (2015). Two distinct modes for propagation of histone PTMs across the cell cycle. *Genes Dev.* 29, 585–590. <https://doi.org/10.1101/gad.256354.114>.
- Andreassen, P.R., Lohez, O.D., Lacroix, F.B., and Margolis, R.L. (2017). Tetraploid State Induces p53-dependent Arrest of Nontransformed Mammalian Cells in G1. *MBoC* 12, 1315–1328. <https://doi.org/10.1091/mbc.12.5.1315>.
- Bakhom, S.F., and Cantley, L.C. (2018). The Multifaceted Role of Chromosomal Instability in Cancer and Its Microenvironment. *Cell* 174, 1347–1360. <https://doi.org/10.1016/j.cell.2018.08.027>.
- Baslan, T., Morris, J.P., Zhao, Z., Reyes, J., Ho, Y.-J., Tsanov, K.M., Bermeo, J., Tian, S., Zhang, S., Askan, G., et al. (2022). Ordered and deterministic cancer genome evolution after p53 loss. *Nature* 1–8. <https://doi.org/10.1038/s41586-022-05082-5>.
- Dewhurst, S.M., McGranahan, N., Burrell, R.A., Rowan, A.J., Grönroos, E., Endesfelder, D., Joshi, T., Mouradov, D., Gibbs, P., Ward, R.L., et al. (2014). Tolerance of Whole-Genome Doubling Propagates Chromosomal Instability and Accelerates Cancer Genome Evolution. *Cancer Discovery* 4, 175–185. <https://doi.org/10.1158/2159-8290.CD-13-0285>.
- Dixon, J.R., Jung, I., Selvaraj, S., Shen, Y., Antosiewicz-Bourget, J.E., Lee, A.Y., Ye, Z., Kim, A., Rajagopal, N., Xie, W., et al. (2015). Chromatin architecture reorganization during stem cell differentiation. *Nature* 518, 331–336. <https://doi.org/10.1038/nature14222>.
- Falk, M., Feodorova, Y., Naumova, N., Imakaev, M., Lajoie, B.R., Leonhardt, H., Joffe, B., Dekker, J., Fudenberg, G., Solovei, I., et al. (2019). Heterochromatin drives compartmentalization of inverted and conventional nuclei. *Nature* 570, 395–399. <https://doi.org/10.1038/s41586-019-1275-3>.
- Gemble, S., Wardenaar, R., Keuper, K., Srivastava, N., Nano, M., Macé, A.-S., Tijhuis, A.E., Bernhard, S.V., Spierings, D.C.J., Simon, A., et al. (2022). Genetic instability from a single S phase after whole-genome duplication. *Nature* 604, 146–151. <https://doi.org/10.1038/s41586-022-04578-4>.

- Ito, S., Fujiyama-Nakamura, S., Kimura, S., Lim, J., Kamoshida, Y., Shiozaki-Sato, Y., Sawatsubashi, S., Suzuki, E., Tanabe, M., Ueda, T., et al. (2012). Epigenetic Silencing of Core Histone Genes by HERS in *Drosophila*. *Molecular Cell* 45, 494–504. <https://doi.org/10.1016/j.molcel.2011.12.029>.
- Lieberman-Aiden, E., van Berkum, N.L., Williams, L., Imakaev, M., Ragoczy, T., Telling, A., Amit, I., Lajoie, B.R., Sabo, P.J., Dorschner, M.O., et al. (2009). Comprehensive mapping of long-range interactions reveals folding principles of the human genome. *Science* 326, 289–293. <https://doi.org/10.1126/science.1181369>.
- Liu, Y., Nanni, L., Sungalee, S., Zufferey, M., Tavernari, D., Mina, M., Ceri, S., Oricchio, E., and Ciriello, G. (2021). Systematic inference and comparison of multi-scale chromatin sub-compartments connects spatial organization to cell phenotypes. *Nature Communications* 12, 2439. <https://doi.org/10.1038/s41467-021-22666-3>.
- Margolis, R.L., Lohez, O.D., and Andreassen, P.R. (2003). G1 tetraploidy checkpoint and the suppression of tumorigenesis. *J Cell Biochem* 88, 673–683. <https://doi.org/10.1002/jcb.10411>.
- Mirny, L.A., Imakaev, M., and Abdennur, N. (2019). Two major mechanisms of chromosome organization. *Current Opinion in Cell Biology* 58, 142–152. <https://doi.org/10.1016/j.ceb.2019.05.001>.
- Nagano, T., Lubling, Y., Várnai, C., Dudley, C., Leung, W., Baran, Y., Mendelson Cohen, N., Wingett, S., Fraser, P., and Tanay, A. (2017). Cell-cycle dynamics of chromosomal organization at single-cell resolution. *Nature* 547, 61–67. <https://doi.org/10.1038/nature23001>.
- Olshen, A.B., Venkatraman, E.S., Lucito, R., and Wigler, M. (2004). Circular binary segmentation for the analysis of array-based DNA copy number data. *Biostatistics* 5, 557–572. <https://doi.org/10.1093/biostatistics/kxh008>.
- Padeken, J., Methot, S.P., and Gasser, S.M. (2022). Establishment of H3K9-methylated heterochromatin and its functions in tissue differentiation and maintenance. *Nat Rev Mol Cell Biol* 23, 623–640. <https://doi.org/10.1038/s41580-022-00483-w>.
- Rao, S.S.P., Huntley, M.H., Durand, N.C., Stamenova, E.K., Bochkov, I.D., Robinson, J.T., Sanborn, A.L., Machol, I., Omer, A.D., Lander, E.S., et al. (2014). A 3D map of the human genome at kilobase resolution reveals principles of chromatin looping. *Cell* 159, 1665–1680. <https://doi.org/10.1016/j.cell.2014.11.021>.
- Sanchez-Vega, F., Mina, M., Armenia, J., Chatila, W.K., Luna, A., La, K.C., Dimitriadoy, S., Liu, D.L., Kantheti, H.S., Saghafeina, S., et al. (2018). Oncogenic Signaling Pathways in The Cancer Genome Atlas. *Cell* 173, 321–337.e10. <https://doi.org/10.1016/j.cell.2018.03.035>.
- Santaguida, S., Richardson, A., Iyer, D.R., M'Saad, O., Zasadil, L., Knouse, K.A., Wong, Y.L., Rhind, N., Desai, A., and Amon, A. (2017). Chromosome Mis-segregation Generates Cell-Cycle-Arrested Cells with Complex Karyotypes that Are Eliminated by the Immune System. *Dev Cell* 41, 638–651.e5. <https://doi.org/10.1016/j.devcel.2017.05.022>.
- Sinha, D., Duijf, P.H.G., and Khanna, K.K. (2019). Mitotic slippage: an old tale with a new twist. *Cell Cycle* 18, 7–15. <https://doi.org/10.1080/15384101.2018.1559557>.
- Topham, C.H., and Taylor, S.S. (2013). Mitosis and apoptosis: how is the balance set? *Current Opinion in Cell Biology* 25, 780–785. <https://doi.org/10.1016/j.ceb.2013.07.003>.

Reviewer Reports on the First Revision:

Referees' comments:

Referee #1 (Remarks to the Author):

I think the additional analysis of trans contacts in response to my second major point (including rebuttal Figure 1) will be of interest to readers and should be included in the manuscript (perhaps as a supplementary figure with a long-ish caption, alluded to briefly in the main text). I agree with the argument given that the "false in cis" contacts are unlikely to affect the main conclusions of this study.

I don't agree that the way the O/E calculation is done is "quite simple," as claimed. For example, step 1 reads like this: "compute the matrix of observed vs. expected contacts (O/E matrix) between each pair of chromosomes (the expected number of contacts accounts for the chromosome length and the total number of contacts made by that chromosome)." This description is insufficiently detailed to reproduce the calculation, and furthermore the notion of "expected" here differs from the standard one (which conditions on genomic distance between cis contacts). How exactly is the denominator computed? Furthermore, my understanding is that, for two chromosomes of length n and m , respectively, this step yields an n -by- m matrix. So then step two ("compute the Pearson correlation between each pair of rows of the O/E matrix") is not symmetric; you will get a different result for chromosome A vs. B compared to B vs. A. And finally, the figure shows a single value per pair of chromosomes, so presumably there is a missing step in which all of the Pearson correlations are averaged. The authors then claim that they can use this analysis to somehow "test if each pair of chromosomes interacts more or less frequently than expected." This sounds like a hypothesis test, but I don't know how such a test would be performed. Please add a complete description of how this is done to Methods.

Thank you for doing the changepoint analysis; I think it helps to make the story more convincing.

I appreciated the further explanation for ED Figure 9. I noticed that the main text claims that "both genome wide sub-compartment changes and CoREs were highly correlated with changes of histone mark intensities," but the figure itself only shows scatter plots. I recommend adding the Pearson and/or Spearman correlation to each panel, to directly support the claim in the text.

Referee #2 (Remarks to the Author):

The authors have done an excellent job of addressing my initial concerns with the manuscript. I would suggest the authors should include the results regarding post-WGD. I think this is a very important analysis. If there is scope, I would consider looking at additional time-points.

I have no additional comments.

Referee #3 (Remarks to the Author):

The authors adequately addressed most of my comments. In particular, the authors made an effort to address my three main concerns:

(1) The revised version of the manuscript shows much more convincingly that the changes in chromatin interactions are largely independent of the chromosomal instability (CIN) that characterizes WGD cells/tumors. This is a very important point, and the manuscript has certainly improved in this regard.

(2) The revised manuscript presents gene expression comparison of diploid and tetraploid TP53^{-/-}RPE1 cells. The authors found the expected transcriptional changes, but didn't find any substantial effect of loss of boundary insulation and/or compartment segregation on gene expression. I think these negative results should be shown in the manuscript (and not merely in the Rebuttal Letter; also see my general note below). Since a key conclusion of the paper is that WGD promotes malignant transformation by (also) altering the chromatin 3D organization, the lack of expression changes related to loss of boundary insulation and compartment segregation is notable, potentially suggesting that most of the functional effect is mediated through the CoREs (as is shown in Fig. 5). This should be at least discussed.

In this respect, the new scRNAseq data analysis is informative and supports the notion that compartment repositioning underlies some (non-negligible fraction) of the transcriptional changes induced by WGD.

(3) The revised manuscript used additional methods for WGD induction, alleviating the generalizability concern (and also the concern that nocodazole treatment underlies some of the observed changes).

A general note: in multiple occasions, the authors present Rebuttal Figures, stating that they "prefer not to include them in the manuscript" (or not stating anything). In my mind, most (or all) of these data should be included in the manuscript, as readers will find them interesting. (Assuming that Readers might have similar doubts/concerns/questions as Reviewers, I don't find much point in Rebuttal Figures; they should be the exception, not the rule.)

Referee #4 (Remarks to the Author):

Lambuta et al. present a revised manuscript on their study of the impact of whole genome doubling (WGD) on 3D genome organization and oncogenesis. They have really done a tremendous amount of work as part of this revision, including new experiments, additional controls, and new analyses. I think that the Pablociclib rescue experiments are an important addition and begin to indicate the underlying mechanisms governing their observations on loss of chromatin segregation. With these revisions, I believe that the study is likely now suitable for publication in Nature. I do still have some very minor comments. I have to say that I don't think any of the comments I am raising should be

any kind of barrier to this study finally being accepted for publication, as the comments I have are all quite minor and are more related to phrasing or presentation. Overall, I would commend the authors on their excellent work and I am excited to see it in print.

Minor comments:

-The abstract is a little dense and seems to be too reliant on conveying details, I think it would be better for a general reader if it reflected more of the big picture topics mentioned in the first paragraph of the intro in the main text.

-Very minor, but line 170 they say that WGD cells exhibited an “important reduction (~50%)” of CTCF and H3K9me3. It is unclear to me what the word “important” is in this sentence. Like what would distinguish an important vs. unimportant reduction? It seems to me that adding subjective words like this should be avoided, they could just say that there was a reduction.

-I think in the CP-A with WT TP53 or with the CDK4/6i treatment (Fig. 2h, i), the differences are pretty clear, but the relevant comparison plot is in figure 1 (Fig.1g,h). Is it possible to have the same plot in panel 2 showing the CP-A TP53-/- WGD vs. control plot (Fig1g,h) also in Fig. 2, it would make it easier to draw the clear comparison?

-Related to Figure 5 and the CoREs analysis. This is interesting and the data look convincing, but it is important to consider that the control here is both not WGD and grown in culture plates. I think it is worthwhile for the authors to mention this limitation, I don't think they can really tell whether the observed differences are due to WGD vs. tumorigenesis vs. growth in a different environment (in vivo in a mouse). I think it is worth mentioning this limitation.

Author Rebuttals to First Revision:

Referee #1 (Remarks to the Author):

I think the additional analysis of trans contacts in response to my second major point (including rebuttal Figure 1) will be of interest to readers and should be included in the manuscript (perhaps as a supplementary figure with a long-ish caption, alluded to briefly in the main text). I agree with the argument given that the "false in cis" contacts are unlikely to affect the main conclusions of this study.

We have now included this additional analysis in the manuscript (Extended Data Fig. 3f) and included a description of the analysis in the Methods ("Analysis of contacts between homologous chromosomes after WGD").

I don't agree that the way the O/E calculation is done is "quite simple," as claimed. For example, step 1 reads like this: "compute the matrix of observed vs. expected contacts (O/E matrix) between each pair of chromosomes (the expected number of contacts accounts for the chromosome length and the total number of contacts made by that chromosome)." This description is insufficiently detailed to reproduce the calculation, and furthermore the notion of "expected" here differs from the standard one (which conditions on genomic distance between cis contacts). How exactly is the denominator computed? Furthermore, my understanding is that, for two chromosomes of length n and m , respectively, this step yields an n -by- m matrix. So then step two ("compute the Pearson correlation between each pair of rows of the O/E matrix") is not symmetric; you will get a different result for chromosome A vs. B compared to B vs. A. And finally, the figure shows a single value per pair of chromosomes, so presumably there is a missing step in which all of the Pearson correlations are averaged. The authors then claim that they can use this analysis to somehow "test if each pair of chromosomes interacts more or less frequently than expected." This sounds like a hypothesis test, but I don't know how such a test would be performed. Please add a complete description of how this is done to Methods.

We apologize for not having provided a detailed-enough description of the method. We now introduced a thoroughly expanded description of the procedure in the Methods ("Pseudo-bulk single-cell Hi-C analysis") and we also briefly explain here the procedure.

The observed vs. expected matrix is a 23×23 matrix where each entry is the ratio of the observed number of contacts between the two chromosomes and the corresponding expected number of contacts. Since chromosomes have different lengths, the expected number of contacts between two pairs of chromosomes can be very different among different pairs. To take this into account, we use the same strategy as in (Liebermann et al. 2009): given two chromosomes A and B, the expected number of contacts is computed as the product between the total number of inter-chromosomal interactions made by chromosome A and the total number of inter-chromosomal interactions made by chromosome B, divided by twice the total number of inter-chromosomal interactions. Finally, given the sparsity of the data in the single-cell Hi-C dataset, we computed the Spearman correlation coefficient between each pair of rows/chromosomes in the observed vs. expected matrix, thus obtaining a 23×23 correlation matrix between chromosomes (see the first two heatmaps of Extended Data Figure 6a). When comparing the correlation matrices of control and WGD pseudo-bulk Hi-C we

added 1 to both correlation values (since correlations span between -1 and 1), such that a meaningful ratio could be computed for each pair of chromosomes (see last plot of Extended Data Figure 6a).

Thank you for doing the changepoint analysis; I think it helps to make the story more convincing.

I appreciated the further explanation for ED Figure 9. I noticed that the main text claims that "both genome wide sub-compartment changes and CoREs were highly correlated with changes of histone mark intensities," but the figure itself only shows scatter plots. I recommend adding the Pearson and/or Spearman correlation to each panel, to directly support the claim in the text.

We have now added the correlation values in each scatterplot (Extended Data Fig. 10f).

Referee #2 (Remarks to the Author):

The authors have done an excellent job of addressing my initial concerns with the manuscript. I would suggest suggest the authors should include the results regarding post-WGD. I think this is a very important analysis. If there is scope, I would consider looking at additional time-points.

I have no additional comments.

We wish to thank the reviewer for the support and appreciating our revised work.

Referee #3 (Remarks to the Author):

We wish to that the reviewer for the support and appreciating our revised work. As detailed below, we have now addressed the few remaining concerns.

The authors adequately addressed most of my comments. In particular, the authors made an effort to address my three main concerns:

(1) The revised version of the manuscript shows much more convincingly that the changes in chromatin interactions are largely independent of the chromosomal instability (CIN) that characterizes WGD cells/tumors. This is a very important point, and the manuscript has certainly improved in this regard.

(2) The revised manuscript presents gene expression comparison of diploid and tetraploid TP53-/- RPE1 cells. The authors found the expected transcriptional changes, but didn't find any substantial effect of loss of boundary insulation and/or compartment segregation on gene expression. I think these negative results should be shown in the manuscript (and not merely in the Rebuttal Letter; also see my general note below). Since a key conclusion of the paper is that WGD promotes malignant transformation by (also) altering the chromatin 3D organization, the lack of expression changes related to loss of boundary insulation and compartment segregation is notable, potentially suggesting that most of the functional effect is mediated through the CoREs (as is shown in Fig. 5). This should be at least discussed.

We have now included the results related to the association between LCS and expression changes observed in WGD cells (Extended Data Fig. 4d,e).

In this respect, the new scRNAseq data analysis is informative and supports the notion that

compartment repositioning underlies some (non-negligible fraction) of the transcriptional changes induced by WGD.

(3) The revised manuscript used additional methods for WGD induction, alleviating the generalizability concern (and also the concern that nocodazole treatment underlies some of the observed changes).

A general note: in multiple occasions, the authors present Rebuttal Figures, stating that they “prefer not to include them in the manuscript” (or not stating anything). In my mind, most (or all) of these data should be included in the manuscript, as readers will find them interesting. (Assuming that Readers might have similar doubts/concerns/questions as Reviewers, I don’t find much point in Rebuttal Figures; they should be the exception, not the rule.)

We agree with the reviewer, and we have now included all Rebuttal Figures in the manuscript as Extended Data Figure panels. In addition, we also agree with the new Nature policy to publish the entire revision history.

Referee #4 (Remarks to the Author):

Lambuta et al. present a revised manuscript on their study of the impact of whole genome doubling (WGD) on 3D genome organization and oncogenesis. They have really done a tremendous amount of work as part of this revision, including new experiments, additional controls, and new analyses. I think that the Pablociclib rescue experiments are an important addition and begin to indicate the underlying mechanisms governing their observations on loss of chromatin segregation. With these revisions, I believe that the study is likely now suitable for publication in Nature. I do still have some very minor comments. I have to say that I don’t think any of the comments I am raising should be any kind of barrier to this study finally being accepted for publication, as the comments I have are all quite minor and are more related to phrasing or presentation. Overall, I would commend the authors on their excellent work and I am excited to see it in print.

We thank the reviewer for their enthusiastic support, and we have now further addressed the remaining minor comments they raised.

Minor comments:

-The abstract is a little dense and seems to be too reliant on conveying details, I think it would be better for a general reader if it reflected more of the big picture topics mentioned in the first paragraph of the intro in the main text.

In this revised version of the manuscript, we have edited the abstract to remove more technical points and focus on the main concepts introduced in this study.

-Very minor, but line 170 they say that WGD cells exhibited an “important reduction (~50%)” of CTCF and H3K9me3. It is unclear to me what the word “important” is in this sentence. Like what would distinguish an important vs. unimportant reduction? It seems to me that adding subjective words like this should be avoided, they could just say that there was a reduction.

We have now removed “important” from the text.

-I think in the CP-A with WT TP53 or with the CDK4/6i treatment (Fig. 2h, i), the differences are pretty clear, but the relevant comparison plot is in figure 1 (Fig.1g,h). Is it possible to have the same plot in

panel 2 showing the CP-A TP53^{-/-} WGD vs. control plot (Fig1g,h) also in Fig. 2, it would make it easier to draw the clear comparison?

As the reviewer correctly points out, **Fig. 2h,i** should be compared to **Fig. 1g,h** (CP-A TP53^{-/-} WGD vs. control). However, to introduce this panel in Fig. 2 as well would mean duplicating a figure panel, which we would rather avoid. We made clear in the text which figure panels should be compared.

-Related to Figure 5 and the CoREs analysis. This is interesting and the data look convincing, but it is important to consider that the control here is both not WGD and grown in culture plates. I think it is worthwhile for the authors to mention this limitation, I don't think they can really tell whether the observed differences are due to WGD vs. tumorigenesis vs. growth in a different environment (in vivo in a mouse). I think it is worth mentioning this limitation.

We amended the text to mention this limitation.

Reviewer Reports on the Second Revision:

Referees' comments:

Referee #1 (Remarks to the Author):

The authors have addressed all of my remaining concerns.